# Exonuclease-enhanced prime editors

**Dong-Jiunn Jeffery Truong** [1,2,4], **Julian Geilenkeuser** [1,2,4],
**Stephanie Victoria Wendel** [1,2], **Julius Clemens Heinrich Wilming** [1,2],
**Niklas Armbrust** [1,2], **Eva Maria Hildegard Binder** [1,2], **Tobias Heinrich Santl**[1,2],
**Annika Siebenhaar** [1,2], **Christoph Gruber**[3], **Teeradon Phlairaharn** [1,2],
**Milica Živanić** [1,2] & **Gil Gregor Westmeyer** [1,2] ✉

Prime editing (PE) is a powerful gene-editing technique based on targeted gRNA-templated reverse transcription and integration of the de novo synthesized single-stranded DNA. To circumvent one of the main bottlenecks of the method, the competition of the reverse-transcribed 3′ flap with the original 5′ flap DNA, we generated an enhanced fluorescence-activated cell sorting reporter cell line to develop an exonuclease-enhanced PE strategy ('Exo-PE') composed of an improved PE complex and an aptamer-recruited DNA-exonuclease to remove the 5′ original DNA flap. Exo-PE achieved better overall editing efficacy than the reference PE2 strategy for insertions ≥30 base pairs in several endogenous loci and cell lines while maintaining the high editing precision of PE2. By enabling the precise incorporation of larger insertions, Exo-PE complements the growing palette of different PE tools and spurs additional refinements of the PE machinery.

PE entails the reverse transcription of a Cas9-bound RNA template at a targeted DNA site, successful insertion of the generated 3′ flap and repair of the DNA locus. The initial demonstration of PE by Anzalone et al.[1] was a remarkable achievement given the complex spatial arrangement and orchestration of the different molecular functions, the delicate ratio of affinities between self-complementary PE gRNA (pegRNA) and primer-binding site (PBS), and the downstream cellular processes resulting in the incorporation of the intended edit.

One of the critical bottlenecks emphasized in the original publication[1] is the need for the de novo synthesized 3′ DNA flap to outcompete the original 5′ end, which is, in particular, disfavored for longer stretches.

The DNA repair machinery has to subsequently maintain the desired edit in lieu of the original sequence. The PE3 strategy attempts to bias the outcome towards the desired edit by causing a secondary nick on the unedited strand, flagging it for repair. However, this strategy carries the risk of an increased rate of insertions and deletions (indels). The PE4 strategy inhibits DNA mismatch repair (MMR), although it seems to be less effective for longer edits[2]. PE5 combines both of these strategies simultaneously.

Furthermore, alternative PE strategies use paired pegRNAs to generate two de novo 3′ DNA flaps that anneal together to replace the original segment. These 'paired PE' strategies have been customized for different types of edits (short substitutions, insertions or large deletions)[3]. However, using a secondary nick in the paired PE strategy still poses the danger of unintentionally creating a staggered double-strand break (DSB).

Here, we describe developing and optimizing a complementary PE strategy called Exo-PE, which involves recruiting a 5′–3′ exonuclease to the editing site. This active recruitment generates an engineered gap for the invasion of the de novo synthesized 3′ flap, thereby relieving the bottleneck of flap competition and thus enabling also larger inserts without a secondary nick.

## Results

### Fluorescent reporter line for gene-editing events

To improve the precision and throughput of our gene-editing studies, we first developed a fluorescence-activated cell sorting (FACS) screening system that comprehensively captures both successful PE- or homology-directed repair (HDR)-mediated

[1]Institute for Synthetic Biomedicine, Helmholtz Munich, Neuherberg, Germany. [2]Department of Bioscience, TUM School of Natural Sciences and TUM School of Medicine,Technical University of Munich, Munich, Germany. [3]Institute of Developmental Genetics, Helmholtz Munich, Neuherberg, Germany. [4]These authors contributed equally: Dong-Jiunn Jeffery Truong, Julian Geilenkeuser. ✉e-mail: gil.westmeyer@tum.de

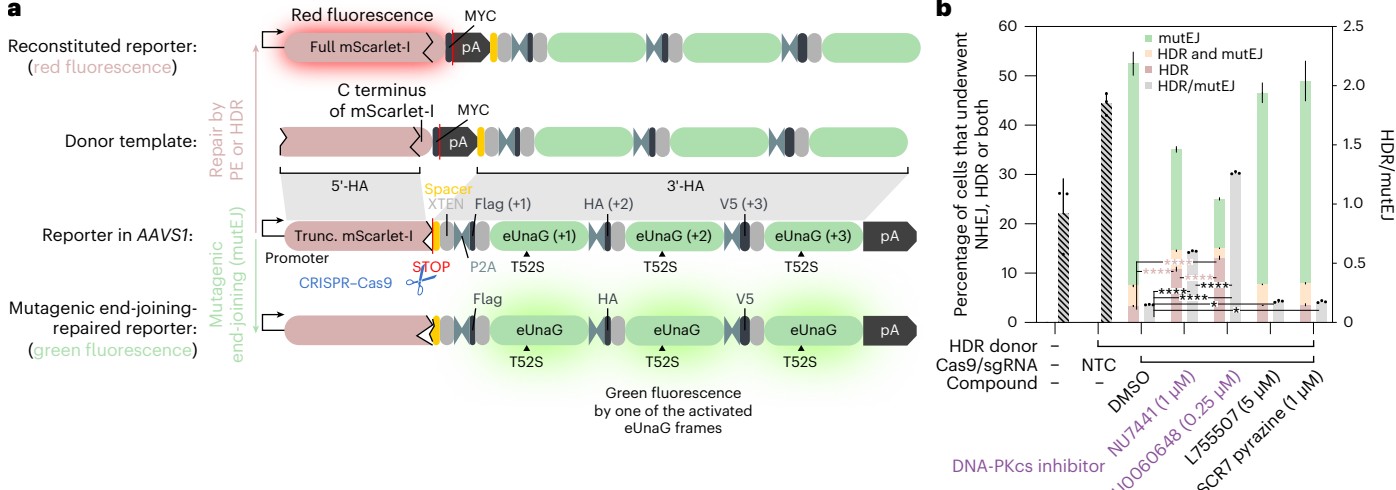

**Fig. 1 | Development of an eTLR to monitor all editing events. a**, eTLR is based on C-terminally truncated (Trunc.) mScarlet-I followed by a stop codon and a concatenation via P2A sites of three different frames coding for the green fluorescent protein eUnaG$_{T52S}$. Out-of-frame ATGs were removed to prevent unintended translation initiation. Disruption of the stop codon via, for example, NHEJ/MMEJ/SSA results in readthrough and the activation of one of the eUnaG frames (eUnaG (+1/2/3)) (green fluorescence). Instead, successful repair via PE or HDR results in full-length mScarlet-I (red fluorescence). Cells with a biallelic integration of eTLR were analyzed via FACS, where red fluorescence reports successful HDR or PE, while green fluorescence indicates a mutagenic end-joining event. A signal on both the red and green channels (shown in orange) indicates that both events have occurred, each on a separate allele. **b**, HEK293 cells carrying biallelic copies of the eTLR system in *AAVS1* were transfected with a promoterless repair template for mScarlet-I and an all-in-one CRISPR–Cas9 plasmid targeting eTLR. At 16 h after transfection, the indicated compounds were added to the cells. At 3 days after transfection, cells were analyzed for red and green fluorescence by flow cytometry to quantify the effect of the compounds on the repair outcome. Colored bars quantify the fraction of cells with green, red, or both (orange) signals representing the different editing outcomes (left *y* axis), whereas the gray bars (right *y* axis) display the ratio of HDR over mutagenic end-joining events as a measure of editing precision. Selected results of a Bonferroni MCT after one-way ANOVA were shown for the HDR events as well as editing precision (HDR/mutEJ) and are indicated by asterisks; *$P < 0.05$, ****$P < 0.0001$. Bars, mean ± s.d. (*n* = 3 biological replicates). Please note that only very few red or green events were recorded by FACS for the condition without donor and Cas9, and the non-targeting control (NTC) such that the corresponding HDR/mutEJ ratios are not informative (shaded bars).

outcomes as well as mutagenic end-joining (mutEJ) events leading to indels (Fig. 1).

In distinction to previous systems that could only report approximately one-third of unintended edits[4], our enhanced traffic light reporter (eTLR) system reports all frameshifts resulting from indels at the target site. Specifically, the reporter design allows the C-terminally truncated red fluorescent mScarlet to be restored by PE or HDR, whereas any indels occurring in any of the frames are detected via the expression of the small green fluorescent protein enhanced UnaG (eUnaG)[5] enabled by translational readthrough into one of the frames (Fig. 1a). We removed all out-of-frame stop codons by synonymous codon replacements, except for M51-T52 (ATG–ACN), where neither methionine nor threonine could be synonymously substituted to remove the +2 frame *opal* stop codon. Therefore, T52S had to be introduced into eUnaG to remove this out-of-frame stop codon. Moreover, to minimize leaky background fluorescence, we removed all out-of-frame start codons (ATG) to prevent translation initiation by cryptic promoters (Extended Data Fig. 1a,b). If desired, it would also be possible to decode the exact frame by immunofluorescence against the frame-specific epitope tags (Flag, HA, V5).

We first confirmed the functionality of the three reading frames by creating −1/−2/−3 nucleotide deletions to simulate mutagenic end-joining events at the target site, which led to visible green fluorescence (Extended Data Fig. 1c).

To validate that the degree of green or red fluorescence of eTLR reflect the different genome editing outcomes, we used CRISPR–Cas9-driven HDR of the truncated mScarlet-I, whose efficiency was modulated by established pharmacological compounds. We transfected a clonal cell line carrying biallelic copies of the eTLR system, with CRISPR–Cas9 components directed against the editing site together with a promoterless HDR template containing the 54 nucleotide missing C terminus of mScarlet-I and a new polyadenylation site (Fig. 1b and Extended Data Fig. 1d).

The biallelic eTLR cell line can thus distinguish three different outcomes: red fluorescent cells indicate successful homologous recombination, and green cells indicate mutagenic end-joining events. Detection of both red and green fluorescence for one cell in FACS analysis (Methods) indicates that one allele was repaired via PE (or HDR), whereas the second reporter copy on the other allele was repaired by mutagenic end-joining (mutEJ), such as nonhomologous end-joining. The fraction of red fluorescent cells was used to determine the desired editing efficacy, and the ratio of total red to green cells was used to measure editing precision[4].

**Optimization of the PE enzyme and its nuclear localization**

First, we used the eTLR system to generate an improved PE (iPE) by optimizing the protein component (Fig. 2). Specifically, we incorporated an enhanced nuclear localization sequence (NLS) motif (superNLS) (Fig. 2a,b) and a codon-optimized reverse transcriptase (RT). Moreover, we removed several potential splice sites contained in the original unoptimized prime editor[1], which may lead to mRNA mis-splicing[6] and nuclear retention by splice factors (Fig. 2a, marked in red). We also re-examined which terminus of the Cas9 nickase (nCas9) is optimal for the RT fusion, comparing a version with RT fused C-terminally (iPE-C), and one with RT fused N-terminally (iPE-N) (Fig. 2a).

We then benchmarked iPE-N/C against PEmax[2], which contains a C-terminally-fused codon-optimized RT domain, additional mutations to nCas9 (R221K + N394K)[7], and improved nuclear localization. iPE-N and iPE-C showed an increased editing efficacy over PEmax in a PE3 setting using an engineered pegRNA (epegRNA) with a 3′-protection motif (tevopreQ1)[2] (Fig. 2c).

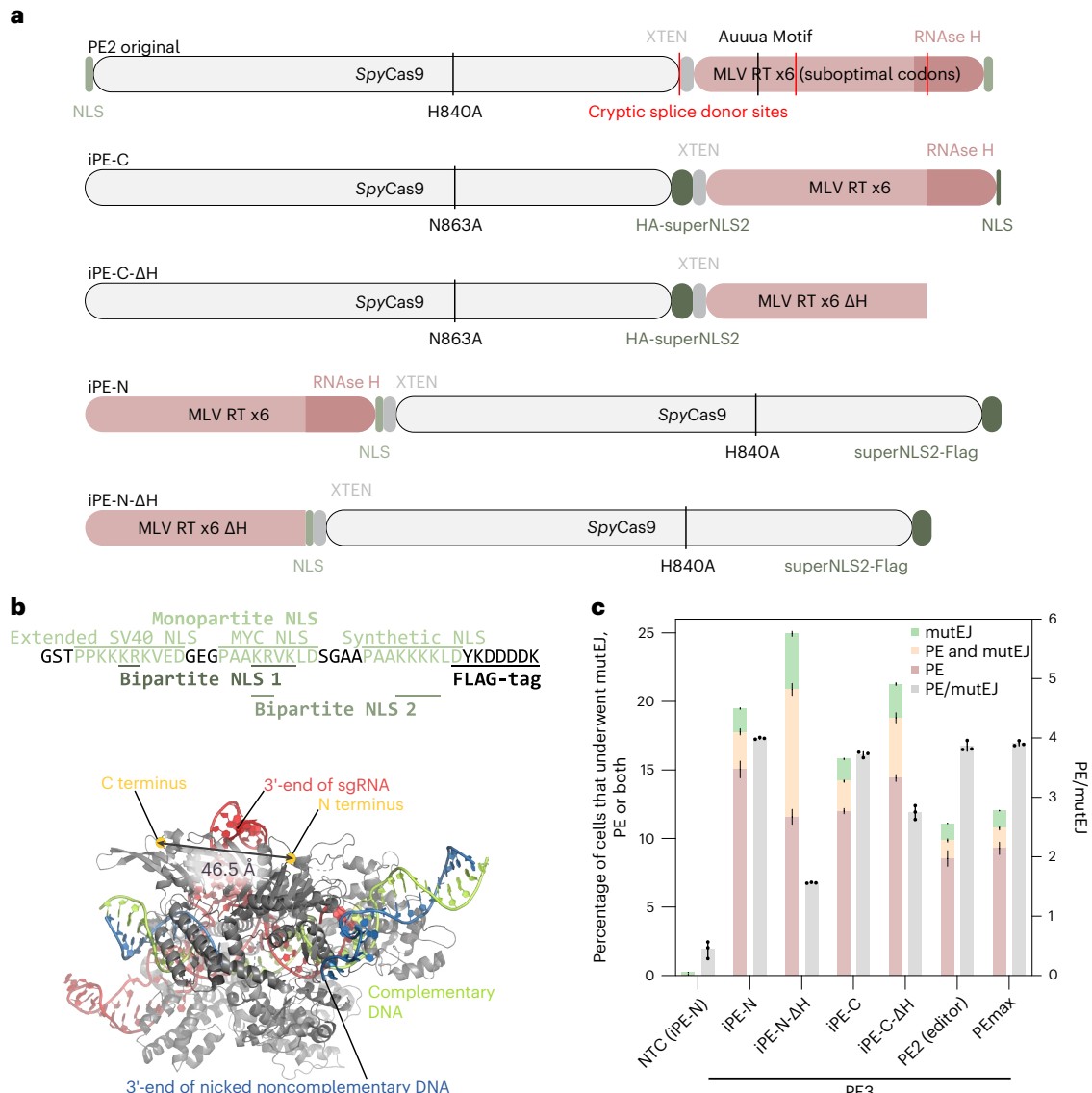

**Fig. 2 | Development of an iPE in combination with engineered/protected pegRNA containing an RNA aptamer. a**, Schematic representation of the original PE construct from ref. 1 (top) and optimized PE constructs. Cryptic splice donor sites were detected using NetGene2 (ref. 6). **b**, Top, depiction of the superNLS, harboring three mono- and two bipartite NLS sequences. Bottom, structure of the ternary Cas9–sgRNA–DNA complex with nicked nontargeting DNA (PDB, 6VPC). **c**, Comparison of the original PE2 editor and the optimized PEmax[2] with iPE with RT fused N-terminally or C-terminally (iPE-N and iPE-C) and its ΔRNAse H variants, combined with a 3'-tevopreQ1-modified pegRNA, in a PE3 setup. Bars represent mean ± s.d. (*n* = 3 biological replicates).

In the eTLR system, we also found that deleting the RNAse H domain of RT (iPE-N/C-ΔH) increased the indel frequency and thus reduced the precision for the N-terminal configuration, whereas the same deletion had only an attenuated effect in the C-terminal fusion (Fig. 2c).

The original PE2 enzyme[1] was not significantly worse than PEmax when combined with engineered pegRNAs (Fig. 2c). However, when pegRNAs without tevopreQ1 were used, the original PE2 enzyme performed much worse compared with iPE-N in both the PE2 and PE3 strategy (Extended Data Fig. 2a; PE2, *P* = 0.0007; PE3, *P* < 0.0001; two-tailed unpaired *t*-test), indicating optimized enzymes can compensate for suboptimal pegRNAs.

When further comparing iPE-N and iPE-C on eTLR, we found that iPE-N had a slightly higher editing efficacy for the PE3 and paired PE strategy than the C-terminal fusion (iPE-C; Extended Data Fig. 2b; red bars, *P* = 0.0222; one-way analysis of variance (ANOVA) with Bonferroni multiple comparison test (MCT)) also when the RT was exchanged

with MarathonRT (Extended Data Fig. 2c; *P* < 0.0001; two-tailed unpaired *t*-test).

We have thus identified optimal iPE proteins with an N- or C-terminal fusion of RT to nCas9 (iPE-N or iPE-C) featuring an improved NLS, linker length, and codon usage.

## Recruiting an exonuclease to the PE complex

We next assessed whether installing a 5′-exonuclease directly on iPE could more efficiently free up space for inserting the de novo synthesized 3′ flap, comprising the desired edit and the invasion/homology region. We searched for highly active 5′-DNA-exonucleases/flap endonucleases and identified T5-bacteriophage and T5-like 5′–3′-DNA-exonucleases[8,9], which also possess flap endonuclease activity as potential candidates for creating engineered gaps at the primary PE site. Since both termini of *Streptococcus pyogenes* Cas9 are located on the opposite side of the cleft where the target DNA is bound and nicked (Fig. 2b), we reasoned that a direct fusion of the exonuclease might not

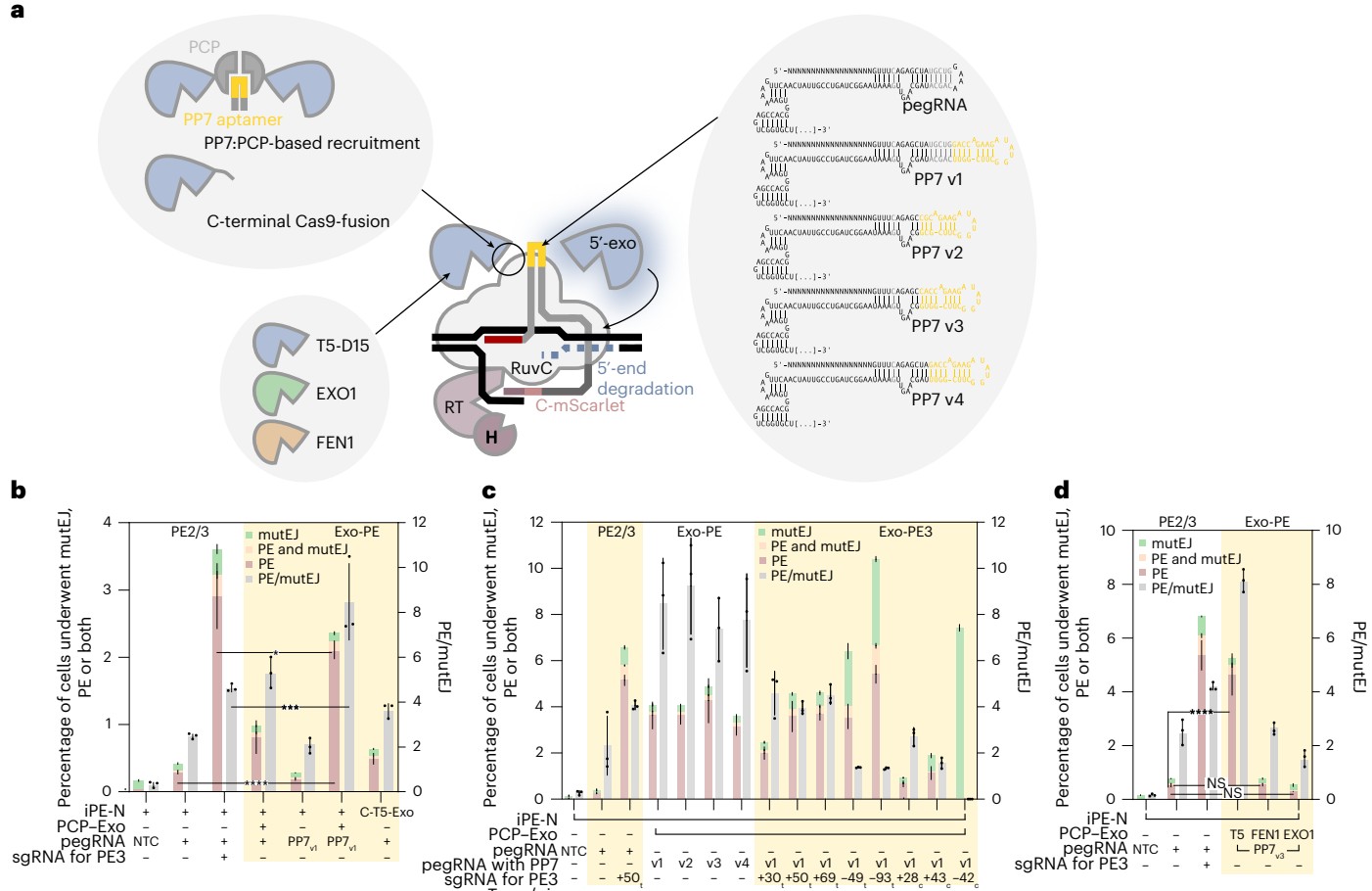

**Fig. 3 | Development of an Exo-PE strategy using eTLR. a**, Schematic of different Exo-PE designs varying in recruitment mechanism, choice of specific exonuclease, and version of the PP7 aptamer grafted into the pegRNA scaffold. **b**, Comparison between PE2, PE3 (white graph area) and 5c-DNA-exonuclease-enhanced PE2 (Exo-PE, yellow graph area). Selected results of Bonferroni MCT after one-way ANOVA were shown for the fractional PE events and editing precision, and are indicated by asterisks; *$P < 0.05$, ***$P < 0.001$, ****$P < 0.0001$. Bars, mean ± s.d. ($n = 3$ biological replicates). **c**, Effects of different PP7/pegRNA scaffold designs (shown in **a** and the impact of an additional up- or downstream

cis- or trans-acting nicking sgRNA on the efficacy and precision of Exo-PE. Bars, mean ± s.d. ($n = 3$ biological replicates). **d**, Alternative PCP fusions with human-derived 5′-DNA-exonucleases/flap endonucleases domains from FEN1 and EXO1 without their native C-terminal interaction peptides were tested as an alternative to the bacteriophage 5′-DNA-exonuclease/flap endonuclease. Selected results of Bonferroni MCT after one-way ANOVA were shown for the fractional PE events, and are indicated by asterisks; NS $P > 0.05$, ****$P < 0.0001$. Bars, mean ± s.d. ($n = 3$ biological replicates).

be an optimal configuration and therefore opted for aptamer-based recruitment[10]. We thus introduced the *Pseudomonas* phage PP7 aptamer[11] into the tetraloop junction between the crRNA and the tracr-RNA. We also fused the 5′–3′-exonuclease to the PP7 aptamer binding coat protein (PCP) together with an NLS (PP7:PCP-based recruitment; Fig. 3a). When coexpressing the PP7-tagged pegRNA and the PCP–NLS–5′-Exo, a substantial sevenfold increase in efficacy compared with PE2 could be observed ($P < 0.0001$; one-way ANOVA all versus all excluding nontargeting controls; Bonferroni MCT; Fig. 3b), whereas, without exonuclease recruitment (using a pegRNA lacking the PP7 aptamer), efficacy dropped to only twofold above PE2 level. By contrast, a direct fusion of the 5′–3′-DNA-exonuclease to the C-terminus of nCas9 (C-T5-Exo, last bars) barely improved the editing efficacy. Compared with the PE3 strategy, the editing efficacy of Exo-PE was slightly lower ($P = 0.0149$) but with a higher editing precision ($P = 0.0008$; one-way ANOVA all versus all excluding non-targeting controls; Bonferroni MCT).

Subsequently, we refined the PP7 aptamer-tagged pegRNA for Exo-PE, and continued with PP7 v3 (Fig. 3a, right schematic), which showed a trend towards increased efficacy while maintaining improved precision (Fig. 3c). Additional modifications of iPE were shown to be beneficial in a previous study (a mutation in nCas9 (K918N) and a histone-like dsDNA binding protein (Sso7d) (iPE$_{K918N,Sso7d}$, see sketch in Extended Data Fig. 3a))[12,13], but were not consistently shown to be beneficial, and were therefore omitted them from subsequent experiments. Other conditions such as the combination of Exo-PE with PE3 (Exo-PE3) (Fig. 3c), catalytically inactive sgRNAs[14] (dsgRNA; Extended Data Fig. 2d), ssDNA-binding domains (Pot1pC[15]) (Extended Data Fig. 3b), or replacement of phage-derived exonucleases with human exonucleases FEN1 and EXO1 (Fig. 3d) did not yield any additional benefit. More interestingly, the T5-like exonuclease from *Klebsiella pneuomoniae* siphophage Sugarland was nearly as effective for Exo-PE as T5 (80% homology of T5; Extended Data Fig. 3b) when tevopreQ1 (Q1) was used to increase the steady-state concentration of full-length pegRNAs[16] ($P = 0.0221$, ANOVA all versus iPE-N$_{K918N}$ (ref. 12) + T5 with Bonferroni MCT). By contrast, T7 exonuclease showed a decreased efficacy ($P < 0.0001$). Thus, we conclude that the best 5′-exonuclease for iPE used with the Exo-PE strategy remained the T5 phage exonuclease C-terminally fused to PCP.

We also re-examined whether further optimization of the pegRNAs could improve Exo-PE performance even more. For our initial experiments, we already used an optimized sgRNA scaffold in which

we removed the polyT/U-stretch and introduced either an RNA folding nucleation site via an extended neutral hairpin[17] or the PP7 aptamer hairpin to improve RNA folding. Removing the 3′-tRNA from the pegRNA (which cleaves off the 3′-uridine stretch via endogenous RNAse P/Z) led to a strong decrease in efficacy in PE3 (iPE-N$_{K918N-Sso7d}$; all 3′-modified pegRNA versus 3′-unmodified pegRNA; $P = 0.0001$; one-way ANOVA with Bonferroni MCT; Extended Data Fig. 3a,c), probably due to RNA destabilization[18]. Similarly, removing the 3′-uridines by HDV ribozymes was almost as effective ($P = 0.0412$) as tRNAs. However, prolonging the pegRNA half-life by adding the 3′-pseudoknot tevopreQ1 (ref. 16), which we had also already used in previous experiments, was more effective than removing the destabilizing terminal U-stretch ($P < 0.0001$). The minimal pseudoknot (mpknot) from the same study[16] as tevopreQ1 was not effective ($P > 0.9999$; Extended Data Fig. 3c), while other pseudoknots (BWYV, PEMV) or hairpin structures were similarly effective as tevopreQ1 in Exo-PE (all versus tevopreQ1; $P = 0.1736$ and $P > 0.9999$ for BWVV and PEMV, respectively; one-way ANOVA with Bonferroni MCT; Extended Data Fig. 3d).

The best editing efficacy for iPE on eTLR was thus obtained with a pegRNA containing a PP7$_{v3}$ aptamer and tevopreQ1 (7$_{v3}$/Q1), resulting in a ~50-fold better efficacy than the original editor[1] for PE3 and ~30-fold better efficacy for Exo-PE (all versus all; $P < 0.0001$; one-way ANOVA with Bonferroni MCT; Extended Data Fig. 3e). Compared with the optimized PE enzyme, termed PEmax[2], iPE$_{K918N-Sso7d}$ had a 50% higher efficacy while maintaining the same editing precision in PE3 ($P < 0.0001$; Extended Data Fig. 3e). iPE$_{K918N}$ in Exo-PE was as effective as PEmax in PE3 ($P = 0.1620$), but with higher editing precision ($P = 0.0019$).

## Benchmarking of Exo-PE against PE2

After extensive optimization of the prime editor complex using the eTLR reporter, we derived an optimal configuration (Fig. 4a) and validated its performance with the Exo-PE strategy against standard PE strategies at established reference sites in the HEK293T genome[16] (Figs. 4–6).

We hypothesized that Exo-PE would be particularly advantageous for challenging larger insertions and therefore selected a 30-base pair (bp) stretch encoding a Flag-tag and a 54-bp flippase recognition target (FRT) site as the insertions. Editing efficacy and precision were determined from amplicon sequencing data, with efficacy given as the total percentage of correct edits and precision calculated as the percentage of correct edits versus all edits.

Please note that the HEK3 locus in HEK293T contains a monoallelic SNP 9 bp downstream of the HEK3 editing site in the flap incorporation area. For the VEGFA locus, we had constructed a similar case in which complete PE introduced an additional substitution 5 bp downstream of the insertion site (Fig. 4b). Both outcomes (insertion with and without the additional substitution) were registered separately as correct edits (Figs. 4c,d and 5, blue and light blue bars) and subsequently aggregated.

We first compared Exo-PE with PE2 (Fig. 4c), which is the most direct benchmark as it also requires only a single pegRNA and no additional nicking sgRNA (required for PE3) or second pegRNA (used in paired PE). For editing efficacy, a three-way ANOVA reported significant effects for PE strategy ($F = 1,538.637$, $P < 0.0001$), locus ($F = 265.677$, $P < 0.0001$), and iPE-N/C ($F = 14.545$, $P = 0.0005$), as well as an interaction for locus and PE strategy ($F = 193.464$, $P < 0.0001$). Since the factor iPE-N/C and its interactions explained less than 2% of the variance, we averaged over this factor to conduct Bonferroni MCT for Exo-PE versus PE2 at each locus, showing that Exo-PE significantly increased editing efficacy for the 30-bp Flag insertion over PE2 in all tested loci except RNF2 (Fig. 4c and see full statistical results in Supplementary Table 1), corresponding to a mean increase of 14.4 percentage points averaged across loci and iPE-N/C.

For editing precision, an analogous analysis revealed that Exo-PE over PE2 achieved an at least equal precision for all loci except RUNX1 (Bonferroni MCT from a three-way ANOVA with effects for PE strategy ($F = 32.712$, $P < 0.0001$), locus ($F = 5.182$, $P = 0.0087$) and iPE-N/C ($F = 6.665$, $P = 0.0164$; Fig. 4c).

We also tested Exo-PE against PE2 for the insertion of a 54-bp sequence containing an FRT site (Fig. 4d) and found an average increase (over loci and iPE-N/C) in editing efficacy of 9.96 percentage points (three-way ANOVA with effects for PE strategy ($F = 511.47$, $P < 0.0001$), locus ($F = 288.548$, $P < 0.0001$), iPE-N/C ($F = 12.5$, $P = 0.0017$) and a PE strategy/locus interaction ($F = 156.780$, $P < 0.0001$)). The editing efficacies for FRT and Flag insertion were correlated for Exo-PE and lower for FRT (Fig. 4e).

We then repeated the experiment with identical conditions in HeLa cells and again found an improved, or on par, editing efficacy of Exo-PE compared with PE2 for Flag and FRT insertions for all loci (three-way ANOVA with Bonferroni MCT; Extended Data Fig. 4a,b). The editing efficacy from HeLa cells was correlated to that in HEK293T cells, but was generally lower, causing higher variance for FRT insertion data (Extended Data Fig. 4c).

We also examined off-target activity of Exo-PE compared with PE2 for four known Cas9 off-targets of the HEK3 locus[19]. Although no off-target activity was detected for most loci, the most commonly affected HEK3 off-target site ('OT1'), with a perfectly matching PBS of the pegRNA, displayed an increased number of editing events when Exo-PE was used (Supplementary Table 2).

## Comparison of Exo-PE with PE4

We also assessed the effect of MMR inhibition via the coexpression of dominant-negative MLH1 (MLH1dn), which was recently shown to improve PE2 efficacy and precision for small edits without secondary nicking sgRNAs.

Similarly, Exo-PE displayed improved editing efficacy compared with PE4 (mean difference across loci and iPE-N/C was 12.55 percentage points, from a three-way ANOVA identifying main effects for PE strategy ($F = 438.402$, $P < 0.0001$), locus ($F = 499.14$, $P < 0.0001$), iPE-N/C ($F = 17.035$, $P < 0.0001$) and an interaction of PE strategy/locus ($F = 49.917$, $P < 0.0001$); Fig. 5).

Exo-PE4 (a combination of Exo-PE and PE4) did not seem to provide an additional increase in efficacy over Exo-PE. The effect of iPE-C/N, while significant for efficacy, was small and accounted for only 0.42% of the total variation.

---

**Fig. 4 | Editing efficacy and precision of Exo-PE benchmarked against PE2 for two different insertions in multiple loci. a**, Schematic of the final prime editor system for the Exo-PE strategy, composed of an iPE combined with an engineered pegRNA bearing a PP7 aptamer to recruit a T5 phage 5′–3′-DNA-exonuclease C-terminally fused to PCP to facilitate the integration of the de novo synthesized 3′ flap carrying the desired edit. **b**, HEK293T contains a heterozygous SNP 9 bp downstream of the insertion site in the HEK3 locus. For VEGFA, an additional substitution of 5 bp downstream of the insertion site was also included in the pegRNA design. Both SNP and substitution increase the effective insertion size while narrowing the extent of the homology region of the pegRNA. **c,d**, Editing efficacy (correct edits) and precision were determined for two insertion types, Flag (**c**) and FRT (**d**), for iPE-N (N) and iPE-C (C) with either the PE2 or the Exo-PE editing strategy in HEK293T cells by amplicon sequencing. Precision was calculated as the proportion of correctly edited reads within all altered (nonwild-type) reads. Correct editing outcomes merely lacking the additional substitution as shown in **b** were registered separately as correct edits (light blue). Selected results of Bonferroni MCT averaged over iPE-N/C after three-way ANOVA (locus, PE strategy, iPE-N/C) for editing efficacy (results reported in the text) are indicated by asterisks; NS $P > 0.05$, *$P < 0.05$, ****$P < 0.0001$. Bars, mean ± s.d. ($n = 3$ biological replicates). **e**, Replotting of the editing efficacies shown in **c** and **d** for FRT against Flag insertions for the loci DNMT1, RNF2 and VEGFA. A linear regression is shown for Exo-PE (Pearson $r = 0.9244$, $P = 0.0084$) with 0.95 confidence bands shown in dashed lines. See Supplementary Table 1 for complete statistical results.

Again, Exo-PE exhibited a similar editing precision to that of PE2/PE4, with overall levels at ~90% for all loci except *RUNX1* (Fig. 5). Detailed analysis of *RUNX1* editing outcomes revealed a propensity for a specific thymidine insertion in all conditions, as well as a (partial) duplication of the pegRNA homology region at the insertion site only in Exo-PE conditions (Extended Data Fig. 5). The duplication effect also occurred

occasionally at other loci, but for both Exo-PE and PE2 and at very low frequencies (<0.2%). In this context, we also evaluated cell viability and proliferation and found a slight reduction of the bioluminescent signal in a commercial assay for Exo-PE and PE4 at times, indicating that both editing strategies can lead to a slight reduction of the proliferation rate during transient expression (Extended Data Fig. 6).

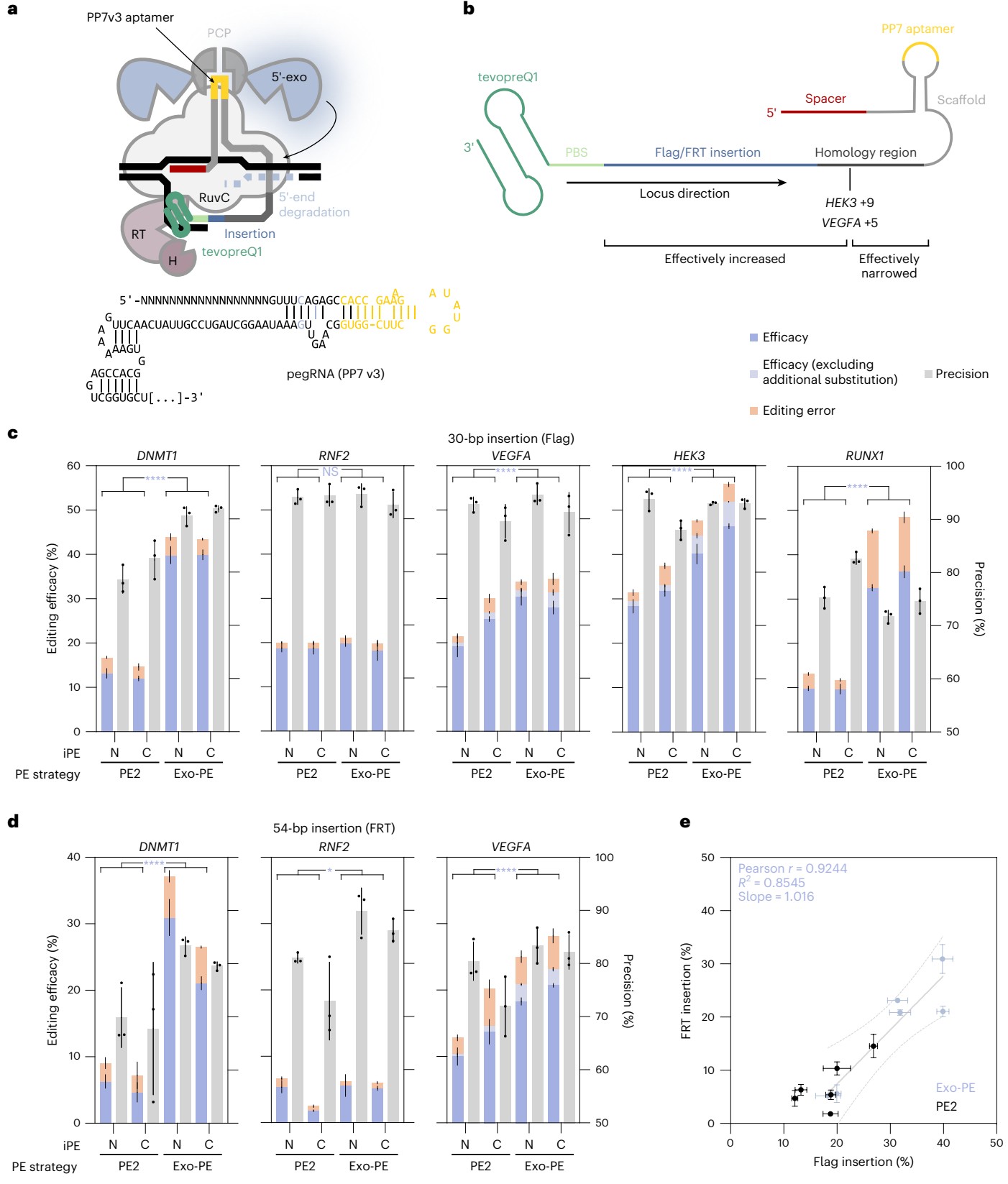

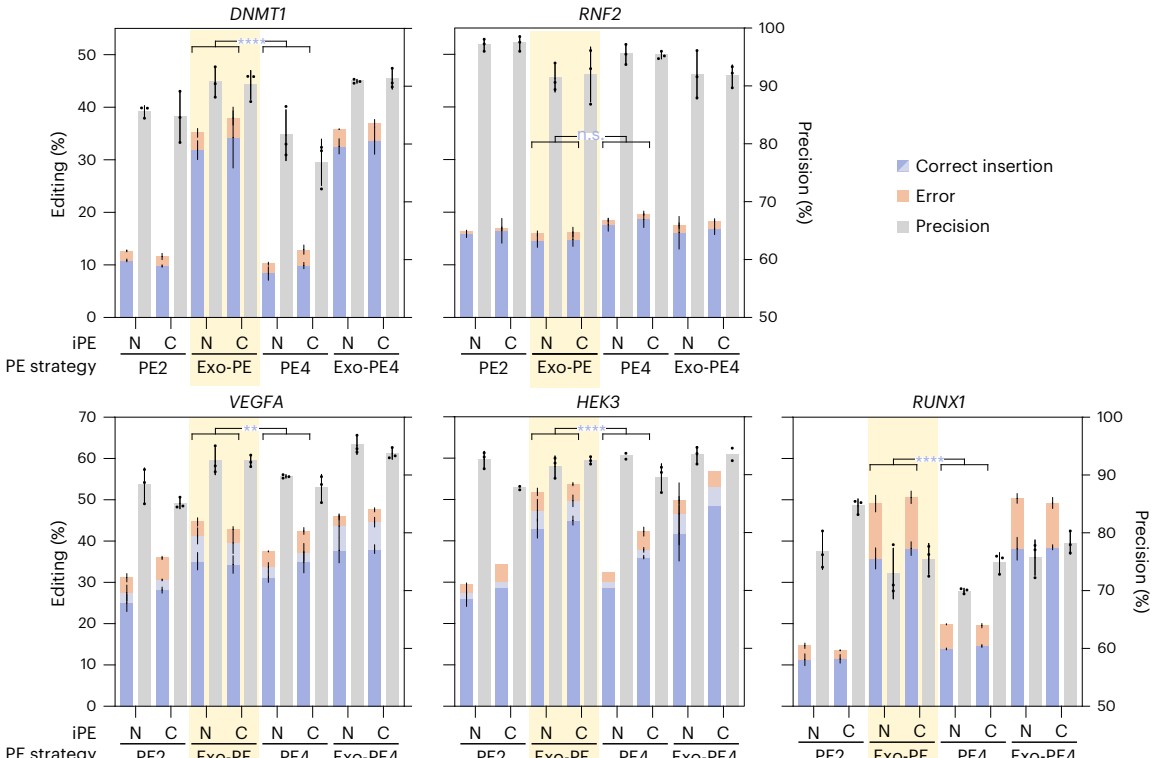

**Fig. 5 | Benchmarking of the Exo-PE strategy for 30-bp insertions against PE4.** Editing efficacy (correct edits) and precision (correct edits divided by all altered (nonwild-type) reads) were determined from amplicon sequencing. PE4 refers to PE2 + MLH1dn to inhibit MMR, Exo-PE4 refers to Exo-PE + MLH1dn. Each strategy was evaluated using both iPE-N and iPE-C. HEK293T contains a heterozygous SNP 9 bp downstream of the insertion site in the *HEK3* locus; for *VEGFA*, an additional substitution 5 bp downstream of the insertion site was included in the pegRNA design (Fig. 4b). Selected results of Bonferroni MCT averaged over iPE-N/C (after three-way ANOVA with results reported in the text) are indicated by asterisks; NS $P > 0.05$, $**P < 0.01$, $***P < 0.001$, $****P < 0.0001$. Bars, mean ± s.d. ($n = 3$ biological replicates, except for *HEK3* C-PE2, N-PE4 and C-Exo-PE4 where $n = 2$).

In addition, we examined the performance of Exo-PE at genomic sites associated with diseases such as prion disease[20] and *CDKL5* deficiency disorder[21] for both the 30-bp insertion (Extended Data Fig. 7a) and therapeutically relevant substitutions (Extended Data Fig. 7b). Compared to PE2, Exo-PE again increased efficacy for insertions (three-way ANOVA with Bonferroni MCT) but not for substitutions, consistent with previous observations for the respective edit types at other loci.

### Exo-PE for protein tagging via fluorescence complementation
We next sought to use Exo-PE for a practical application in biomedical research, inspired by a large-scale in-situ protein tagging study[22]. We designed pegRNAs to extend the endogenous locus with sequences coding for the 11 amino acids of a split fluorescent protein, namely monomeric mNeonGreen2 (mNG2)[23]. A HEK293T cell line containing the stably integrated split-mNG2(1–10) was subjected to PE for inserting the missing mNG2(11) peptide at the N or C terminus of functionally expressed genes. Fluorescence complementation consequently indicates the successful insertion of the peptide without compromising the expression of the tagged gene (Extended Data Fig. 8a). We observed a main effect of Exo-PE across the first panel of loci (two-way ANOVA, Exo-PE/PE2, locus; $F (1, 28) = 40.35$, $P < 0.0001$; Extended Data Fig. 8b). We also targeted two additional loci (*GAPDH* and *ENO1*-N, Extended Data Fig. 8c) and again found that Exo-PE had a greater editing efficacy than PE2 ($P = 0.0349$ for both loci, Bonferroni MCT), while PE3 also showed substantial efficacy.

### Comparison of Exo-PE with PE3 and PE5
We then extended the comparison with PE3 and PE5, both of which require a secondary nick, for the 30-bp Flag insertion in an additional set of loci. We again selected the reference loci tested in ref. 2, so that we could use the identical nicking sites (Fig. 6).

As before, Exo-PE demonstrated superior editing efficacy for the insertions across loci, with precision also being similar or better in all cases (Bonferroni MCT based on a three-way ANOVA with significant main effects for PE strategy ($F = 1135.566$, $P < 0.0001$), locus ($F = 262.932$, $P < 0.0001$) and the interaction of both factors ($F = 39.102$, $P < 0.0001$; Fig. 6). However, both PE3 and PE5 showed a substantial reduction in mean editing precision across loci and iPE-N/iPE-C by 66.4 and 63.2 percentage points, respectively, compared with Exo-PE. Closer investigation revealed that both strategies suffered from a large proportion of indels occurring between the two nicking sites, even when the insertion was incorporated successfully (Extended Data Fig. 9).

When we subsequently evaluated Exo-PE performance on single-base substitutions at the same endogenous sites, no improvements in efficacy were found over PE2, while PE4 resulted in a mean increase across loci and iPE-N/C of 5.485 percentage points (based on a three-way ANOVA identifying main effects for PE strategy ($F = 34.091$, $P < 0.0001$), locus ($F = 499.217$, $P < 0.0001$), iPE-N/C ($F = 11.273$, $P = 0.001$), and an interaction of PE strategy/locus ($F = 31.973$, $P < 0.0001$); Extended Data Figs. 7b and 10a,b). The average precision of PE3 and PE5 across loci and iPE-N/C was much higher (53.4% and 60.5%) on substitution-type edits (Extended Data Fig. 10a) than for mid-size insertions (5.3% and 8.7%), respectively (Fig. 6).

In summary, we found that Exo-PE to exhibited a significantly higher, or at least equal editing efficacy against PE2, PE4 (PE2 with MMR inhibition), and dual-nick strategy without (PE3) and with MMR inhibition (PE5) in each of the independent benchmarking experiments (Figs. 4–6 and see full statistical results in Supplementary Table 1).

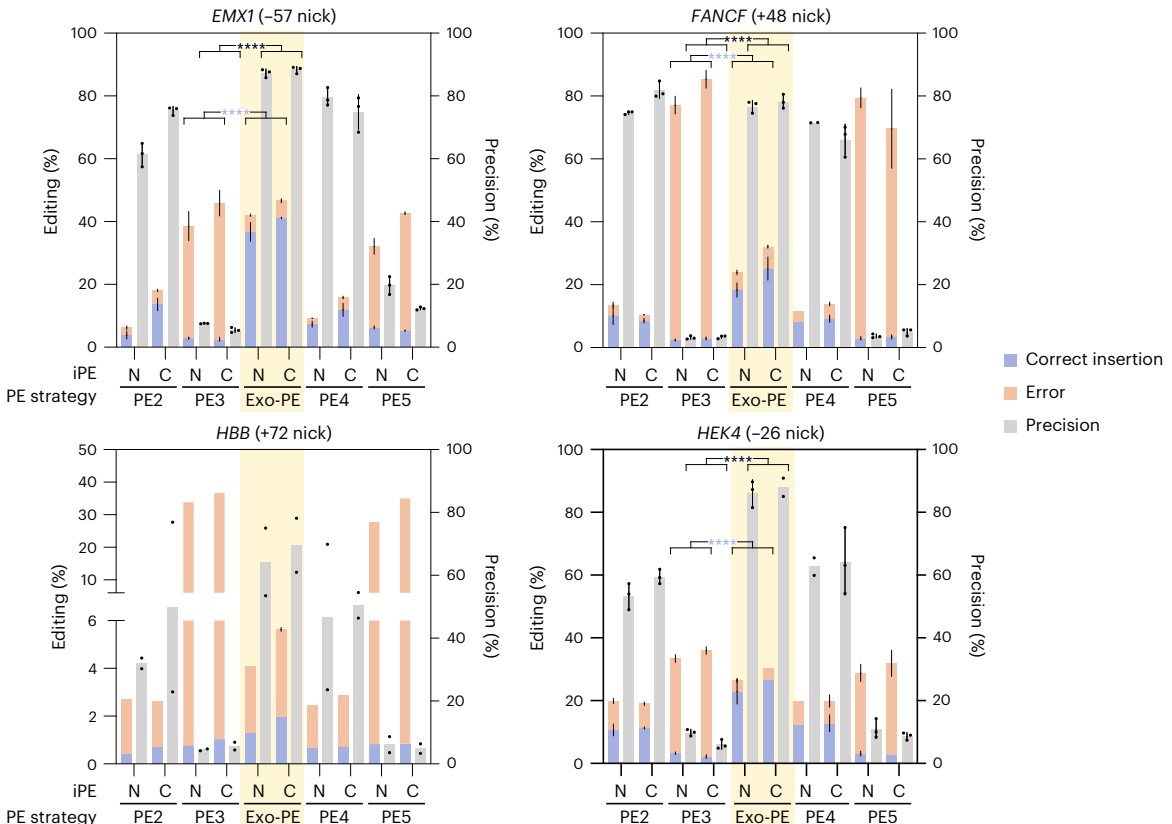

**Fig. 6 | Comparison of the Exo-PE strategy for 30-bp insertions with dual-nick strategies.** Exo-PE was benchmarked against the dual-nick strategies PE3 and PE5 (PE3 + MLH1dn) for 30-bp insertions on an additional set of endogenous loci. Nick positions of ngRNAs for PE3/PE5 strategies are indicated in parentheses. Selected results of Bonferroni MCT averaged over iPE-N/C (from three-way ANOVA with results reported in the text) are indicated by asterisks; ****$P < 0.0001$. Bars, mean ± s.d. ($n = 3$, biological replicates, except for *FANCF* (N-PE4), *HEK4* (C-Exo-PE), *HEK4* (N-PE4) and *HBB* (all conditions) where $n = 2$).

Over all 11 loci in which Exo-PE was compared directly against PE2 for Flag insertions in HEK293T cells, Exo-PE showed a significant increase in efficacy in 9 out of 11 loci with an average difference across loci and iPE-N/C of 14.2 percentage points (Supplementary Fig. 1) based on a three-way ANOVA with factors PE strategy ($F = 1,079.564$, $P < 0.0001$), locus ($F = 167.58$, $P < 0.0001$) and iPE-N/C ($F = 20.203$, $P < 0.0001$), with the latter factor accounting for only 0.59% of the total variation. We also observed an interaction of Locus/Strategy ($F = 46.9$, $P < 0.0001$, 13.69% of total variation) driven by the *RNF2* locus, for which Exo-PE did not show an increased efficacy (full statistical results in Supplementary Table 1). Except for the *RUNX1* locus (Extended Data Fig. 5), Exo-PE displayed similar or slightly increased precision at all nine loci where it significantly increased efficacy.

## Discussion

We demonstrated that we could iteratively achieve substantial improvements in the PE machinery using an improved reporter system (eTLR) which, in contrast to previous versions[4], indicates all mutagenic indel events. eTLR allowed us to rapidly identify an improved PE with optimized codons and linkers, including a potent superNLS (iPE-C/N), which, combined with engineered pegRNAs protected by (t)evopreQ1 (ref. 16), offered the best alternative to PEmax.

We then tackled the flap competition as a central bottleneck for PE efficacy, especially for larger insertions, by recruiting a 5′–3′ exonuclease to iPE via aptamer-mediated recruitment to create space for the invasion of the reverse-transcribed 3′ flap. Exo-PE was then validated on several endogenous loci and showed a substantial improvement in editing efficacy for 30–57-bp insertions (Flag, FRT sites, split C-mNeonGreen) in several cell lines when benchmarked against the standard techniques PE2, PE3, PE4 and PE5. Exo-PE improved editing

efficacy without compromising editing precision compared with PE2 and PE4. Although iPE-C displayed a slightly better performance than iPE-N for many loci, it may be beneficial to test both variants for a specific target locus.

The preserved editing precision of Exo-PE is particularly valuable when contrasted with the dual-nicking strategy PE3, which displayed substantially increased indel rates for the insertion-type edits. An increased indel frequency was reported previously for larger insertions[24], while smaller substitutions and insertions generated with PE3 resulted in only a minor increase in indel frequency in most cases[1]. In general, single-nick techniques such as PE2/PE4 and base editing hold a key advantage over DSB-induced HDR-dependent CRISPR editing techniques in that staggered DSBs can be avoided. While Exo-PE inherits this safety feature from PE2 and thus maintains high editing precision, it is conceivable that two nicking sites used in PE3 (and paired PE strategies) could generate a staggered DSB break leading to increased editing errors. Furthermore, nicking at the secondary site in PE3 is likely to occur even after successful editing as long as the PE machinery is expressed, which may (or needs to) be the case for a prolonged period of time, for example, after virus-mediated gene delivery. Nonetheless, if increased editing efficacy is the primary objective in a given experimental setting where indels can be tolerated, PE3 could be highly effective also for longer insertions.

It is conceivable that the flap competition contributes to an apparent specificity for on-target versus off-target effects. Since this competition is mitigated by Exo-PE, it might explain the increased editing versus PE2 on the main OT1 for the *HEK3* locus, for which the PBS of the reference pegRNA[24] provides a perfect match. It may thus be important when using Exo-PE to ensure that the PBS does not perfectly match

other sites to maintain the specificity benefit of each hybridization event before flap binding[24].

MMR inhibition via PE4/PE5 did not provide any benefit when inserting large sequences such as a Flag-tag, most likely as a consequence of ≥14-bp insertions not being recognized by the MMR pathway, a PE-inhibiting process[22,25]. We also observed that either exonuclease or MLHdn expression could transiently lead to a mild reduction in proliferation. While this did not affect the quality of our experiments, one could reduce the amount of exonuclease in situations where viability is a major concern, or even invest in a split-exonuclease recruitment approach.

In contrast to the improvements in editing efficacy for insertions, we did not observe a similar benefit of Exo-PE for short substitutions, although it occasionally outperformed PE4 at certain loci. In our experiments, we also did not find PE4 to be substantially more effective than PE2, as reported in the initial publication[2]. This may be due to differences in the pegRNA/nicking sgRNA scaffold compared to the literature[26,27] and the lower amount of MLH1dn plasmid (~33% less) we used to keep the concentration of other genetic components constant.

With PE-mediated insertions gaining increasing interest[25], Exo-PE thus provides a complementary strategy to PE4, with Exo-PE showing superior performance for larger edits and PE4 for smaller edits. Exo-PE may therefore be particularly useful for applications in basic research or biological engineering, where the addition of epitopes, affinity handles or degradation motifs may be of interest. During the revision of this manuscript, a study in plants showed an improvement in the efficacy of PE2 by adding a T5 exonuclease to the PE editor, but only when fused to the N terminus of nCas9 and not when they attempted to recruit it via a tandem insertion of MS2/F6 aptamers[28].

For larger insertions, we showed that the editing efficacy of Exo-PE can approach or even exceed that of PE3, while maintaining the superior precision of PE2. The Exo-PE approach may thus be attractive in cases where PE3 or paired PE are not feasible, for instance, if a high editing precision is desired, in the absence of a second suitable proximal PAM or in other organisms in which PE3 did not lead to enhanced editing efficacy[29]. The lack of a second nick also considerably reduces the combinatorial complexities for pegRNA optimization.

In the future, it will be interesting to directly benchmark Exo-PE against the many different variants of paired PE strategies, which have been shown to enable substantially longer insertions than Exo-PE, although these methods necessitate a careful optimization of the paired pegRNAs for specific loci[3]. Exo-PE may also catalyze those paired pegRNA approaches by degrading the endogenous 5′-intermediates, but active flap degradation could also abolish the target site for the paired pegRNA. As shown by the increased efficacy for inserting the FRT site, Exo-PE may also be beneficial for combining PE with recombinase-dependent insertions of longer DNA stretches[30].

Given the complex orchestration of multiple processes required for PE, Exo-PE adds an effective functionality to the PE machinery, which can further advanc the versatility of this powerful gene-editing technology.

## Online content

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

## Methods

### Molecular cloning

**Genetic constructs.** All pegRNAs used in this study (unless indicated) contained a modified pegRNA scaffold containing modifications from ref. [26] and ref. [27], to improve expression yield and a 3'-tRNA (M1–7 tRNA[31]) or an evopreQ1 (ref. [2]) motif upstream of the 6× T-DNA-dependent RNA polymerase III termination signal. All pegRNA sequences used in this study are provided in Supplementary Table 3. All amino acid sequences of coding components of iPE can also be found in Supplementary Table 3, as well as corresponding codon-optimized nucleotide sequences including the RT domain in which we removed a potential splice site. The expression constructs for iPE-N (214734), iPE-C (214735), PCP-Exo (214736), and the AAVS1-eTLR donor construct (214737) have been provided to Addgene.

**Polymerase chain reaction.** Single-stranded primer deoxyribonucleotides (Integrated DNA Technologies) were resolubilized (100 µM) in nuclease-free water. A PCR reaction with plasmid and genomic DNA templates was performed with Platinum SuperFi II PCR Master Mix (Thermo Fisher Scientific) according to the manufacturer's protocol. PCR reactions were purified by DNA agarose gel electrophoresis and subsequent DNA extraction using Monarch DNA Gel Extraction Kit (New England Biolabs (NEB)).

**DNA digestion with restriction enzymes.** DNA products were digested with restriction enzymes according to the manufacturer's protocol (NEB) in 40 µl with 2–3 µg of plasmid DNA. Afterward, fragments were gel-purified by DNA agarose gel electrophoresis, and DNA was extracted using Monarch DNA Gel Extraction Kit (NEB).

**Molecular cloning.** Concentration of purified DNA was measured using a spectrophotometer (NanoDrop 1000, Thermo Fisher Scientific). For ligations with T4 DNA ligase (Quick Ligation Kit, NEB), 50–100 ng backbone DNA (DNA fragment containing the DNA replication origin) was used in 20-µl volume, with molar 1:1–3 backbone:insert ratios, at room temperature for 5–10 min. Gibson assemblies were performed with ~75 ng backbone DNA and a molar 1:1–5 backbone:insert ratios in a 15-µl reaction volume, using NEBuilder HiFi DNA Assembly Master Mix (2×) (NEB) for 15–60 min at 50 °C.

**DNA agarose gel electrophoresis.** Agarose (Agarose Standard, Carl Roth) gels (1% (m/m)) were prepared in 1× Tris-acetate-EDTA buffer and 1:10,000 SYBR Safe stain (Thermo Fisher Scientific). Gel electrophoresis was carried out for 20–40 min at 120 V. For size determination, the 1-kb Plus DNA Ladder (NEB) was used. DNA samples were mixed before loading with Gel Loading Dye (Purple, 6×) (NEB).

**Bacterial strains for molecular cloning.** Chemically competent stable *Escherichia coli* cells (NEB) were used to propagate circular plasmid DNA. Carbenicillin (Carl Roth) was used for selection during plasmid amplification at a final concentration of 100 µg ml⁻¹. All bacterial cultures were prepared in Lysogeny Broth (LB) medium and on LB agar plates with carbenicillin (Carl Roth).

**Bacterial transformations.** Chemical transformation of *E. coli* was performed by mixing 1–5 µl of ligation or Gibson reaction with 50 µl thawed, chemically competent stable *E. coli* (NEB) and incubating on ice for 5–30 min. Cells were subsequently heat-shocked at 42 °C for 30 s, followed by a 5 min incubation on ice, and, finally, 950 µl SOC-medium (NEB) was added to the cell suspension. After outgrowth for 10–30 min at 37 °C, cells were plated on agar plates containing appropriate antibiotics, followed by overnight incubation at 37 °C or 48 h incubation at room temperature.

**Plasmid purification and Sanger sequencing.** *E. coli* colonies with correct potential constructs were inoculated from agar plates in 2 ml LB medium at 37 °C with the respective antibiotics and incubated for at least 6 h or overnight. Plasmid DNA was extracted with QIAprep Plasmid MiniSpin (Qiagen) according to the manufacturer's protocol and sent for Sanger sequencing (GENEWIZ, Azenta Life Sciences). Sanger-sequencing-validated clones were inoculated into 100 ml LB medium containing the respective antibiotic selection agent and incubated overnight at 37 °C. Plasmid DNA was extracted using the Plasmid Maxi Kit (Qiagen).

**Genomic DNA isolation.** At 72 h after transfection in 96-well format, genomic DNA was isolated with the Quick-DNA 96 Kit (Zymo Research) according to the manufacturer's protocol, with an elution volume of 30 µl.

**Amplicon PCR and purification.** PCR was performed as described above using ~50 ng of gDNA and appropriate primers for each target. Amplicon lengths were designed to approach 250 bp for sequencing. PCR purification was performed using the DNA Clean and Concentrator-5 Kit (Zymo Research) according to the manufacturer's protocol, with an elution volume of 30 µl. All primer sequences are provided in Supplementary Table 3.

**Amplicon sequencing and analysis.** Following an initial PCR on genomic DNA, a second outer PCR using barcoded primers was performed. PCR products of each experiment were purified as described above, normalized and combined into a single tube. The mixture was gel-purified, normalized to 20 ng µl⁻¹, and submitted for Amp-EZ sequencing (Azenta). The resulting fastq files containing paired reads were analyzed with Geneious via barcode separation and CRISPR editing analysis within the entire range covered by reads or at least the full sequence area between the genomic primer-binding sites.

### Mammalian cell culture

**Cell lines and maintenance.** HEK293T (Sigma-Aldrich, catalog no. ECACC 12022001), HEK293 eTLR (originating from HEK293, catalog no. ECACC 85120602) and HeLa cells (catalog no. ECACC 93021013) were cultivated at 37 °C and 5% CO$_2$ in an H$_2$O-saturated atmosphere in Gibco Advanced DMEM (Thermo Fisher Scientific) with 10% FBS (Gibco, Thermo Fisher Scientific), GlutaMAX (Gibco, Thermo Fisher Scientific) and 100 µg ml⁻¹ penicillin-streptomycin (Gibco, Thermo Fisher Scientific). Cells were passaged twice a week at 90% confluency by aspirating the medium, washing with DPBS (Gibco, Thermo Fisher Scientific), and detaching the cells with 2–3 ml of an Accutase solution (Gibco, Thermo Fisher Scientific) for 5–10 min at room temperature until a visible detachment of the cells was observed. Accutase was subsequently inactivated in 7.5 ml FBS-containing medium. Cells were then transferred into a new flask at an appropriate density for maintenance or were counted and plated in multiwell plates for subsequent plasmid transfection.

**Generation of eTLR reporter line.** The HEK293 eTLR cell line, which reports all three frames after an NHEJ/MMEJ-mediated indel event, was created by cloning the reporter coding sequencer between a CAG promoter and a bovine growth hormone (bGH) polyadenylation (pA) signal. This CAG_eTLR_bGH-pA construct was again cloned into a vector containing homology arms for the first intron of the PPP1R12C gene (alias AAVS1) with 0.8 kbp for each homology arm. In addition, for selection, a puromycin N-acetyltransferase (PuroR) gene trap was created by inserting a splice acceptor, the coding sequence for puromycin N-acetyltransferase and a bGH-pA between the CAG promoter and the 5'-homology arm. This donor construct was provided to Addgene. To create a HEK293 cell line containing the eTLR reporter, HEK293 cells were cotransfected with this donor, a Cas9-NLS expression plasmid

and a gRNA plasmid (spacer sequence GGGGCCACTAGGGACAGGAT) 24 h postseeding in a six-well plate (600,000 cells per 3 ml per well in a six-well plate) following the manufacturer's protocol (X-tremeGENE HP, Roche). At 48 h after transfection, cells were selected for 2 weeks with 0.5 µg ml⁻¹ puromycin dihydrochloride (Gibco, Thermo Fisher Scientific) in the presence of 0.5 µM AZD7648 (MedChemExpress, catalog no. HY-111783) (a DNA-PKcs inhibitor[32]) and a CAG-promoter-driven i53 (a 53BP1 inhibitor) to inhibit NHEJ and shift the DNA repair towards HDR[33,34]. The surviving polyclonal population carrying the eTLR system stably in the AAVS1 locus was monoclonalized using limiting dilution in 96-well plates. Clones were tested for the number of eTLR copies by transfecting the clones in a 48-well with a CRISPR–Cas9 plasmid against the eTLR system and a donor to repair the C-terminally truncated mScarlet-I. Clones that showed mutually exclusive fluorescence (either red or green, but not green and red) contained one copy of the eTLR reporter. Clones that also showed green and red fluorescence in parallel had two copies of the eTLR system. We chose a clone containing two copies of the eTLR system to mimic the diploid nature of most autosomal genes.

**Plasmid transfection into mammalian cells.** Cells were transfected with X-tremeGENE HP (Roche) or jetOPTIMUS (Polyplus Transfection) according to the manufacturer's protocol. Total DNA amounts were kept constant in all transient experiments to yield reproducible complex formation and comparable results. In 96-well plate experiments, a total of 100 ng, 300 ng and 2.4 µg in total per well was used for 96-well, 48-well and 6-well plate transfections, respectively. The ratios of plasmid DNA were 1:1 for pegRNAs to prime editors and 2:1 for pegRNAs to auxiliary components like exonucleases, nicking gRNAs and MLHdn. Cells were plated 1 day before transfection (25,000 cells per well in 100 µl for 96-well plates; 75,000 cells per well in 500 µl for 48-well plates; 600,000 cells per well in 3 ml for 6-well plates). At 24 h after transfection, 100 µl fresh medium was added per well in a 96-well transfection and at 48 h after transfection 100 µl medium per well was removed and replaced with fresh medium in 96-well transfections.

**Small molecule manipulation of NHEJ.** For modulation of NHEJ in the HEK293 eTLR line with NU7441 (MedChemExpress) or KU0060648 (alias KU-57788, MedChemExpress), the compounds were added to the cells 24 h after transfection. Control cells received the same volume of DMSO. HEK293 eTLR cells were transfected in a 48-well plate with a plasmid harboring the indicated gene-editing constructs to analyze repair events. At 72 h after transfection, cells were detached with Accutase, pelleted (200 relative centrifugal force, 5 min), and resuspended in ice-cold 0.4% formalin for 10 min. Fixed cells were pelleted again (200 relative centrifugal force, 5 min) and resuspended in ice-cold 200 µl DPBS for FACS analysis.

**Viability assay.** The viability of transfected cells was assessed 72 h after transfection before gDNA isolation using the RealTime-Glo MT Cell Viability Assay (Promega) according to the manufacturer's instructions (endpoint method, sample volume 20 µl, incubation time 60 min). Luminescence was measured on the Centro LB 960 plate reader (Berthold Technologies) with 0.5-s acquisition time.

**FACS analysis.** FACS analysis was performed on the BD FACSaria II system (controlled with the BD FACSDiva Software (v.6.1.3, BD Biosciences)). In brief, the main population of cells was gated first according to their FSC-A and SSC-A. Second, single cells were gated using FSC-A and FSC-W. The final gate (red and green fluorescence) was used to determine the number of undergoing mutEJ or PE. The events in the red/green/red-green gate were normalized to the number of cells in the single-cell gate. See Supplementary Information for a depiction of the FACS gating strategy.

## Statistics and reproducibility

Statistics were calculated as specified in each figure using R (ref. 35) (v.4.3.1 with the emmeans package[36] to calculate multiple comparisons and marginal effects) and Prism (v.9, GraphPad). Mean and s.d. were calculated across biological replicates.

For experiments on eTLR, the precision was calculated as the ratio of PE/HDR events over mutEJ events. For next-generation sequencing experiments, precision was calculated as the proportion of correct editing events among all editing events (correct/(correct + error) × 100).

We did not use statistical analysis to determine sample size or to randomize the experiments, nor did we blind the investigators to the allocation of the experiments or their outcomes. Despite those limitations, we made an effort to reduce any biases introduced during the sample preparation by using the use of master mixes and multichannel pipettes.

No data were excluded from the analyses except for some next-generation sequencing experiments (Fig. 5 and 6 and Extended Data Figs. 4, 7 and 10) for which certain replicates had to be excluded from calculating the mean value due to obvious technical errors (no reads). For the HBB locus in Fig. 6, one replicate was lost during sample processing.

The comparison Exo-PE versus PE2 from Fig. 4c was reproduced independently several months later on the identical loci, as shown in Fig. 5. with highly correlated efficacies for both editing strategies (Pearson $r(8) = 0.8646$ and $0.915$, $P = 0.0012$ and $0.0002$, respectively; Supplementary Fig. 2). Tables of all statistical tests are provided in Supplementary Table 1. While Bonferroni-corrected MCT of the three-way ANOVA was conducted with averaging over the weak factor iPE-N/C, test results from nonaveraged data are also provided in Supplementary Table 1.

## Reporting summary

Further information on research design is available in the Nature Portfolio Reporting Summary linked to this article.

## Data availability

PDB-ID of shown *Sp*Cas9 structure (Fig. 2b) is https://doi.org/10.2210/pdb6VPC/pdb. The plasmids for iPE-N, iPE-C, PCP-Exo and the AAVS1-eTLR donor construct are available via Addgene. Source data are provided with this paper.

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

## Acknowledgements

We thank R. Le Gleut and R. Meixner from the Core Facility Statistical Consulting at Helmholtz Munich as well as S. Haug from TUM for statistical consulting. We acknowledge support from the Federal Ministry of Education and Research (BMBF) and the Free State of Bavaria under the Excellence Strategy of the Federal Government

and the Länder through the ONE MUNICH project Munich Multiscale Biofabrication (to J.G. and G.G.W.). G.G.W. acknowledges support from the European Research Council (ERC-CoG 865710), G.G.W. and D.-J.J.T. acknowledge support from the European Innovation Council (EIC Pathfinder 101115574).

## Author contributions

D.-J.J.T. conceived the study, designed all constructs, cogenerated all constructs, coconducted all cell and biochemical experiments, coanalyzed all data and generated all figures. J.G., S.V.W., J.C.H.W., N.A., E.M.H.B., T.H.S., A.S. and C.G. cogenerated all constructs, coconducted all cell and biochemical experiments and coanalyzed all data regarding PE. T.P. and M.Ž. coconducted cell and biochemical experiments to initially characterize the eTLR system. J.G. and S.V.W. conducted the editing experiments on endogenous loci and cogenerated the corresponding figures. D.-J.J.T., J.G., S.V.W., N.A. and G.G.W. designed the experiments. J.G, D.-J.J.T., S.V.W. and G.G.W. wrote the manuscript. D.-J.J.T. coordinated the experimental activities. G.G.W. supervised the research.

## Funding

## Competing interests

The authors declare no competing interests.

## Additional information

**Extended data** is available for this paper at https://doi.org/10.1038/s41592-023-02162-w.

**Correspondence and requests for materials** should be addressed to Gil Gregor Westmeyer.

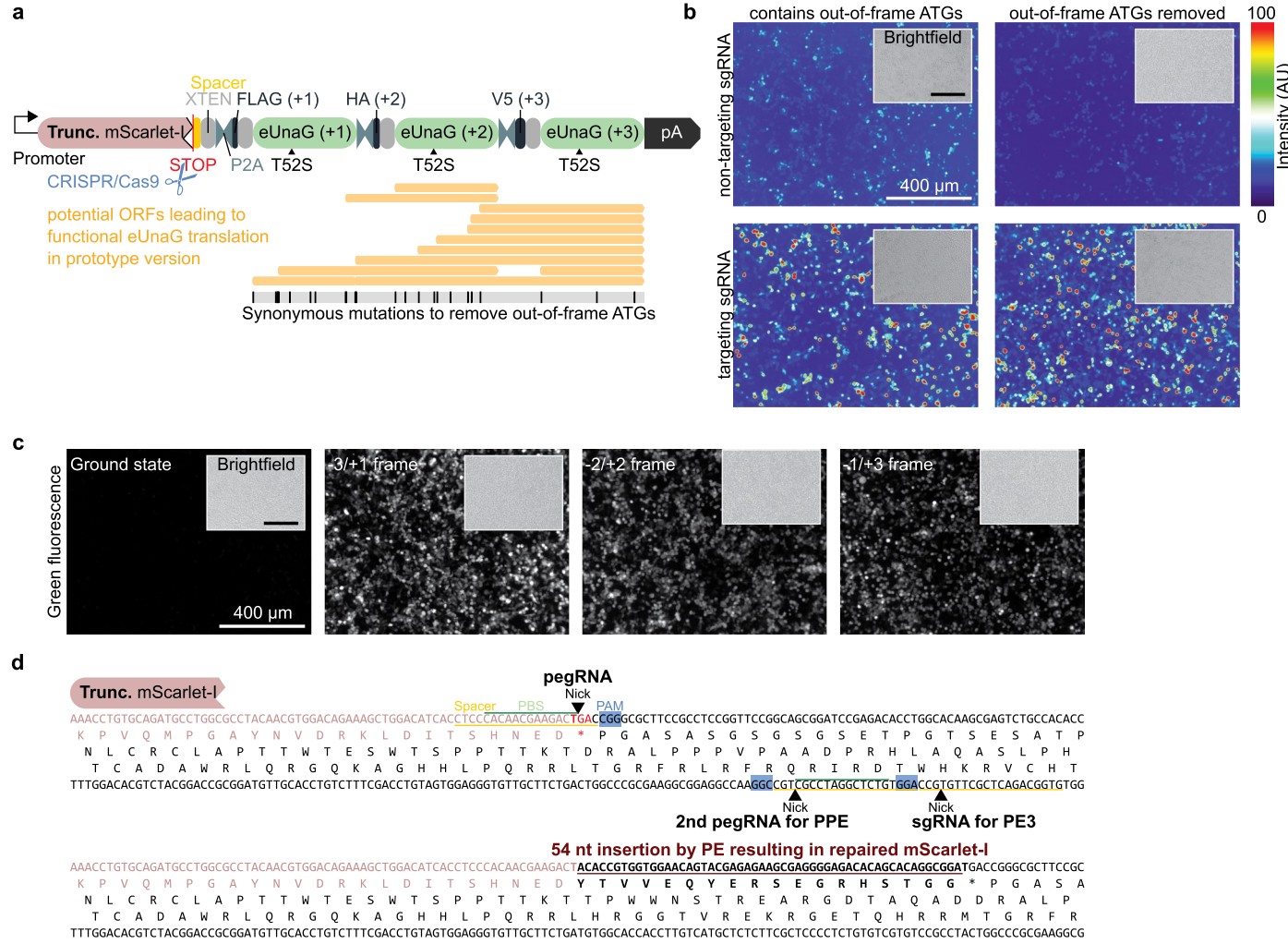

**Extended Data Fig. 1 | Optimization and validation of the enhanced Traffic Light Reporter (eTLR). a**, Depiction of possible ORFs inducing the functional translation of eUnaG in the prototype version of eTLR and incorporated mutations to remove these out-of-frame ATGs, resulting in the final version of eTLR. A prototype version of the eTLR system, which contains out-of-frame ATGs, was compared with the final version of the eTLR that does not have out-of-frame ATGs, which may otherwise result in the cryptic translation from internal 'noisy' transcription within the reporter's coding sequences. **b**, Representative false-colored epifluorescence images from HEK293T cells two days after transfection with the indicated reporter constructs with non-targeting controls (ntc) or targeting sgRNA against the cotransfected reporter. Results shown were reproduced for a total of two times independently. **c**, Representative epifluorescence images from HEK293T cells, transfected with the eTLR reporter shown in a, but carrying a − 1, −2, and −3 nt deletion at the potential nick/DSB site to mimic the three possible eUnaG-T52S reading frames after an error-prone repair. Results shown were reproduced for a total of two times independently. **d**, Depiction of the eTLR reporter editing site for PE analysis with annotated nicking/cuttings sites of the used pegRNAs/sgRNAs in this study.

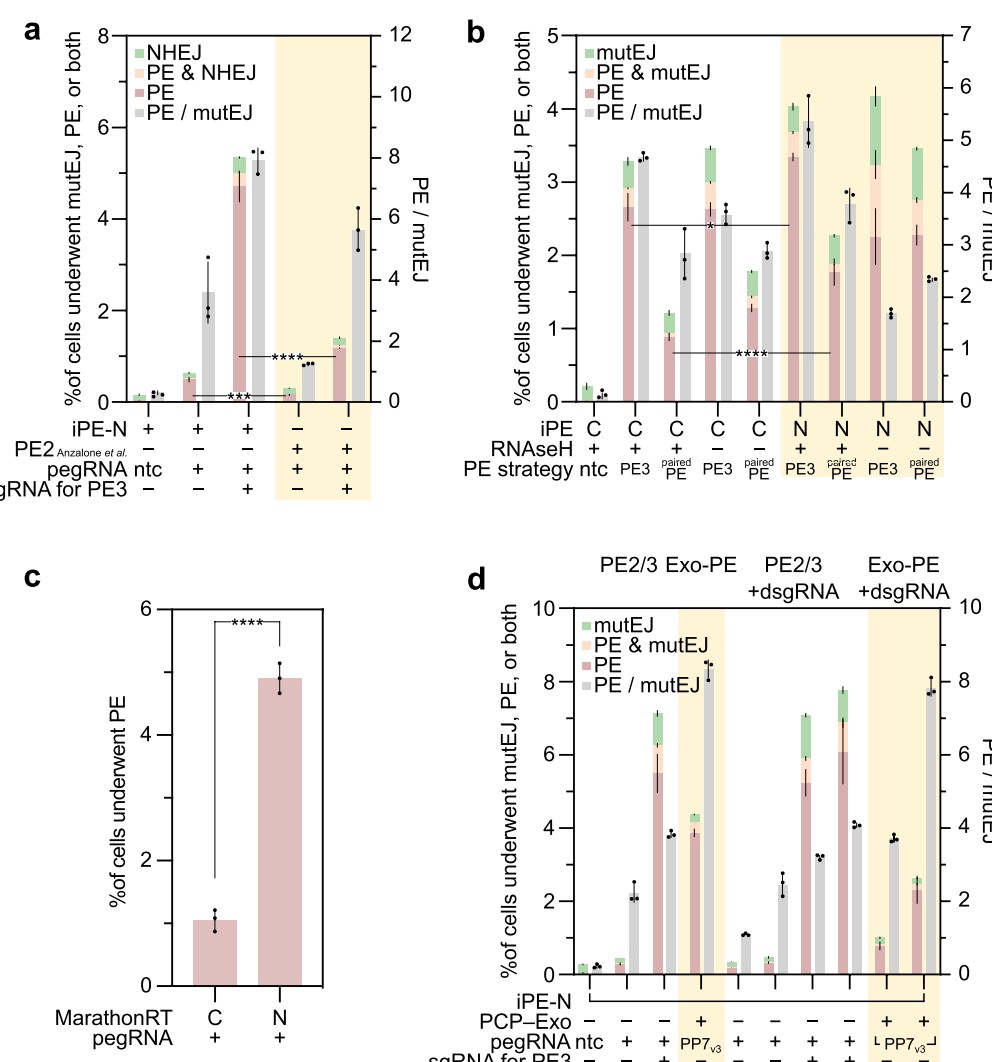

**Extended Data Fig. 2 | Screening for additional optimizations of the improved Prime Editor protein (iPE) architecture. a**, Comparison of the performance in 'PE2' or 'PE3' of the best RT variant identified in b (N-terminal RT with RNAse H) with the original PE2 construct (see a, top construct) from the original study[2]. Two-tailed unpaired *t*-tests were shown on the fractional PE events; ***P < 0.001, ****P < 0.0001. Data are shown as the mean ± SD (*n* = 3, biological replicates). **b**, Effects of iPE-C (C) or iPE-N (N) fusions of nucleotide-optimized versions of the hexamutant (x6) Moloney murine leukemia virus (MLV) reverse transcriptase with or without its RNAse H domain, analyzed by eTLR in HEK293. ntc: non-targeting control pegRNA; PE3: 'PE3' strategy. PE, mutEJ, and PE & mutEJ events (colored bars, left y-axis) and PE/mutEJ-ratio (gray bars, right y-axis). Selected results of Bonferroni MCT after one-way ANOVA were shown on

the PE events and are indicated by asterisks; *P < 0.05, ***P < 0.001, ****P < 0.0001. Bars represent the mean ± SD (*n* = 3, biological replicates). **c**, Comparison of an N-terminal or C-terminal fusion variant of the MarathonRT reverse transcriptase (group II intron maturase) to Cas9 nickase (nCas9) on prime editing efficacy. In this case, a prototype variant of the reporter shown in Fig. 1 without UnaG was co-transfected instead of genomically integratedused, in which PE recovers repairs the missing C-terminus (54 nt, 18 aa) of a reporter expressing a C-terminally truncated mScarlet-I, resulting in red fluorescence. Result of a two-tailed unpaired *t*-tests is shown for the PE events; ****P < 0.0001. Data represent the mean ± SD (*n* = 3, biological replicates). **d**, Effects of an additional proximal binding of a catalytically dead sgRNA (dsgRNAs) on 'PE2', 'PE3', and 'Exo-PE'. Bars represent the mean ± SD (*n* = 3, biological replicates).

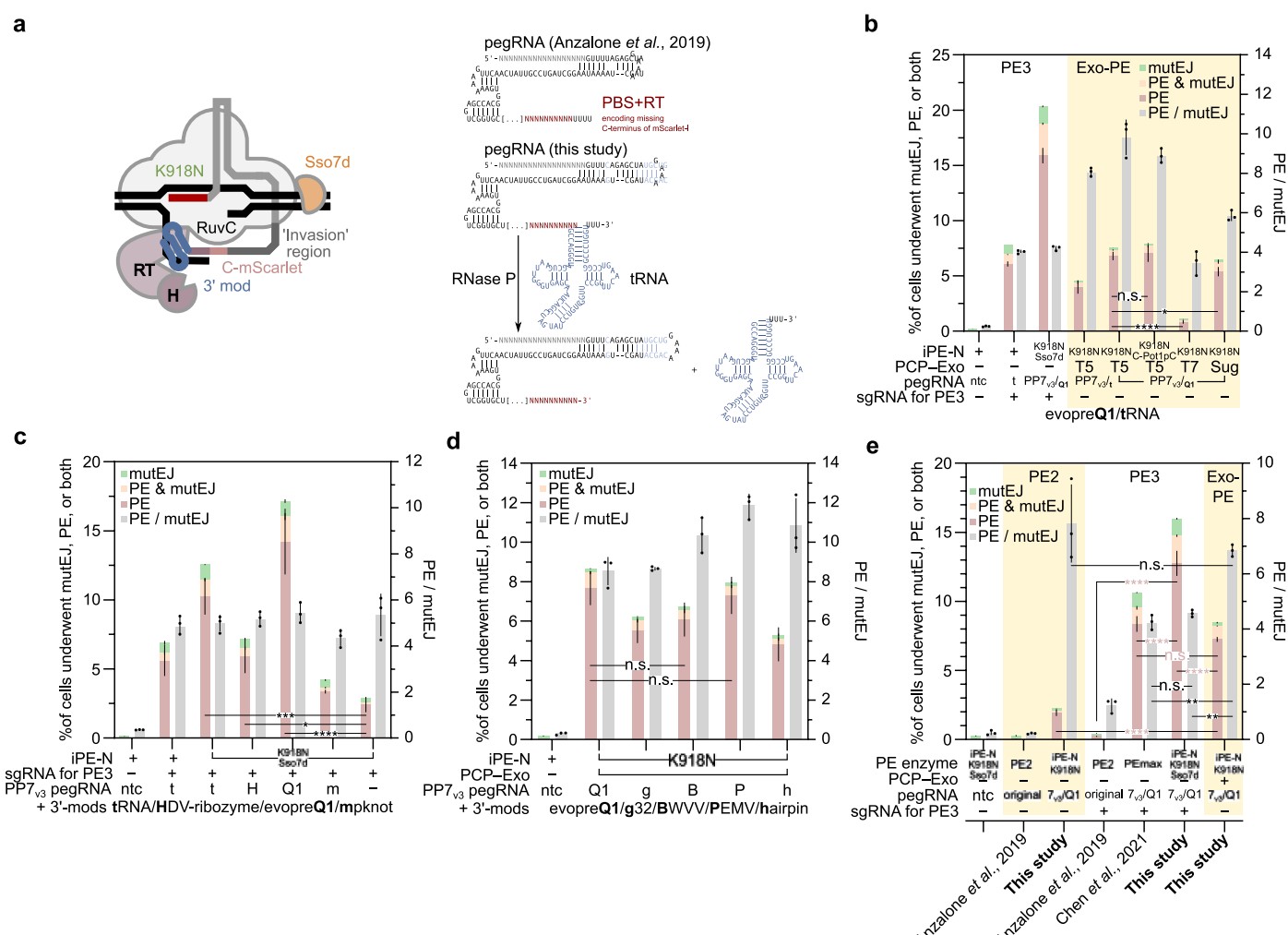

**Extended Data Fig. 3 | Screening for additional optimizations of the pegRNA architecture. a**, Schematic depiction of various modifications made to the PE architecture such as different 3' modifications of the pegRNA, mutations to the Cas9 enzyme (K918N), and addition of a dsDNA binding domain (Sso7d). We also compare a detailed depiction of the pegRNA from the original publication to the optimized pegRNA used in this study. **b**, An ssDNA binding domain (Pot1pC) was tested for its ability to enhance 'Exo-PE'. Furthermore, T5-like 5'-3'-DNA-exonuclease from *Klebsiella pneumoniae* Siphophage Sugarland as well as from *Escherichia* phage T7 was tested as an alternative 5'-3'-DNA-exonuclease/ flap endonuclease. For all subfigures, additional modifications (K918N and Sso7d) were utilized as described in the respective labels. Selected results of Bonferroni MCT after one-way ANOVA were shown for the fractional PE events, and are indicated by asterisks; n.s. *P* > 0.05, **P* < 0.05, *****P* < 0.0001. Data are given as the mean ± SD (*n* = 3, biological replicates). **c**, Comparison of different 3'-modifications instead of the 3'-tRNA (t) with iPE$_{K918N-Sso7d}$ in 'PE3'. Selected results of Bonferroni MCT after one-way ANOVA were shown for the fractional

PE events, and are indicated by asterisks; **P* < 0.05, ****P* < 0.001, *****P* < 0.0001. Bars represent the mean ± SD (*n* = 3, biological replicates). **d**, Comparison of additional 3'- secondary or tertiary motifs for 3'-stabilization to increase Exo-PE efficacy. Different secondary or tertiary RNA structures were tested for their ability to enhance Exo-PE: M1-7 tRNA, evopreQ1 pseudoknot, g32 pseudoknot, BWYV-FL1 pseudoknot, PEMV pseudoknot, and a simple RNA stem-loop (hairpin). Selected results of Bonferroni MCT after one-way ANOVA were shown for the fractional PE events; n.s. *P* > 0.05. Bars represent the mean ± SD (*n* = 3, biological replicates). **e**, Comparison of the top-performing iPE versions (iPE alterations are depicted in the sketch) to the original editor complex (PE2)[1] and the improved editor PEmax[2] for the editing strategies 'PE2', 'PE3', and 'Exo-PE'. iPE-N has a pegRNA harboring a PP7 aptamer (7$_{v3}$) and 3'-tevopreQ1[16] motif (Q1). Selected results of Bonferroni MCT after one-way ANOVA are shown for the fractional PE events and editing precision and are indicated by asterisks; n.s. *P* > 0.05, ***P* < 0.01, *****P* < 0.0001. Bars represent the mean ± SD (n = 3, biological replicates).

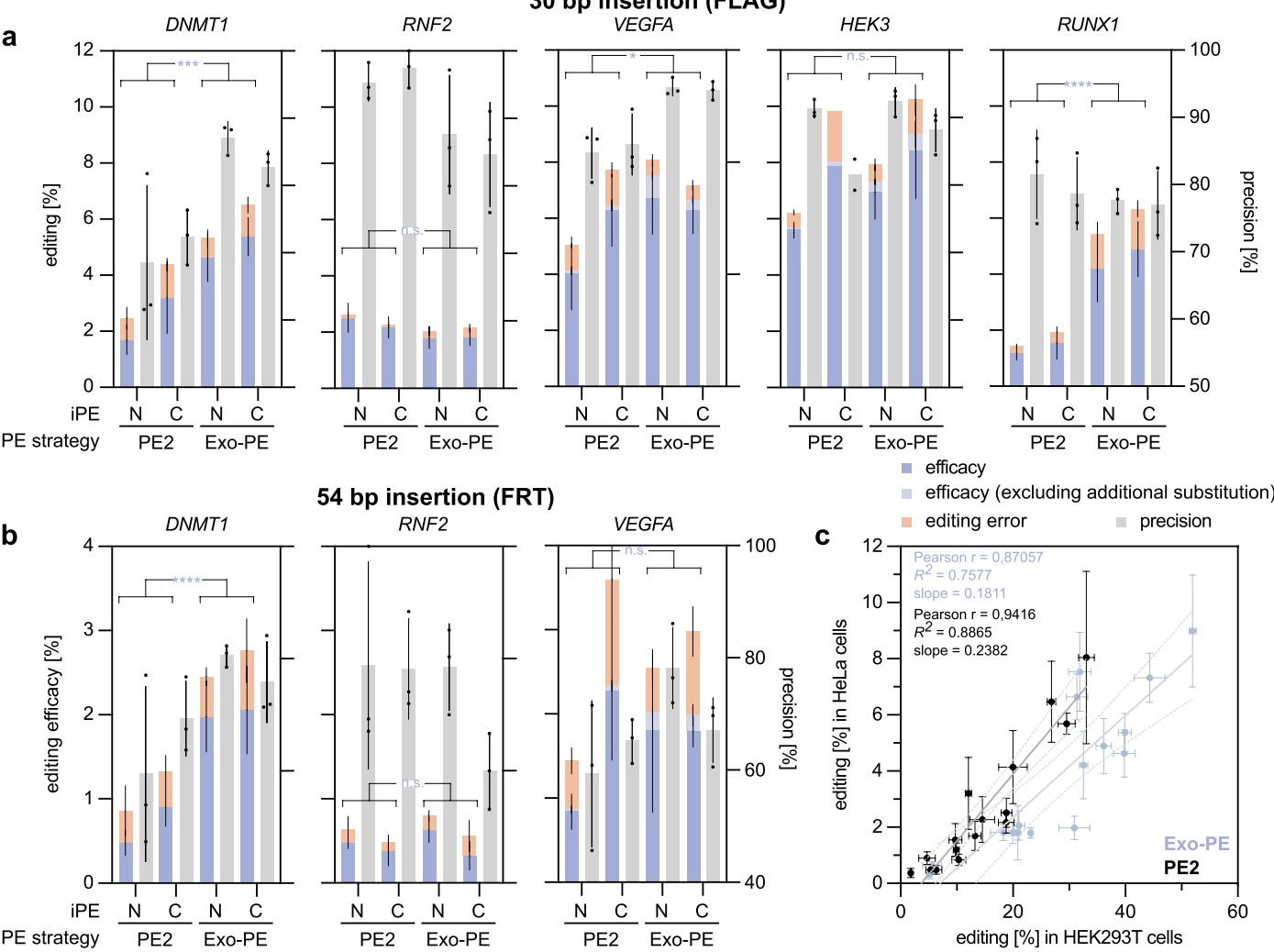

**Extended Data Fig. 4 | Editing efficacy and precision of 'Exo-PE' and 'PE2' for two different insertions in HeLa cells. a, b**, Experimental conditions and analyses are analogous to HEK293T data in Fig. 4c,d. Efficacy in HeLa was lower than in HEK293T cells, resulting in a larger SD for editing precision. Selected results of Bonferroni MCT for efficacy averaged over iPE-N/C after three-way ANOVA (showing main effects for panel a: PE strategy ($F = 32.762$, $P < 0.0001$), locus ($F = 46.97$, $P < 0.0001$), ($F = 9.085$, $P = 0.0045$), and the PE strategy/locus interaction ($F = 5.738$, $P = 0.001$) and panel b: PE strategy ($F = 13.665$, $P = 0.0011$), locus ($F = 25.373$, $P < 0.0001$) and PE strategy/locus interaction ($F = 7.46$, $P = 0.003$)) are indicated by asterisks; n.s. $P > 0.05$, *$P < 0.05$, ***$P < 0.001$, ****$P < 0.0001$. Bars represent mean ± SD ($n = 3$, biological replicates, except for *HEK3* C-PE2 where $n = 2$). **c**, Replotting of all efficacy data points for FLAG insertion from this figure and Fig. 4c,d shows the correlations of editing performance between HEK293T and HeLa ($P < 0.0001$ for both editing strategies). Gray lines indicate the 0.95 confidence bands of the linear regression. Please see Supplementary Table 1 for complete statistical results.

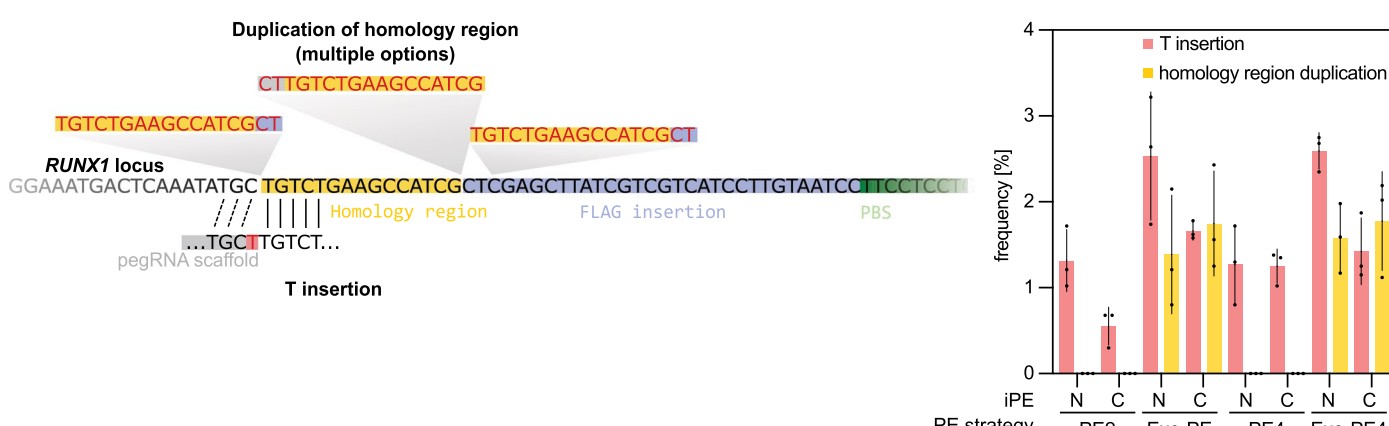

**Extended Data Fig. 5 | Editing errors in the *RUNX1* locus.** All conditions featuring 'Exo-PE' ('Exo-PE', 'Exo-PE4') led to a (partial) duplication of the pegRNA homology region at the insertion site (yellow), as well as an increased insertion of a single T at the end of the homology region (red). The T insertion could be linked to additional homology (dotted lines) in the pegRNA scaffold. Bars represent the mean ± s.d (*n* = 3, biological replicates).

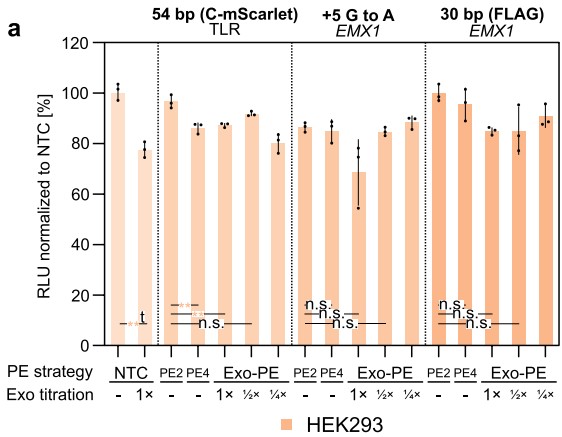

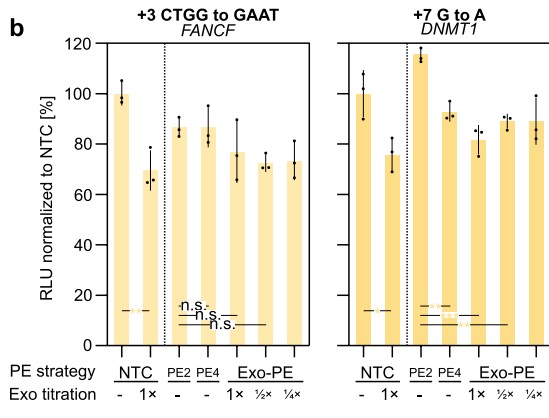

**Extended Data Fig. 6 | Evaluation of viable cell abundance upon 'Exo-PE' editing.** The abundance of viable cells was determined with a bioluminescent assay based on the cellular conversion of a luciferase substrate (see methods). iPE-C was tested in either 'PE2', 'PE4', or 'Exo-PE' for different insertion and substitution edits of varying length in **a**, HEK293, or **b**, HEK293T. For 'Exo-PE', the amount of T5-Exonuclease added to the experimental setup was titrated ('Exo-PE' with ½ Exo, ¼ Exo). Relative light unit [RLU] values were normalized against the non-targeting control (NTC). Cell viability significantly decreased upon the addition of untargeted T5-exonuclease (NTC+Exo). In comparison, 'Exo-PE' and 'PE4' showed a trend towards decreased RLU values compared to 'PE2', which was, however, not consistently observed across the different loci and different amounts of 'Exo-PE'. Selected results of Bonferroni MCT after a one-way ANOVA for each edit type are indicated by asterisks; n.s. $P > 0.05$, $*P < 0.05$, $**P < 0.01$, $***P < 0.001$. For the comparison of NTC vs. NTC+ 1x Exo, unpaired two-tailed t-tests were performed. Bars represent the mean ± s.d ($n = 3$, biological replicates).

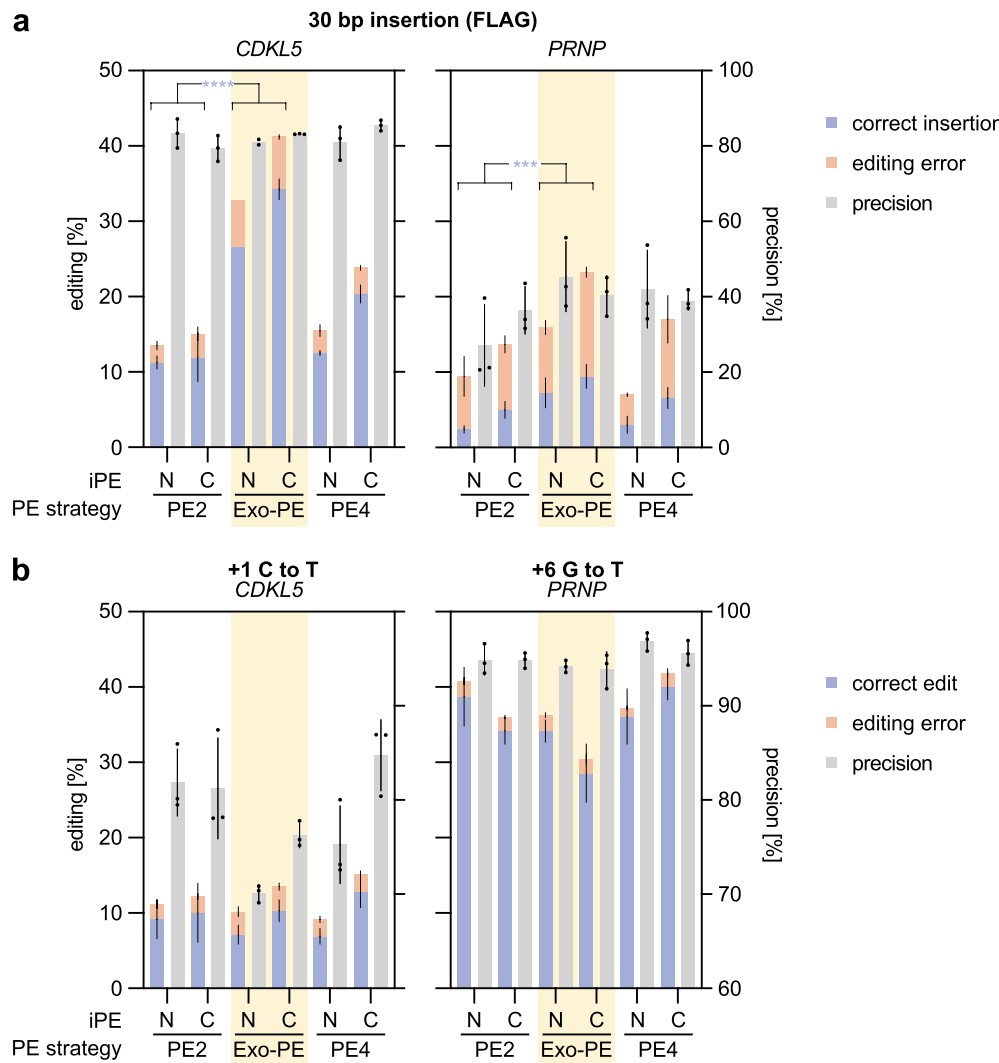

**Extended Data Fig. 7 | Performance of 'Exo-PE' at additional genomic sites with biomedical relevance. a**, Evaluation of 'Exo-PE' against the single-nick strategies 'PE2' and 'PE4' at disease-associated sites within the *CDKL5* and *PRNP* gene for 30 bp FLAG insertions. Selected results of Bonferroni MCT for efficacy averaged over iPE-N/C after three-way ANOVA (showing main effects for PE strategy ($F = 185,354$, $P < 0.0001$), locus ($F = 699.236$, $P < 0.0001$), iPE-N/C ($F = 68.975$, $P < 0.0001$), and the PE strategy/locus interaction ($F = 70.895$, $P < 0.0001$) are indicated by asterisks; ***$P < 0.001$, ****$P < 0.0001$. Bars represent the mean ± SD ($n = 3$, biological replicates, except for *CDKL5* N-Exo-PE (FLAG insertion) where $n = 2$). **b**, Evaluation of 'Exo-PE' against the single-nick strategies 'PE2' and 'PE4' for single-base therapeutic substitutions at the same genomic sites. Bars represent the mean ± SD ($n = 3$, biological replicates).

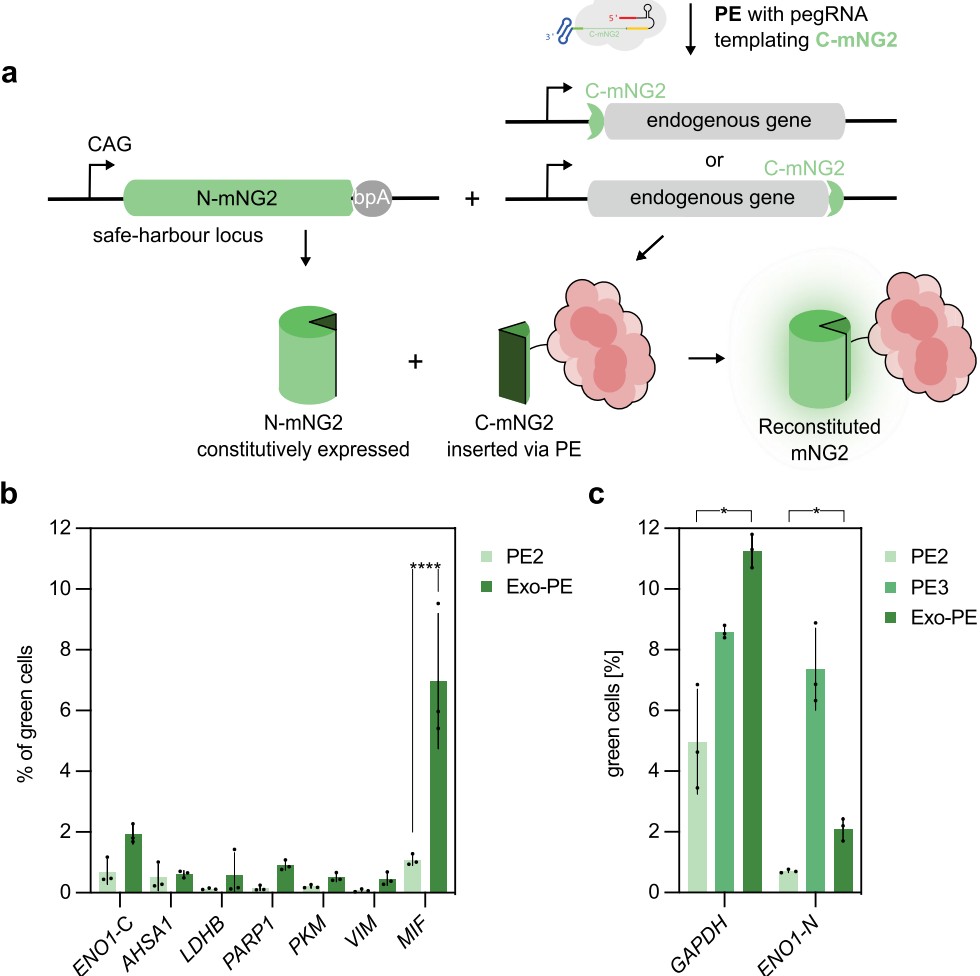

**Extended Data Fig. 8 | Insertion of a protein tag for bimolecular fluorescence complementation. a**, Schematic showing the Insertion of 48-57 bp (depending on 3xGly linker) coding for 11 amino acids of the split mNeonGreen (C-mNG2) to the C- or N-terminus of a target of interest. Corresponding proteins can be fluorescently labeled by on-target complementation of the split mNG2(1-10) (N-mNG2) co-expressed in a respective reporter cell line (HEK293T N-mNG2). **b,c**, Nine loci were targeted (*ENO1* was tagged either C- or N-terminally), and

% of green cells was determined via FACS. **b**, 'Exo-PE' showed an overall higher editing efficacy (Two-way ANOVA, $F_{(1, 28)}$ = 40.35, $P < 0.0001$), selected results of Bonferroni MCT are shown as ****$P < 0.0001$. Bars represent the mean ± SD ($n = 3$, biological replicates). **c**, A direct comparison of iPE for the three editing strategies 'PE2', 'PE3', and 'Exo-PE' (PCP-Exo) was conducted. Selected results of Bonferroni MCT from a two-way ANOVA (locus, editing strategy) are shown as *$P < 0.05$. Bars represent the mean ± SD ($n = 3$, biological replicates).

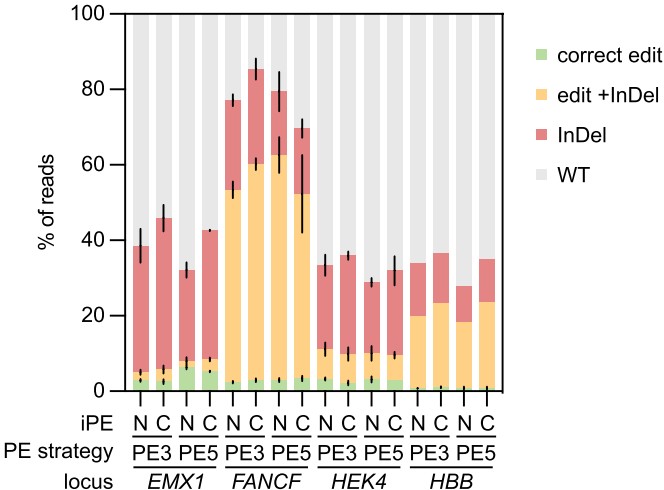

**Extended Data Fig. 9 | Frequency of editing errors (InDel events) in editing strategies with a secondary nick ('PE3'/'PE5') for the 30 bp FLAG-tag insertion.** Secondary nicks led to a substantial reduction in precision for insertion-type edits due to InDel events, which, depending on the target locus, often co-occurred with the incorporation of the intended edit (orange). The analysis is based on data shown in Fig. 6. Stacked bars represent the mean ± SD (n = 3, biological replicates, except *HBB* where n = 2).

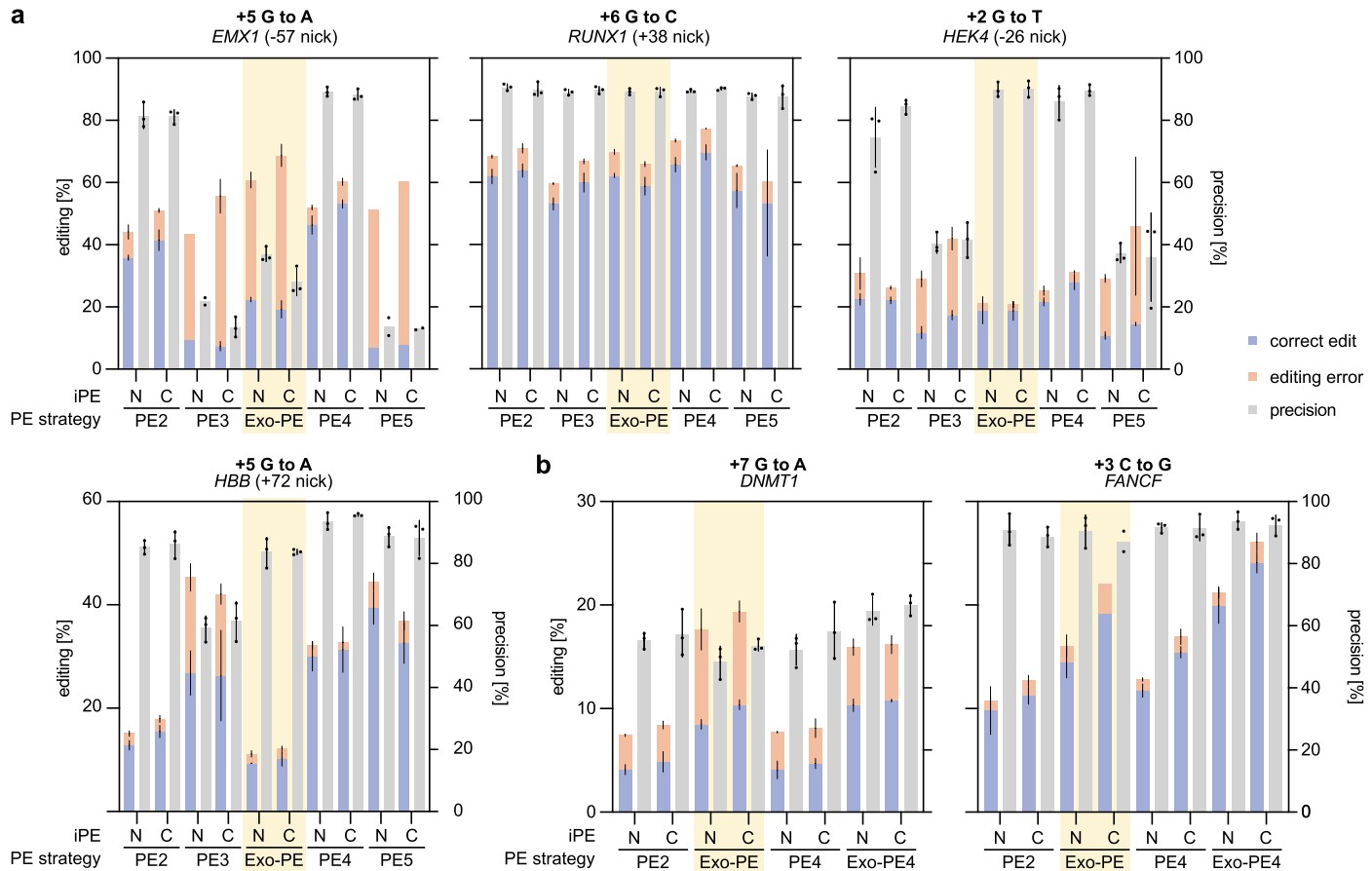

**Extended Data Fig. 10 | Benchmarking of the 'Exo-PE' strategy for substitution edits. a**, Substitutions on loci according to Fig. 4c. Nick positions of ngRNAs for 'PE3'/'PE5' strategies are indicated in brackets. **b**, Substitutions on two additional loci. Bars represent the mean ± SD (*n* = 3, biological replicates, except for *EMX1* N-PE3, *EMX1* N-PE5, *EMX1* C-PE5 and *FANCF* C-Exo-PE where *n* = 2).

# Reporting Summary

## Statistics

For all statistical analyses, confirm that the following items are present in the figure legend, table legend, main text, or Methods section.

| n/a | Confirmed | |
|---|---|---|
| ☐ | ☒ | The exact sample size (*n*) for each experimental group/condition, given as a discrete number and unit of measurement |
| ☐ | ☒ | A statement on whether measurements were taken from distinct samples or whether the same sample was measured repeatedly |
| ☐ | ☒ | The statistical test(s) used AND whether they are one- or two-sided<br>*Only common tests should be described solely by name; describe more complex techniques in the Methods section.* |
| ☐ | ☒ | A description of all covariates tested |
| ☐ | ☒ | A description of any assumptions or corrections, such as tests of normality and adjustment for multiple comparisons |
| ☐ | ☒ | A full description of the statistical parameters including central tendency (e.g. means) or other basic estimates (e.g. regression coefficient) AND variation (e.g. standard deviation) or associated estimates of uncertainty (e.g. confidence intervals) |
| ☐ | ☒ | For null hypothesis testing, the test statistic (e.g. *F*, *t*, *r*) with confidence intervals, effect sizes, degrees of freedom and *P* value noted<br>*Give P values as exact values whenever suitable.* |
| ☒ | ☐ | For Bayesian analysis, information on the choice of priors and Markov chain Monte Carlo settings |
| ☒ | ☐ | For hierarchical and complex designs, identification of the appropriate level for tests and full reporting of outcomes |
| ☐ | ☒ | Estimates of effect sizes (e.g. Cohen's *d*, Pearson's *r*), indicating how they were calculated |

*Our web collection on statistics for biologists contains articles on many of the points above.*

## Software and code

Policy information about availability of computer code

| Data collection | Geneious Prime 2023.1.2, BD FACSDiva 6.1.3 |
|---|---|
| Data analysis | Geneious Prime 2023.1.2, BD FACSDiva 6.1.3, NetGene2 (v2.42), R (3.4.1) with emmeans package |

For manuscripts utilizing custom algorithms or software that are central to the research but not yet described in published literature, software must be made available to editors and reviewers. We strongly encourage code deposition in a community repository (e.g. GitHub). See the Nature Portfolio guidelines for submitting code & software for further information.

## Data

Policy information about availability of data

All manuscripts must include a data availability statement. This statement should provide the following information, where applicable:
- Accession codes, unique identifiers, or web links for publicly available datasets
- A description of any restrictions on data availability
- For clinical datasets or third party data, please ensure that the statement adheres to our policy

Statistical source data are provided with this paper. All other data supporting the findings of this study are available from the corresponding author on request.

## Research involving human participants, their data, or biological material

Policy information about studies with human participants or human data. See also policy information about sex, gender (identity/presentation), and sexual orientation and race, ethnicity and racism.

| | |
|---|---|
| Reporting on sex and gender | n.a. |
| Reporting on race, ethnicity, or other socially relevant groupings | n.a. |
| Population characteristics | n.a. |
| Recruitment | n.a. |
| Ethics oversight | n.a. |

Note that full information on the approval of the study protocol must also be provided in the manuscript.

# Field-specific reporting

Please select the one below that is the best fit for your research. If you are not sure, read the appropriate sections before making your selection.

☒ Life sciences    ☐ Behavioural & social sciences    ☐ Ecological, evolutionary & environmental sciences

For a reference copy of the document with all sections, see nature.com/documents/nr-reporting-summary-flat.pdf

# Life sciences study design

All studies must disclose on these points even when the disclosure is negative.

| | |
|---|---|
| Sample size | Preliminary eTLR-based experiments showed that there was only a small variation over independent replicates, so we decided on n = 3 for subsequent experiments. |
| Data exclusions | Some NGS data points were excluded because of technical failure (no reads). |
| Replication | Exo-PE effect were successfully replicated with independent biological samples across multiple loci and cell lines, please see "Statistics and Reproducibility" statement. |
| Randomization | All of our experiments were performed in multi-well cell culture plates. Wells were allocated to the different treatment groups specified by the particular experiment without determining any specific property of the cells in a given well for selecting a specific condition. Processing of the samples was performed without prior knowledge of what exact condition a given sample belonged to avoid processing bias. |
| Blinding | All experiments were performed, if possible, with master mixes and with multichannel pipettes to exclude unconscious biases during experimental setup. However, the individual experimenter was not blinded for the conditions as this was not feasible. |

# Reporting for specific materials, systems and methods

We require information from authors about some types of materials, experimental systems and methods used in many studies. Here, indicate whether each material, system or method listed is relevant to your study. If you are not sure if a list item applies to your research, read the appropriate section before selecting a response.

## Materials & experimental systems

| n/a | Involved in the study |
|---|---|
| ☒ | ☐ Antibodies |
| ☐ | ☒ Eukaryotic cell lines |
| ☒ | ☐ Palaeontology and archaeology |
| ☒ | ☐ Animals and other organisms |
| ☒ | ☐ Clinical data |
| ☒ | ☐ Dual use research of concern |
| ☒ | ☐ Plants |

## Methods

| n/a | Involved in the study |
|---|---|
| ☒ | ☐ ChIP-seq |
| ☐ | ☒ Flow cytometry |
| ☒ | ☐ MRI-based neuroimaging |

# Eukaryotic cell lines

Policy information about <u>cell lines and Sex and Gender in Research</u>

| | |
|---|---|
| Cell line source(s) | HEK293T cells (ECACC: 12022001, Sigma-Aldrich), HEK293 cells (85120602, Sigma-Aldrich), HeLa (93021013, Sigma-Aldrich) |
| Authentication | All cells were purchased from trusted vendors. |
| Mycoplasma contamination | Cell lines were regularly tested for mycoplasma contamination with Hoechst 3334, which visualizes non-nuclear cytosolic speckles in case of contamination. |
| Commonly misidentified lines (See <u>ICLAC</u> register) | No commonly misidentified cell lines were used in this study. |

# Flow Cytometry

## Plots

Confirm that:

☒ The axis labels state the marker and fluorochrome used (e.g. CD4-FITC).

☒ The axis scales are clearly visible. Include numbers along axes only for bottom left plot of group (a 'group' is an analysis of identical markers).

☒ All plots are contour plots with outliers or pseudocolor plots.

☒ A numerical value for number of cells or percentage (with statistics) is provided.

## Methodology

| | |
|---|---|
| Sample preparation | Cells were detached with Accutase, pelleted (200 rcf and 5 min), and resuspended in ice-cold 0.4% formalin for 10 min. Fixed cells were pelleted again (200 rcf and 5 min) and resuspended in ice-cold 200 µl DPBS for FACS analysis. |
| Instrument | BD FACSaria II (BD Biosciences) |
| Software | BD FACSDiva Software (Version 6.1.3, BD Biosciences) |
| Cell population abundance | For FACS analysis, at least 30k events were recorded per conditon. |
| Gating strategy | The main population of the cells was gated first according to their FSC-A and SSC-A. Secondly, single cells were gated using FSC-A and FSC-W. The final gate (red and green fluorescence) was used to determine the number of undergoing mutEJ or prime editing |

☒ Tick this box to confirm that a figure exemplifying the gating strategy is provided in the Supplementary Information.

