## [Peer Review File · Nature Methods]

Peer Review Information

Manuscript Title: Exonuclease-enhanced prime editors

Corresponding author name(s): Gil Westmeyer

Editorial Notes: None

Reviewer Comments & Decisions:

Decision Letter, initial version:

Dear Professor Westmeyer,

Your Brief Communication entitled "Modular architecture for enhanced prime editors" has now been seen by 3 reviewers, whose comments are attached. While they find your work of some potential interest, they have raised concerns which in our view are sufficiently important that they preclude publication of the work in Nature Methods.

We will consider looking at a revised manuscript only if further experimental data allow you to address all the major criticisms of the reviewers (unless, of course, something similar has by then been accepted at Nature Methods or appeared elsewhere). This includes submission or publication of a portion of this work somewhere else.

The required new data include, but are not limited to experiments to fully address the concerns on lack of validation on multiple endogenous targets and benchmarking against existing PE systems. We hope you understand that until we have read the revised paper in its entirety we cannot promise that it will be sent back for peer-review.

If you are interested in revising this manuscript for submission to Nature Methods in the future, please contact me to discuss your appeal before making any revisions. Otherwise, we hope that you find the reviewers' comments helpful when preparing your paper for submission elsewhere.

Sincerely,
Madhura

Madhura Mukhopadhyay, PhD
Associate Editor
Nature Methods

Reviewers' Comments:

Reviewer #1:

Remarks to the Author:

Prime editing (PE) is the latest of CRISPR/Cas based precise genome editing technologies. It has attracted a considerable attention of various research communities due to a premise of theoretically unrestricted genome editing in all three Woese's domains of life. In order to improve PE, Truong et al tested multiple ideas, including novel ones such as fusion of RNaseH and 5' to 3' exonucleases. The authors started off by designing and testing a nice reporter system which presumably can report nearly all possible genome editing outcomes. Then, they attached an exonuclease to the prime editor to remove the 5' unedited flap resulting (with other improvements) in "Exo-PE" and investigated the N terminal fusion of reverse transcriptase (RT) to Cas9 nickase (with addition of RNaseH fusion protein and optimized NLS) resulting in "ePE". The authors also incorporated previously published beneficial mutations into Cas9 nickase of prime editor, increased local concentration of PE at the target site with histone like proteins as well as investigated several enhanced pegRNAs. Overall, the developed reporter system facilitated rapid screening and comparison of different PE systems. This manuscript builds upon several recent research articles demonstrating improved prime editing. However, this reviewer has major concerns about this study and find the results preliminary (and the study as incomplete).

Major concerns:

The most significant concern that I have for this work is that the whole study is based on one single target site (in the reporter gene) with one pegRNA design (by inserting a ~50 nt sequence) for the same edit in a single cell type (HEK293T). It is well known that PE efficiencies very greatly depending on target sites and edits. Different PE systems may favor certain editing outcomes than others. It is nice that the authors can use a streamlined reporter system to rapidly identify some promising PE systems. However, after such a screen, they must benchmark the PE systems by testing many endogenous target sites with different editing outcomes (deletions, insertions, and nucleotide changes) in multiple human cell lines. Without seeing such demonstrations and validations, this reviewer remains unconvinced about the improved PE systems. Along this line, it is surprising to see that PPE (paired prime editor) performed poorly in this study. The authors only cited on PPE paper. However, such PPE systems have been reported in at least four peer-reviewed publications 1-4, and they all claimed higher editing efficiencies than PE2 and PE3. It is likely the PPE design/system that the authors used for the only target site and only pegRNA is not optimal or very poor (and hence not a good representative for PPE). Again, this just

suggest the importance of comprehensive testing at multiple sites with different edits in order to compare and benchmark the PE systems.

Second, the authors developed a comprehensive FACS screening assay reducing the amount of time and work needed to attain the preliminary results. However, the most important results of the manuscript (comparing improved prime editors with previously published prime editors) should be validated with NGS. When the authors start more testing on more target sites, they should also use NGS to precisely verify and quantify the editing outcomes.

Minor points:

1. The article is written in the style of a full “Article” with 5 main figures. I don’t know why it is called “Brief Communication”.
2. The authors did not share a lot of information about the vectors used in this study. I recommend adding vector maps, list of vectors and primers used in this study, as well as nucleotide sequence of newly developed reporter used in FACS assay in Supplemental file. Additionally, relevant vectors could be deposited to plasmid depository like Addgene.
3. I suggest the authors compare MLH1dn recruitment strategy (Chen et al, 2021) with their strategy recruiting exonuclease since both strategies aim to solve problem from a similar vantage point.
4. Does the scale bar in Fig 1b represent the same size on brightfield insets of fluorescent microscopy photos?
5. It would be informative from spatial perspective if scalebar (in nm or angstroms) would be added to Fig 2c.
6. The architecture of ePE is depicted in Fig 2A. It would be faster to find the ePE architecture if it was labeled in Fig 2A, similar as it is done in Fig 5D.
7. In the last paragraphs before the section “Cas9-enhancing mutations and histone-like proteins improve primer editing efficacy” please define the “enzyme” in “... higher editing efficacy against the original enzyme...”
8. When K918N mutation is introduced into text, it is not mentioned what is the biological reason introducing this mutation nor it is referenced. The same mutation is referenced later in manuscript.
9. Figure 5A is quite crowded.
10. Supplemental Figure 7A: I couldn’t find any explanation why T5-5’-Exo is N terminally fused to PCP and other exonucleases (FEN1, Exo1) C terminally fused.
11. In subsection “Synergy between enhanced prime editing enzymes and enhanced pegRNAs” there is a contrast between described results in text and data presented in Fig 5B about EvoPreQ1 pseudoknot.

References:

- 1 Choi, J. et al. Precise genomic deletions using paired prime editing. Nature Biotechnology, doi:10.1038/s41587-021-01025-z (2021).

2 Lin, Q. et al. High-efficiency prime editing with optimized, paired pegRNAs in plants. *Nat Biotechnol* 39, 923-927, doi:10.1038/s41587-021-00868-w (2021).

3 Zhuang, Y. et al. Increasing the efficiency and precision of prime editing with guide RNA pairs. *Nature Chemical Biology* 18, 29-37, doi:10.1038/s41589-021-00889-1 (2022).

4 Anzalone, A. V. et al. Programmable deletion, replacement, integration and inversion of large DNA sequences with twin prime editing. *Nature Biotechnology*, doi:10.1038/s41587-021-01133-w (2021).

Reviewer #2:

Remarks to the Author:

Comments:

Truong et al. report an enhanced traffic light reporter (eTLR) to differentiate all possible NHEJ/MMEJ-mediated insertions or deletions (indels) events from a successful HDR/primer editing (PE)- mediated events. With this reporter system, authors describe several optimizations to improve the prime editing. Firstly, they optimize the protein architecture of PE to create enhanced PE (ePE), by N-terminal fusion of RT to nCas9, optimization of RNA codon usage, supply NLS, introduction of enhancing mutation K918N, and addition of histone-like domain (Sso7d). Secondly, they introduce an aptamer into pegRNA, which recruits a 5'-DNA-exonuclease to the primary nick to degrade the 5'-flap actively without secondary nicks. The authors demonstrate that these strategies increase prime editing efficiencies and the combination ePE with engineered pegRNA further facilitate efficacy in cell line with eTLR system. Although it is a useful reporter system to optimize PE, this study is a relatively narrow or weak improvement strategy. The authors may test or combine more strategies, such as PE4/PE5, twin-PE, or induce mutations on RT or PBS, to further improve the editing efficiency.

Major issues:

1. While the authors claim that these strategies improve prime editing efficacies in their well-designed eTLR system, and even "Exo-PE" achieve editing efficacy comparable to PE3 with fewer undesired indels, it's lack of more data that whether ePE or Exo-PE is used as a general effective improved strategy. More endogenous loci need to be tested in more other cell types, such as in previous study HEK3, DNMT1, VEGFA, RUNX1 or others in HEK293, HeLa, U2OS or other cells.
2. Although the authors mention many disadvantages of PE3, such as increased indels, and limitations of PE3 in non-mammalian systems, the improved strategy is still mostly based on PE3. In Fig. 2d-e, ePE increases editing efficiency based on PE2, while significantly increase in combination with PE3. Thus, the significantly improvement is mainly attributed to PE3.
3. In Fig.3, although addition of Cas9 mutation K918N or fusion of histone-like domains C-Sso7d E53L or K918N improve PE-efficacy, along with detrimental effects on editing precision (PE/mutEJ), mutEJ-

mediated indels events are also increased. And these improvements are also based on PE3. Thus, the authors need to clarify whether ePE are a supplement to PE3 or an alternative strategy.

4. The authors describe an aptamer-tagged pegRNA to improve editing efficacy, which is introduction of PP7 aptamer into the tetraloop junction between the crRNA and the tracrRNA, by recruiting overexpressed PCP:superNLS:5'-Exo to alleviate the incorporation of the de novo synthesized flap. In Fig. 4b-c, PP7/Pv3-tagged pegRNA and aptamer-mediated recruitment indeed provide a 7-fold increase in efficacy and few indel compared to PE2, but still slightly lower than PE3 efficacy along with a higher editing precision. However, whether these improvements have a big impact on actual genome editing, authors need to test endogenous sites to clarify this.

5. In Fig.5e, combining with a 3'-stabilization motif, Exo-PE (ePE/K918N/7v3/Q1) significantly a higher editing precision compared to original PE3, in this reporter. I recommend that authors compare the Exo-PE with PE4 or twin-PE, which are current alternative strategies for PE3.

Minor points:

1. The eTLR system is indeed well-designed and simple reporter to screen and compare editing efficacy and indels of different PE versions. There should be a parallel comparison of all three HDR/PE, mutEJ, and the ratio between HDR/PE and mutEJ values. For example, in Fig. 1c, it's obviously not true that two samples with no HDR donor or Cas9/sgRNA has a high HDR/mutEJ, which is due to weak HDR or mutEJ value. To better use it, I recommend that the authors could set evaluation criteria or baseline.
2. In this study, based on original, the authors created Exo-PE or ePE, and further various version. The authors need to rename and make a more clarity naming.
3. In Fig. 4c, Pv3 should be described. What's different to PP7v3?

Reviewer #3:

Remarks to the Author:

In this study, Truong et al present multiple improvements of the prime editor using rational design. To systematically improve the prime editing architecture, they firstly developed an enhanced traffic light reporter (eTLR) system, which can report all editing events. Using the eTLR system to measure editing efficiency, they have evaluated a series of optimizations on the PE architecture, including the rearrangement of the RT to the N-terminal of the Cas9 nickase (RT-nCas9), a new codon-optimization, improving the nuclear localization using superNLS, introducing Cas9-enhancing mutations (ePEK918N) and histone-like domains (ePEsso7d), and recruiting a 5'-DNA-exonuclease. They demonstrate that their optimization strategies could be combined with an engineered pegRNA to further improve PE editing efficiencies. However, the authors need to incorporate additional data that is critical for acceptance of this manuscript, especially the evaluation of their technology at endogenous genomic target sites using their optimized PE. I recommend accepting the manuscript only if the following concerns can be addressed in a revised manuscript.

Major concerns:

1. The authors need to evaluate the editing efficiency of their newly designed tools, ePEK918N-Sso7d and Exo-PE, at endogenous targets. This is a critical data point for all newly developed genome editing tools as many times the improvements in editing when evaluated using a reporter system do not translate to improvements in editing at endogenous sites. I recommend the authors compare editing efficiencies between their optimized prime editors and previous generations of prime editors across at least 10 target sites in at least two different cell types.

2. Considering the improvements in on-target editing efficiency of the newly optimized prime editors described in this manuscript, the authors should also evaluate the off-target effects of these improvements. Endogenous off-target effects of the optimized prime editor should be assessed across at least 10 endogenous off-target targets with less than 3 mismatches between the on-target and off-target pegRNAs.

2. Can the authors explain why co-expressing the pegRNA with the PCP:superNLS:5'-Exo is more efficient than PE2 in Fig. 4b? The authors should include a more detailed description in the revised manuscript.

3. Can the authors describe why the CAG-driven pegRNAs show lower editing efficiencies compared to the U6-driven pegRNAs with a CG pair in the pegRNA scaffold in Extended Data Fig. 4?

4. I suggest the authors introduce the function of and thoroughly describe Sso7d in the context of “we next hypothesized that PE efficacy might benefit from an increased local concentration at nucleosomes without increasing the general expression strength. Therefore, we fused the histone-like 7d protein from the archaea *Sulfolobus solfataricus* (Sso7d with E35L to inactivate cryptic ds/ssRNase activity)²⁸ to the C-terminus of ePE.”

Minor concerns:

1. Please clarify “Pv3” in figure 4 legends.

Author Rebuttal to Initial comments

Letter to Referees: Exonuclease-enhanced prime editors**Reviewers' Comments:****General remarks to all Reviewers**

We like to thank all Reviewers for their constructive and detailed feedback, which helped us to improve the manuscript substantially. In the process, we have refocused the manuscript on the new prime editing strategy 'Exo-PE', based on recruiting a 5'-3' exonuclease to and enhanced Prime Editor (ePE) protein to enzymatically generate space for incorporating the reverse transcribed 3' flap.

In short, we have now re-confirmed the improvements combined in ePE and then validated the improved performance of 'Exo-PE' on 5 loci as requested (DNMT1, RNF2, VEGFA, HEK3, RUNX1) with 2 different challenging edits (30 and 54 bp insertions) in 2 different cell types.

In addition, we targeted 9 additional endogenous loci in which we inserted the DNA coding for 11 amino acids of a split fluorescence protein to enable biomolecular fluorescence complementation as a biotechnological application in analogy to this prominent research project (<https://opencell.czbiohub.org/about>).

Please find our detailed point-by-point responses below.

Reviewer #1:

Remarks to the Author:

Prime editing (PE) is the latest of CRISPR/Cas based precise genome editing technologies. It has attracted a considerable attention of various research communities due to a premise of theoretically unrestricted genome editing in all three Woese's domains of life. In order to improve PE, Truong et al tested multiple ideas, including novel ones such as fusion of RNaseH and 5' to 3' exonucleases. The authors started off by designing and testing a nice reporter system which presumably can report nearly all possible genome editing outcomes. Then, they attached an exonuclease to the prime editor to remove the 5' unedited flap resulting (with other improvements) in "Exo-PE" and investigated the N terminal fusion of reverse transcriptase (RT) to Cas9 nickase (with addition of RNaseH fusion protein and optimized NLS) resulting in "ePE". The authors also incorporated previously published beneficial mutations into Cas9 nickase of prime editor, increased local concentration of PE at the target site with histone like proteins as well as investigated several enhanced pegRNAs. Overall, the developed reporter system facilitated rapid screening and comparison of different PE systems. This manuscript builds upon several recent research articles demonstrating improved prime editing. However, this reviewer has major concerns about this study and find the results preliminary (and the study as incomplete).

Communication with editor → NMETH-BC47950C_communication_with_editor

R1-0

We thank the reviewer for recognizing the powerful reporter system for gene editing events we developed and the set of substantial improvements on the prime editing architecture we achieved. Please see our point-by-point responses on how we now validated the improved PE systems in detail in line with your suggestions.

Major concerns:

The most significant concern that I have for this work is that the whole study is based on one single target site (in the reporter gene) with one pegRNA design (by inserting a ~50 nt sequence) for the same edit in a single cell type (HEK293T). It is well known that PE efficiencies vary greatly depending on target sites and edits. Different PE systems may favor certain editing outcomes than others. It is nice that the authors can use a streamlined reporter system to rapidly identify some promising PE systems. However, after such a screen, they must benchmark the PE systems by testing many endogenous target sites with different editing outcomes (deletions, insertions, and nucleotide changes) in multiple human cell lines. Without seeing such demonstrations and validations, this reviewer remains unconvinced about the improved PE systems. Along this line, it is surprising to see that PPE (paired prime editor) performed poorly in this study. The authors only cited on PPE paper.

However, such PPE systems have been reported in at least four peer-reviewed publications 1-4, and they all claimed higher editing efficiencies than PE2 and PE3. It is likely the PPE design/system that the authors used for the only target site and only pegRNA is not optimal or very poor (and hence not a good representative for PPE). Again, this just suggests the importance of comprehensive testing at multiple sites with different edits in order to compare and benchmark the PE systems.

R1-1

We have now assessed the improved PE architectures for different endogenous loci, such as DNMT1, RNF2, VEGFA, HEK3, RUNX1, ENO1-C, ENO1-N, AHSA1, LDHB, PARP1, PKM, VIM, MIF, and GAPDH. We find that the Exo-PE strategy has an overall better editing effect in direct comparison to the PE2 strategy with identical pegRNAs (statistical results in the main text, new **Fig. 3**, new **Supplementary Fig. 8**).

We also provide data showing that the type of fusion (N- or C-terminal fusion of RT to nCas9) may have a relevant effect for a given site of interest. In addition, all statistical results are provided in new **Supplementary Table 2**.

With respect to the PPE, we have now compared ePE directly for editing strategies that necessitate a second pegRNA (or sgRNA), including classical PPE and twin-PE (which, in contrast to PPE, does not have 'homology regions') and found a performance on par with PE3 on eTLR but with a decreased editing precision.

Communication with editor → NMETH-BC47950C_communication_with_editor

Figure R1 | Comparison of the 'PE3' vs. 'PPE' vs. 'twin-PE' strategies using the enhanced prime editor complex ePE-N. Bars represent the mean \pm s.d (n = 3, biological replicates).

As Reviewer 1 mentioned, there have been a lot of paired-pegRNA PE (PPE) strategies (Twin-PE, GRAND, HOPE, etc., <https://doi.org/10.1038/s41576-022-00541-1>) and the performance also depends on careful optimizations of the pegRNAs for a specific locus. It is thus conceivable that the performance of those paired-pegRNA strategies can also be further optimized for the long insertion into eTLR.

Still, our goal was to search for a method to improve the 'PE2 strategy' without the necessity for a secondary nicking sgRNA ('PE3 strategy') or a second pegRNA (PPE).

Thus, we have validated Exo-PE directly against PE2 on all endogenous loci with identical reference pegRNAs taken from previous publications to eliminate any confounds from free design parameters and optimization rounds for the second sgRNA or pegRNA.

We also benchmarked against 'PE3' when we applied 'Exo-PE' for fluorescent tagging of target proteins (new **Supplementary Fig. 8**). In this application, 'ExoPE' again outperformed 'PE2', and for one locus, showed similar efficacy as 'PE3'.

Second, the authors developed a comprehensive FACS screening assay reducing the amount of time and work needed to attain the preliminary results. However, the most important results of the manuscript (comparing improved prime editors with previously published prime editors) should be validated with NGS. When the authors start more testing on more target sites, they should also use NGS to precisely verify and quantify the editing outcomes.

R1-2

As stated in our response to your point R1-1, we have now included NGS results for different endogenous loci (4 of 5 different endogenous loci showed better efficacy with Exo-PE compared to PE2) in HEK293T and HeLa cells (new **Figure 3**, new **Supplementary Figure 7**), which demonstrates that Exo-PE yields an overall higher editing efficacy and precision than PE2 using identical pegRNAs. NGS was utilized for an in-depth analysis of editing outcomes. In addition,

Communication with editor → NMETH-BC47950C_communication_with_editor

we used the bimolecular fluorescence complementation system to fluorescently tag target proteins and observed that, in 7 out of 9 loci, Exo-PE was significantly better in terms of editing efficacy (% of green cells, new **Supplementary Figure 8**).

Minor points:

1. The article is written in the style of a full "Article" with 5 main figures. I don't know why it is called "Brief Communication".

R1-3

Based on the Reviewers' feedback, we have now streamlined the manuscript focusing on Exo-PE and have reduced the number of main Figures to 3 (with 8 Supplementary Figures and 2 Supplementary Tables) in line with a Brief Communication.

2. The authors did not share a lot of information about the vectors used in this study. I recommend adding vector maps, list of vectors and primers used in this study, as well as nucleotide sequence of newly developed reporter used in FACS assay in Supplemental file. Additionally, relevant vectors could be deposited to plasmid depository like Addgene.

R1-4

We will submit all of the constructs to Addgene, including detailed meta-information on constructing the FACS reporter line.

3. I suggest the authors compare MLH1dn recruitment strategy (Chen et al, 2021) with their strategy recruiting exonuclease since both strategies aim to solve problem from a similar vantage point.

R1-5

The Exo-PE strategy focuses on avoiding the flap competition by actively degrading the original 5' flap to make space for the RT'ed *de novo* 3' flap. We validated this strategy specifically for longer insertions because, in this capability, PE differentiates itself from base editors and promises advantages over DBS-triggered HDR-dependent editing techniques. Whereas transient MHL1dn co-expression can help to protect short mismatch intermediates from MMR, it was already shown in Chen *et al.* (<https://doi.org/10.5281/zenodo.5551032>) that the introduction of additional (silent) mutations - as present in our long edits (30/ 54 bp) - are just as effective as MLH1dn in evading MMR.

4. Does the scale bar in Fig 1b represent the same size on brightfield insets of fluorescent microscopy photos?

Communication with editor → NMETH-BC47950C_communication_with_editor

R1-6

Yes, the insets (now **Supplementary Figure 1**) are the same segment as shown in the fluorescence channel.

5. It would be informative from spatial perspective if scalebar (in nm or angstroms) would be added to Fig 2c.

R1-7

Thank you for the suggestion. We have done so. The figure is now **Supplementary Fig. 2b**.

6. The architecture of ePE is depicted in Fig 2A. It would be faster to find the ePE architecture if it was labeled in Fig 2A, similar as it is done in Fig 5D.

R1-8

We introduced the suggested labels into the subfigure (now **Supplementary Fig. 2a**).

7. In the last paragraphs before the section "Cas9-enhancing mutations and histone-like proteins improve primer editing efficacy" please define the "enzyme" in "... higher editing efficacy against the original enzyme..."

R1-9

The enzyme referred to in the respective text and figure (now **Supplementary Fig. 2f**) was ePE-N which showed superior performance when compared to the original PE2 enzyme. Please note that in the original paper, PE2 sometimes refers to the enzyme, *i.e.*, the nCas9-RT fusion, and sometimes to 'PE2 strategy' (so prime editing with only the primary pegRNA without secondary nicking sgRNAs). In Supplementary Fig. 2f, we compare the ePE-N enzyme with the original PE2 enzyme in 'PE2' and 'PE3' settings.

8. When K918N mutation is introduced into text, it is not mentioned what is the biological reason introducing this mutation nor it is referenced. The same mutation is referenced later in manuscript.

R1-10

This mutation in Cas9 was describe in Hand *et al.* (2021) (doi: 10.1089/crispr.2020.0092) to increase Cas9's nuclease activity. We thus assessed whether it also enhances the nickase activity of Cas9 H840A or N863A. However, we found that its beneficial effects were true for the eTLR as target but - in the context of the several other improvements in ePE, did not provide additional advantages on other loci. We thus did not move forward with those mutations for the final ePE complexes, which we validated on endogenous loci as described in the text.

Communication with editor → NMETH-BC47950C_communication_with_editor

9. Figure 5A is quite crowded.

R1-8

We have changed the design of the figures to improve readability.

10. Supplemental Figure 7A: I couldn't find any explanation why T5-5'-Exo is N terminally fused to PCP and other exonucleases (FEN1, Exo1) C terminally fused.

R1-9

As shown in Figure R2 below, we did not see any major difference between N- or C-terminally fused PCP to recruit T5 5'exonuclease. For FEN1 and EXO1, we used both exonucleases as N-terminal domains since their native interaction domains (deleted in our chimeras) are located at their respective C-termini. Thus, we also placed PCP as a C-terminal domain. PCP has been used in the literature in both orientations successfully. Indeed, both directional fusions with T5 5'-exonuclease were active, as shown in **Figure R2**.

Figure R2 | Comparison of the site of PCP fusion for recruiting the T5 5'exonuclease to the PE complex.

11. In subsection "Synergy between enhanced prime editing enzymes and enhanced pegRNAs" there is a contrast between described results in text and data presented in Fig 5B about EvoPreQ1 pseudoknot.

R1-10

Communication with editor → NMETH-BC47950C_communication_with_editor

We apologize if the statements in the original text related to Figure 5b (now **Supplementary Figure 3f**) were not in line with the header. The figure shows that 3' stabilization of the pegRNA is essential for editing performance.

--

We would like to thank the Reviewer again for the detailed and productive comments, which greatly helped us to validate the performance of Exo-PE and improve the manuscript.

We believe that Exo-PE is a potent addition to the PE toolbox that should be of interest to other research teams working on improving gene editing.

Communication with editor → NMETH-BC47950C_communication_with_editor

Reviewer #2: Remarks to the Author: Comments:

Truong et al. report an enhanced traffic light reporter (eTLR) to differentiate all possible NHEJ/MMEJ-mediated insertions or deletions (indels) events from a successful HDR/primer editing (PE)- mediated events. With this reporter system, authors describe several optimizations to improve the prime editing. Firstly, they optimize the protein architecture of PE to create enhanced PE (ePE), by N-terminal fusion of RT to nCas9, optimization of RNA codon usage, supply NLS, introduction of enhancing mutation K918N, and addition of histone-like domain (Sso7d). Secondly, they introduce an aptamer into pegRNA, which recruits a 5'-DNA-exonuclease to the primary nick to degrade the 5'-flap actively without secondary nicks. The authors demonstrate that these strategies increase prime editing efficiencies and the combination ePE with engineered pegRNA further facilitate efficacy in cell line with eTLR system. Although it is a useful reporter system to optimize PE, this study is a relatively narrow or weak improvement strategy. The authors may test or combine more strategies, such as PE4/PE5, twin-PE, or induce mutations on RT or PBS, to further improve the editing efficiency.

R2-0

We thank the reviewer for summarizing the set of improvements (N- and alternatively C-terminal fusion of RT, codon optimization, enhanced nuclear localization via superNLS), which we introduced into the PE architecture (ePE). ePE can improve performance for the 'PE2' editing strategy and can also be directly used for the 'Exo-PE' strategy (via PP7 installed on 3'-stabilized pegRNA).

Please see our detailed point-by-point responses below on how we have further benchmarked the performance of 'Exo-PE' on several endogenous loci, showing substantial improvements in the editing efficacy of large insertions (30 and 54 bp) without compromising editing precision and without the need for a second nick.

Major issues:

1. While the authors claim that these strategies improve prime editing efficacies in their well-designed eTLR system, and even "Exo-PE" achieve editing efficacy comparable to PE3 with fewer undesired indels, it's lack of more data that whether ePE or Exo-PE is used as a general effective improved strategy. More endogenous loci need to be tested in more other cell types, such as in previous study HEK3, DNMT1, VEGFA, RUNX1 or others in HEK293, HeLa, U2OS or other cells.

R2-1

We thank the Reviewer for acknowledging the useful features of the eTLR reporter system.

To prove the generalizability of the improved PE architectures, we have now tested the 'Exo-PE' editing strategy based on our enhanced Prime Editor (ePE) for different insertion lengths (30/54

Communication with editor → NMETH-BC47950C_communication_with_editor

bp) on all of the endogenous loci that the Reviewer suggested (DNMT1, RNF2, VEGFA, HEK3, RUNX1) in HEK293T and HeLa cells.

We have further targeted 9 additional endogenous loci (ENO1-C, ENO1-N, AHSA1, LDHB, PARP1, PKM, VIM, MIF, and GAPDH) in an application for fluorescent tagging of target proteins. we found that the Exo-PE strategy based on ePE shows an overall better editing efficacy without a loss in editing precision than 'PE2' (ANOVA analyses in the main text and new **Figures 3, Supplementary Figure 7, and Supplementary Figure 8.**

2. Although the authors mention many disadvantages of PE3, such as increased indels, and limitations of PE3 in non-mammalian systems, the improved strategy is still mostly based on PE3. In Fig. 2d-e, ePE increases editing efficiency based on PE2, while significantly increase in combination with PE3. Thus, the significantly improvement is mainly attributed to PE3.

3. In Fig.3, although addition of Cas9 mutation K918N or fusion of histone-like domains C-Sso7d E53L or K918N improve PE-efficacy, along with detrimental effects on editing precision (PE/mutEJ), mutEJ-mediated indels events are also increased. And these improvements are also based on PE3. Thus, the authors need to clarify whether ePE are a supplement to PE3 or an alternative strategy.

R2-2 and R2-3

We apologize if our terminology was not clear.

Somewhat confusingly, the original manuscript refers to PE2 as both the protein complex and the editing strategy (using a single pegRNA) as opposed to 'PE3', which is an editing strategy based on the identical PE2 protein but with a second sgRNA.

We thus tried to consistently refer to the editing **strategy** with single quote marks, as in 'PE2', while we refer to the **prime editor** protein as PE2.

Consequently, ePE denotes our enhanced prime editor (harboring a superNLS as well as fusions of RT to the C- or N-terminus, ePE-N and ePE-C), which can be used in 'PE2', 'PE3', or 'Exo-PE'.

We showed improvements using ePE for both the 'PE2' strategy and the 'PE3' strategy in the original **Figure 5e** (now **Figure 2b**).

We have now furthermore expanded on the characterization of ePE and compared it with the old PE2 editor and the latest PEmax editor from the same lab, which also comprises an improved NLS and codons and mutations that supposedly increase Cas9 nicking efficacy (R221K+N394K) (**Figure 1b**). The Sso7d fusion and K918N mutation did yield consistent benefits for eTLR but not for other loci, such that we used ePE-N and ePE-C without those additional modifications for validation on endogenous targets.

Communication with editor → NMETH-BC47950C_communication_with_editor

For 'Exo-PE' we have re-confirmed that the improvements shown in eTLR system (reconfigured **Supplementary Figure 3**) carried over for both the C- and N-terminal fusion variants not only in HEK293T cells (new **Figure 3**, new **Supplementary Figures 5, 8**) but also in HeLa cells (new **Supplementary Figure 7**) on multiple previously published targets.

We consider it an advantage of Exo-PE that it works without a second nick but can still show superior editing efficiency to PE2, approaching that of PE3 without losing editing precision.

4. The authors describe an aptamer-tagged pegRNA to improve editing efficacy, which is introduction of PP7 aptamer into the tetraloop junction between the crRNA and the tracrRNA, by recruiting overexpressed PCP:superNLS:5'-Exo to alleviate the incorporation of the de novo synthesized flap. In Fig. 4b-c, PP7/Pv3-tagged pegRNA and aptamer-mediated recruitment indeed provide a 7-fold increase in efficacy and few indel compared to PE2, but still slightly lower than PE3 efficacy along with a higher editing precision. However, whether these improvements have a big impact on actual genome editing, authors need to test endogenous sites to clarify this.

R2-4

We have now re-confirmed that we used the optimal configuration of the ePE editor also for its application with the Exo-PE strategy, which we then validated on all of the endogenous loci that you suggested (see our response **R2-1**, new **Figure 3**) and 9 additional loci for fluorescent tagging of proteins (new **Supplementary Figure 8**).

5. In Fig.5e, combining with a 3'-stabilization motif, Exo-PE (ePE/K918N/7v3/Q1) significantly a higher editing precision compared to original PE3, in this reporter. I recommend that authors compare the Exo-PE with PE4 or twin-PE, which are current alternative strategies for PE3.

R2-5

In the original Fig. 5e (now Figure 2b, pasted below), 'Exo-PE' based on ePE shows an improved editing efficacy on eTLR for the 'PE2' strategy but not the 'PE3' strategy. ePE also showed increased editing efficacy over the editor used by Chen *et al.* called PEmax for the 'PE3' strategy.

Communication with editor → NMETH-BC47950C_communication_with_editor

We have now also compared the performance of ePE for different strategies that necessitate a second sgRNA or pegRNA, including twin-PE, which showed equivalent editing efficacy to 'PE3'.

Figure R1 | Comparison of the 'PE3' vs. 'PPE' vs. 'twin-PE' strategies using ePE-N.
 Bars represent the mean \pm s.d (n = 3, biological replicates).

However, our main objective was to generate an alternative strategy to PE2 with improved editing efficacy but no need for a second nick via an additional sgRNA (PE3) or pegRNA (PPE). Therefore, we chose PE2 as the relevant benchmark for the Exo-PE strategy, which allowed us to use identical pegRNAs taken from reference publications, thus avoiding any confounds from the effect of the second sgRNA or pegRNA, whose optimization has a different set of free parameters.

With respect to MMR inhibition (PE4): whereas recruitment of MHL1dn can help protect short mismatch intermediates from MMR, it has already been shown by Chen et al. (doi:10.1016/j.cell.2021.09.018) that the introduction of additional (silent) mutations is as effective as MLH1dn in evading MMR. Those are present in our long edits (30 and 54 bp).

Minor points:

Communication with editor → NMETH-BC47950C_communication_with_editor

1. The eTLR system is indeed well-designed and simple reporter to screen and compare editing efficacy and indels of different PE versions. There should be a parallel comparison of all three HDR/PE, mutEJ, and the ratio between HDR/PE and mutEJ values. For example, in Fig. 1c, it's obviously not true that two samples with no HDR donor or Cas9/sgRNA has a high HDR/mutEJ, which is due to weak HDR or mutEJ value. To better use it, I recommend that the authors could set evaluation criteria or baseline.

R2-6

We thank reviewer R2 for this positive assessment of the eTLR system. The threshold in our FACS analysis was set in an unbiased fashion such that about 0.1% false positive green was the maximum in the non-targeting control (ntc). We have added a corresponding note in the respective figure legend (now **Supplementary Fig. 1d**) that the HDR/mutEJ ratios are not informative for the controls due to the naturally low number of FACS events.

2. In this study, based on original, the authors created Exo-PE or ePE, and further various version. The authors need to rename and make a more clarity naming.

R2-7

As mentioned in our Response R2-2 above, we apologize for any confusion that the naming scheme may have generated.

We tried to consistently refer to 'Exo-PE' as an alternative editing **strategy** to 'PE2' (without additional nick) or 'PE3' (with additional nick).

In distinction, the enhanced Prime Editor (ePE) is the **core editor** itself, which is compatible with all different editing strategies.

3. In Fig. 4c, Pv3 should be described. What's different to PP7v3?

R2-8

We are very sorry if the nomenclature led to any confusion. We have adjusted the figure labels.

--

We thank the Reviewer again for the careful and helpful comments guiding us to improve the characterization of Exo-PE, which we believe adds useful capabilities to the PE technology.

Communication with editor → NMETH-BC47950C_communication_with_editor

Reviewer #3:

Remarks to the Author:

In this study, Truong et al present multiple improvements of the prime editor using rational design. To systematically improve the prime editing architecture, they firstly developed an enhanced traffic light reporter (eTLR) system, which can report all editing events. Using the eTLR system to measure editing efficiency, they have evaluated a series of optimizations on the PE architecture, including the rearrangement of the RT to the N-terminal of the Cas9 nickase

(RT-nCas9), a new codon-optimization, improving the nuclear localization using superNLS, introducing Cas9-enhancing mutations (ePEK918N) and histone-like domains (ePEsso7d), and recruiting a 5'-DNA-exonuclease. They demonstrate that their optimization strategies could be combined with an engineered pegRNA to further improve PE editing efficiencies. However, the authors need to incorporate additional data that is critical for acceptance of this manuscript, especially the evaluation of their technology at endogenous genomic target sites using their optimized PE. I recommend accepting the manuscript only if the following concerns can be addressed in a revised manuscript.

R3-0

We thank the reviewer for the careful enumeration of the set of improvements we have introduced to the PE editing architecture.

Major concerns:

1. The authors need to evaluate the editing efficiency of their newly designed tools, ePEK918N-Sso7d and Exo-PE, at endogenous targets. This is a critical data point for all newly developed genome editing tools as many times the improvements in editing when evaluated using a reporter system do not translate to improvements in editing at endogenous sites. I recommend the authors compare editing efficiencies between their optimized prime editors and previous generations of prime editors across at least 10 target sites in at least two different cell types.

R3-1

We have now validated ePE with both the PE2 and the new Exo-PE strategy on the endogenous loci DNMT1, RNF2, VEGFA, HEK3, RUNX1 (new **Figure 3** and new **Supplementary Figure 5**) as well as ENO1-N, ENO1-C, AHSA1, LDHB, PARP1, PKM, VIM, MIF, and GAPDH (new **Supplementary Figure 8**). We also validated the data from HEK cells in HeLa cells (new **Supplementary Figure 7**).

We find an overall better editing efficacy of the 'Exo-PE' strategy in direct comparison to the 'PE2' strategy without compromise on editing precision, using identical pegRNAs (ANOVA results in the main text, new **Figure 3**, **Supplementary Figure 7**). We also provide data that the

Communication with editor → NMETH-BC47950C_communication_with_editor

fusion type (N- or C-terminal fusion of RT to nCas9 can have a relevant effect for a given locus of interest. We provide all statistical results in **Supplementary Table 2**).

2. Considering the improvements in on-target editing efficiency of the newly optimized prime editors described in this manuscript, the authors should also evaluate the off-target effects of these improvements. Endogenous off-target effects of the optimized prime editor should be assessed across at least 10 endogenous off-target targets with less than 3 mismatches between the on-target and off-target pegRNAs.

R3-2

We have carefully assessed the editing errors from ePE in the 'Exo-PE' and 'PE2' approaches (new **Supplementary Figure 6**).

We have also assessed the off-target effects for the improved ePE complex for both 'Exo-PE' and 'PE2' in both the N- and C-terminal fusion types (ePE-N, ePE-C) on the well-characterized off-targets (new **Supplementary Table 1**).

We included OT1 of HEK3, which provides a perfect match for the PBS of the reference pegRNA previously used in the literature.

We found that Exo-PE showed increased editing activity on that site compared to PE2. Although it should not be surprising that this excellent target gets edited, we hypothesize in the discussion section that the competition between the flaps in PE2 may also contribute to an apparent specificity in a locus-dependent fashion, which is now partially removed by the activity of the exonuclease. As an obvious conclusion, pegRNA design should avoid matches with other sites to avoid "off"-target effects for the increasingly more potent prime editors.

2. Can the authors explain why co-expressing the pegRNA with the PCP:superNLS:5'-Exo is more efficient than PE2 in Fig. 4b? The authors should include a more detailed description in the revised manuscript.

R3-2b

Thank you for this detailed observation regarding the original Fig. 4b (now moved to new **Supplementary Figure 3a**). We hypothesize that the co-expression of the non-recruited PCP-exonuclease can also address the 5' overhang, albeit less effectively. This is not surprising since recent publications also showed that even the reverse transcriptase of the PE enzyme does not have to be fused nor actively recruited to the nCas9-pegRNA complex (doi:10.1038/s41587-022-01255-9, doi:10.1038/s41587-022-01473-1) for a functional PE machinery.

3. Can the authors describe why the CAG-driven pegRNAs show lower editing efficiencies compared to the U6-driven pegRNAs with a CG pair in the pegRNA scaffold in Extended Data Fig. 4?

Communication with editor → NMETH-BC47950C_communication_with_editor

R3-3

The plasmid size of pCAG-driven pegRNA plasmid (5.2 kbp) is bigger than the pU6-driven pegRNA plasmids (3.5 kbp). Since we used equal masses of plasmids during the transfection, a lower molarity of plasmids was transfected. Since this information does not hold any value for the revised manuscript, we removed the respective figure.

4. I suggest the authors introduce the function of and thoroughly describe Sso7d in the context of "we next hypothesized that PE efficacy might benefit from an increased local concentration at nucleosomes without increasing the general expression strength. Therefore, we fused the histone-like 7d protein from the archaea *Sulfolobus solfataricus* (Sso7d with E35L to inactivate cryptic ds/ssRNase activity)²⁸ to the C-terminus of ePE."

R3-4

Sso7d, as a DNA-binding enhancer, is used by many polymerase vendors, e.g., the Q5 or Phusion polymerase from New England Biolabs. Sso7d was also shown in several publications to enhance general DNA-associated enzymatic activities, such as polymerases, ligases, and integrases, without having detrimental effects on off-target or precision (doi: 10.1093/nar/gkh271, 10.1093/protein/gzt024, 10.1093/protein/gzt024). We introduced the mutation E35L to deactivate the cryptic RNase activity of Sso7d, which is expected to be detrimental to the pegRNA.

Minor concerns:

1. Please clarify "Pv3" in figure 4 legends.

We are sorry about this inconsistency. Pv3 is the same as PP7v3 and will be corrected in the revision.

Once again, we would like to thank the Reviewer for the valuable feedback, which greatly helped us to characterize the performance of Exo-PE. We believe that Exo-PE is a valuable addition to the PE toolbox that should help foster further improvements in gene editing technology.

Decision Letter, first revision:

Dear Gil

Your Brief Communication, "Exonuclease-enhanced prime editors", has now been seen by 3 reviewers. As you will see from their comments below, although the reviewers find your work of considerable potential interest, they have raised a concerns about requirements for appropriate benchmarking. We are interested in the possibility of publishing your paper in Nature Methods, but would like to consider your response to these concerns before we reach a final decision on publication.

We therefore invite you to revise your manuscript to address these concerns. As you will see all three refs have similar recommendations which include testing more sites for PE3 and comparisons against PE4/5.

[Redacted] This URL links to your confidential home page and associated information about manuscripts you may have submitted, or that you are reviewing for us. If you wish to forward this email to co-authors, please delete the link to your homepage.

We hope to receive your revised paper within 8 weeks. If you cannot send it within this time, please let us know. In this event, we will still be happy to reconsider your paper at a later date so long as nothing similar has been accepted for publication at Nature Methods or published elsewhere.

Sincerely,
Madhura

Madhura Mukhopadhyay, PhD
Senior Editor
Nature Methods

Reviewers' Comments:

Reviewer #1:

Remarks to the Author:

Prime editing (PE) is a powerful CRISPR-Cas9 based precise genome editing technique that has gone since its inception through a plethora of refinement due to low genome editing efficiency. In this resubmitted manuscript, Truong et al explored several approaches of improving prime editing efficiency including refining fusion strategies of different effectors to PE machinery and optimizing pegRNA stability. One major improvement over the first version of the manuscript is the assessment of prime editors also in two different mammalian cell lines (HeLa and HEK293T) at several endogenous target sites and validating results with NGS aside from solely testing optimized prime editors using authors' improved traffic light reporter. While this manuscript takes into consideration several previously published approaches of improving PE, it also contributes novelty to the field with the exonuclease usage.

While the authors tested Exo-PE at several endogenous target sites (with different RT fusion in regards to Cas9n), they compare it solely to PE2 (with different RT fusion in regards to Cas9n). In the manuscript, it is explained why the PE3 strategy was not tested, however, I believe it would be appropriate to also compare it to PE4 (PE2 + MLH1dn) strategy previously published (Chen et al, 2021). PE4 on average improves PE efficiency approximately 2-fold over PE2 (Chen et al, 2021) bringing increase in PE editing with Exo-PE on par with "predicted" PE4 activity at several endogenous target sites tested. Therefore, to improve Exo-PE editing efficiency above previously published iterations of PE technique, it would be

prudent to test Exo-PE4 (Exo-PE + MLH1dn) performance. Based on the current data, the Exo-PE system is mostly on par with PE3 for editing efficiency. Without comparison with the best PE systems that are beyond PE3, I am not sure this technology represents a breakthrough for PE technology.

The authors have expanded the methods section significantly; however, I cannot find any information about depositing newly developed PE systems (at least Exo-PE –N, -C) to depository. Furthermore, plasmid maps are still missing as well as list of vectors and primers used in this study.

I appreciate the authors conducted additional experiments on endogenous target sites at two different cell lines. Overall, this work addresses one aspect of prime editing by more efficiently removing the 5' flap using exonuclease, which is novel. Whether Exo-PE represents a better PE system that others will use? Well, I am not sure. I will leave it to the editor to decide whether such novelty merits publishing at Nature Methods.

Reviewer #2:

Remarks to the Author:

Remarks to the Author:

Truong et al. present a revised manuscript that addresses many of the original comments, producing data to support characterization of the new prime editing strategy EXO-PE at some endogenous sites in different cell types. For testing application of this new strategy, they targeted several endogenous loci by inserting a short split fluorescence peptide to enable biomolecular fluorescence complementation. However, some concerns still remain.

Major concerns

1. As a newly developed tool, EXO-PE should be compared with some of the well-characterized prime editors, at least for the case of PE4/PE5 which is mentioned by different reviewers, in different endogenous loci (at least 10 loci) by targeted deep sequence but not in the report system. Though the two strategies are involved in different pathways, the comparative information will be useful for application of EXO-PE, and crucial to further improvements of PE system. In addition, it is interesting to know whether combining EXO-PE with PE4/PE5 could further improve the efficacy of PE system.
2. Better representation of the technology is needed. The authors stated that “v3 showed a trend towards improved efficacy while maintaining improved editing precision in ‘Exo-PE’ (Supplementary Fig. 3b)”, which should be confirmed at more sites, this is vital for further improvement and utilization of EXO-PE. Crucial characterization of EXO-PE was too less shown in the main figures. We suggest to depict

the architecture of ePE and the design of variants for the PP7-aptamer-tagged pegRNA in Fig. 2, it would be easier and faster to get the key info for the readers.

3. Key components (i.e., Sso7d) and key mutations (i.e., N863A, K918N) of EXO-PE were only introduced without explanations or proper references in the context. For example, why N863A is used in some constructs, but H840A in others. Sso7d and K918N should be clearly described and referenced in the main text when mentioned at the very first time. The mutations of MLV RT hexamutant (x6) in Fig. S2a need to be specified, please note that a pentamutant RT is used in PE2/PE3/PEmax (PMID: 31634902).

Minor concerns:

1. We would suggest making a more clarity naming. Some names in the text could be mistaken for published tools, such as PPE (plant prime editor, PMID: 33767395), ePE (enhanced prime editing system, PMID: 34103663).
2. Please clarify “Cryptic splice donor sites” in Fig. S2a.
3. “HEK293 eTLR cells (ECACC: 12022001, Sigma-Aldrich)”, make sure the corresponding catalog number is right?

Reviewer #3:

Remarks to the Author:

Truong et al. demonstrated that ‘Exo-PE’ showed overall better editing efficacy than the ‘PE2’ strategy for ≥ 30 bp insertions without compromising editing precision. Compared to their previously submitted manuscript, the authors performed additional comparisons between Exo-PE and PE2 to demonstrate the superior editing efficiency of Exo-PE when editing endogenous sites. While PE3 typically edits more efficiently than PE2 and PE3 tends to generate more indels as well; it is interesting that the authors found Exo-PE to show a better balance between editing efficiency and accuracy. I recommend the authors perform additional head-to-head comparisons between Exo-PE and PE3 at many endogenous sites and then explore calculating an efficiency/accuracy ratio. I believe that this data is essential to demonstrate a superior therapeutic potential for Exo-PE.

Moreover, previous reports have demonstrated that the repair mechanism between point mutations and indels may be different and that these differences may serve as insightful strategies to further improve the editing of prime editors (Chen et al. Cell. 2021). (For example, they found that “the MLH1dn enhancement of prime editing efficiency declined as the length of the indels increased, consistent with previous reports that MMR repairs IDLs up to 13 nt in length”). Thus, since the authors only tested two

types of fragment insertions when comparing Exo-PE and PE2, the authors should perform additional comparisons for single base substitutions across a number of endogenous sites.

Minor comments:

1. In Sup Fig. 2e, it's interesting that the MarathonRT showed a high editing efficiency in their eTLR reporter system. I suggest the authors compare this RT to the Moloney Murine Leukemia Virus RT in both a reporter system and across many endogenous sites.
2. In Sup Fig. 3e, what does PP7v3/e mean?
3. In Sup Fig. 3e, the lines for P value calculation are misaligned. Please modify.

Author Rebuttal, first revision:

Letter to Referees: Exonuclease-enhanced prime editors**Reviewers' Comments:****General remarks to all Reviewers**

We like to thank all Reviewers for their constructive and detailed feedback, which helped us to improve the manuscript substantially. In the process, we have refocused the manuscript on the new prime editing strategy 'Exo-PE', based on recruiting a 5'-3' exonuclease to and enhanced Prime Editor (ePE) protein to enzymatically generate space for incorporating the reverse transcribed 3' flap.

In short, we have now re-confirmed the improvements combined in ePE and then validated the improved performance of 'Exo-PE' on 5 loci as requested (DNMT1, RNF2, VEGFA, HEK3, RUNX1) with 2 different challenging edits (30 and 54 bp insertions) in 2 different cell types.

In addition, we targeted 9 additional endogenous loci in which we inserted the DNA coding for 11 amino acids of a split fluorescence protein to enable biomolecular fluorescence complementation as a biotechnological application in analogy to this prominent research project (<https://opencell.czbiohub.org/about>).

Please find our detailed point-by-point responses below.

Reviewer #1:

Remarks to the Author:

Prime editing (PE) is the latest of CRISPR/Cas based precise genome editing technologies. It has attracted a considerable attention of various research communities due to a promise of theoretically unrestricted genome editing in all three Woese's domains of life. In order to improve PE, Truong et al tested multiple ideas, including novel ones such as fusion of RNaseH and 5' to 3' exonucleases. The authors started off by designing and testing a nice reporter system which presumably can report nearly all possible genome editing outcomes. Then, they attached an exonuclease to the prime editor to remove the 5' unedited flap resulting (with other improvements) in "Exo-PE" and investigated the N terminal fusion of reverse transcriptase (RT) to Cas9 nickase (with addition of RNaseH fusion protein and optimized NLS) resulting in "ePE". The authors also incorporated previously published beneficial mutations into Cas9 nickase of prime editor, increased local concentration of PE at the target site with histone like proteins as well as investigated several enhanced pegRNAs. Overall, the developed reporter system facilitated rapid screening and comparison of different PE systems. This manuscript builds upon several recent research articles demonstrating improved prime editing. However, this reviewer has major concerns about this study and find the results preliminary (and the study as incomplete).

R1-0

We thank the reviewer for recognizing the powerful reporter system for gene editing events we developed and the set of substantial improvements on the prime editing architecture we achieved. Please see our point-by-point responses on how we now validated the improved PE systems in detail in line with your suggestions.

Major concerns:

The most significant concern that I have for this work is that the whole study is based on one single target site (in the reporter gene) with one pegRNA design (by inserting a ~50 nt sequence) for the same edit in a single cell type (HEK293T). It is well known that PE efficiencies vary greatly depending on target sites and edits. Different PE systems may favor certain editing outcomes than others. It is nice that the authors can use a streamlined reporter system to rapidly identify some promising PE systems. However, after such a screen, they must benchmark the PE systems by testing many endogenous target sites with different editing outcomes (deletions, insertions, and nucleotide changes) in multiple human cell lines. Without seeing such demonstrations and validations, this reviewer remains unconvinced about the improved PE systems. Along this line, it is surprising to see that PPE (paired prime editor) performed poorly in this study. The authors only cited on PPE paper.

However, such PPE systems have been reported in at least four peer-reviewed publications 1-4, and they all claimed higher editing efficiencies than PE2 and PE3. It is likely the PPE design/system that the authors used for the only target site and only pegRNA is not optimal or very poor (and hence not a good representative for PPE). Again, this just suggests the importance of comprehensive testing at multiple sites with different edits in order to compare and benchmark the PE systems.

R1-1

We have now assessed the improved PE architectures for different endogenous loci, such as DNMT1, RNF2, VEGFA, HEK3, RUNX1, ENO1-C, ENO1-N, AHSA1, LDHB, PARP1, PKM, VIM, MIF, and GAPDH. We find that the Exo-PE strategy has an overall better editing effect in direct comparison to the PE2 strategy with identical pegRNAs (statistical results in the main text, new **Fig. 3**, new **Supplementary Fig. 8**).

We also provide data showing that the type of fusion (N- or C-terminal fusion of RT to nCas9) may have a relevant effect for a given site of interest. In addition, all statistical results are provided in new **Supplementary Table 2**.

With respect to the PPE, we have now compared ePE directly for editing strategies that necessitate a second pegRNA (or sgRNA), including classical PPE and twin-PE (which, in contrast to PPE, does not have 'homology regions') and found a performance on par with PE3 on eTLR but with a decreased editing precision.

Figure R1 | Comparison of the 'PE3' vs. 'PPE' vs. 'twin-PE' strategies using the enhanced prime editor complex ePE-N. Bars represent the mean \pm s.d. (n = 3, biological replicates).

As Reviewer 1 mentioned, there have been a lot of paired-pegRNA PE (PPE) strategies (Twin-PE, GRAND, HOPE, etc., <https://doi.org/10.1038/s41576-022-00541-1>) and the performance also depends on careful optimizations of the pegRNAs for a specific locus. It is thus conceivable that the performance of those paired-pegRNA strategies can also be further optimized for the long insertion into eTLR.

Still, our goal was to search for a method to improve the 'PE2 strategy' without the necessity for a secondary nicking sgRNA ('PE3 strategy') or a second pegRNA (PPE).

Thus, we have validated Exo-PE directly against PE2 on all endogenous loci with identical reference pegRNAs taken from previous publications to eliminate any confounds from free design parameters and optimization rounds for the second sgRNA or pegRNA.

We also benchmarked against 'PE3' when we applied 'Exo-PE' for fluorescent tagging of target proteins (new **Supplementary Fig. 8**). In this application, 'ExoPE' again outperformed 'PE2', and for one locus, showed similar efficacy as 'PE3'.

Second, the authors developed a comprehensive FACS screening assay reducing the amount of time and work needed to attain the preliminary results. However, the most important results of the manuscript (comparing improved prime editors with previously published prime editors) should be validated with NGS. When the authors start more testing on more target sites, they should also use NGS to precisely verify and quantify the editing outcomes.

R1-2

As stated in our response to your point **R1-1**, we have now included NGS results for different endogenous loci (4 of 5 different endogenous loci showed better efficacy with Exo-PE compared to PE2) in HEK293T and HeLa cells (new **Figure 3**, new **Supplementary Figure 7**), which demonstrates that Exo-PE yields an overall higher editing efficacy and precision than PE2 using identical pegRNAs. NGS was utilized for an in-depth analysis of editing outcomes. In addition,

we used the bimolecular fluorescence complementation system to fluorescently tag target proteins and observed that, in 7 out of 9 loci, Exo-PE was significantly better in terms of editing efficacy (% of green cells, new **Supplementary Figure 8**).

Minor points:

1. The article is written in the style of a full "Article" with 5 main figures. I don't know why it is called "Brief Communication".

R1-3

Based on the Reviewers' feedback, we have now streamlined the manuscript focusing on Exo-PE and have reduced the number of main Figures to 3 (with 8 Supplementary Figures and 2 Supplementary Tables) in line with a Brief Communication.

2. The authors did not share a lot of information about the vectors used in this study. I recommend adding vector maps, list of vectors and primers used in this study, as well as nucleotide sequence of newly developed reporter used in FACS assay in Supplemental file. Additionally, relevant vectors could be deposited to plasmid depository like Addgene.

R1-4

We will submit all of the constructs to Addgene, including detailed meta-information on constructing the FACS reporter line.

3. I suggest the authors compare MLH1dn recruitment strategy (Chen et al, 2021) with their strategy recruiting exonuclease since both strategies aim to solve problem from a similar vantage point.

R1-5

The Exo-PE strategy focuses on avoiding the flap competition by actively degrading the original 5' flap to make space for the RT'ed *de novo* 3' flap. We validated this strategy specifically for longer insertions because, in this capability, PE differentiates itself from base editors and promises advantages over DBS-triggered HDR-dependent editing techniques. Whereas transient MHL1dn co-expression can help to protect short mismatch intermediates from MMR, it was already shown in Chen *et al.* (<https://doi.org/10.5281/zenodo.5551032>) that the introduction of additional (silent) mutations - as present in our long edits (30/ 54 bp) - are just as effective as MLH1dn in evading MMR.

4. Does the scale bar in Fig 1b represent the same size on brightfield insets of fluorescent microscopy photos?

R1-6

Yes, the insets (now **Supplementary Figure 1**) are the same segment as shown in the fluorescence channel.

5. It would be informative from spatial perspective if scalebar (in nm or angstroms) would be added to Fig 2c.

R1-7

Thank you for the suggestion. We have done so. The figure is now **Supplementary Fig. 2b**.

6. The architecture of ePE is depicted in Fig 2A. It would be faster to find the ePE architecture if it was labeled in Fig 2A, similar as it is done in Fig 5D.

R1-8

We introduced the suggested labels into the subfigure (now **Supplementary Fig. 2a**).

7. In the last paragraphs before the section "Cas9-enhancing mutations and histone-like proteins improve primer editing efficacy" please define the "enzyme" in "... higher editing efficacy against the original enzyme..."

R1-9

The enzyme referred to in the respective text and figure (now **Supplementary Fig. 2f**) was ePE-N which showed superior performance when compared to the original PE2 enzyme. Please note that in the original paper, PE2 sometimes refers to the enzyme, *i.e.*, the nCas9-RT fusion, and sometimes to 'PE2 strategy' (so prime editing with only the primary pegRNA without secondary nicking sgRNAs). In Supplementary Fig. 2f, we compare the ePE-N enzyme with the original PE2 enzyme in 'PE2' and 'PE3' settings.

8. When K918N mutation is introduced into text, it is not mentioned what is the biological reason introducing this mutation nor it is referenced. The same mutation is referenced later in manuscript.

R1-10

This mutation in Cas9 was describe in Hand *et al.* (2021) (doi: 10.1089/crispr.2020.0092) to increase Cas9's nuclease activity. We thus assessed whether it also enhances the nickase activity of Cas9 H840A or N863A. However, we found that its beneficial effects were true for the eTLR as target but - in the context of the several other improvements in ePE, did not provide additional advantages on other loci. We thus did not move forward with those mutations for the final ePE complexes, which we validated on endogenous loci as described in the text.

9. Figure 5A is quite crowded.

R1-8

We have changed the design of the figures to improve readability.

10. Supplemental Figure 7A: I couldn't find any explanation why T5-5'-Exo is N terminally fused to PCP and other exonucleases (FEN1, Exo1) C terminally fused.

R1-9

As shown in Figure R2 below, we did not see any major difference between N- or C-terminally fused PCP to recruit T5 5'exonuclease. For FEN1 and EXO1, we used both exonucleases as N-terminal domains since their native interaction domains (deleted in our chimeras) are located at their respective C-termini. Thus, we also placed PCP as a C-terminal domain. PCP has been used in the literature in both orientations successfully. Indeed, both directional fusions with T5 5'-exonuclease were active, as shown in Figure R2.

Figure R2 | Comparison of the site of PCP fusion for recruiting the T5 5'exonuclease to the PE complex.

11. In subsection "Synergy between enhanced prime editing enzymes and enhanced pegRNAs" there is a contrast between described results in text and data presented in Fig 5B about EvoPreQ1 pseudoknot.

R1-10

We apologize if the statements in the original text related to Figure 5b (now **Supplementary Figure 3f**) were not in line with the header. The figure shows that 3' stabilization of the pegRNA is essential for editing performance.

—

We would like to thank the Reviewer again for the detailed and productive comments, which greatly helped us to validate the performance of Exo-PE and improve the manuscript.

We believe that Exo-PE is a potent addition to the PE toolbox that should be of interest to other research teams working on improving gene editing.

Reviewer #2: Remarks to the Author: Comments:

Truong et al. report an enhanced traffic light reporter (eTLR) to differentiate all possible NHEJ/MMEJ-mediated insertions or deletions (indels) events from a successful HDR/primer editing (PE)- mediated events. With this reporter system, authors describe several optimizations to improve the prime editing. Firstly, they optimize the protein architecture of PE to create enhanced PE (ePE), by N-terminal fusion of RT to nCas9, optimization of RNA codon usage, supply NLS, introduction of enhancing mutation K918N, and addition of histone-like domain (Sso7d). Secondly, they introduce an aptamer into pegRNA, which recruits a 5'-DNA-exonuclease to the primary nick to degrade the 5'-flap actively without secondary nicks. The authors demonstrate that these strategies increase prime editing efficiencies and the combination ePE with engineered pegRNA further facilitate efficacy in cell line with eTLR system. Although it is a useful reporter system to optimize PE, this study is a relatively narrow or weak improvement strategy. The authors may test or combine more strategies, such as PE4/PE5, twin-PE, or induce mutations on RT or PBS, to further improve the editing efficiency.

R2-0

We thank the reviewer for summarizing the set of improvements (N- and alternatively C-terminal fusion of RT, codon optimization, enhanced nuclear localization via superNLS), which we introduced into the PE architecture (ePE). ePE can improve performance for the 'PE2' editing strategy and can also be directly used for the 'Exo-PE' strategy (via PP7 installed on 3'-stabilized pegRNA).

Please see our detailed point-by-point responses below on how we have further benchmarked the performance of 'Exo-PE' on several endogenous loci, showing substantial improvements in the editing efficacy of large insertions (30 and 54 bp) without compromising editing precision and without the need for a second nick.

Major issues:

1. While the authors claim that these strategies improve prime editing efficacies in their well-designed eTLR system, and even "Exo-PE" achieve editing efficacy comparable to PE3 with fewer undesired indels, it's lack of more data that whether ePE or Exo-PE is used as a general effective improved strategy. More endogenous loci need to be tested in more other cell types, such as in previous study HEK3, DNMT1, VEGFA, RUNX1 or others in HEK293, HeLa, U2OS or other cells.

R2-1

We thank the Reviewer for acknowledging the useful features of the eTLR reporter system.

To prove the generalizability of the improved PE architectures, we have now tested the 'Exo-PE' editing strategy based on our enhanced Prime Editor (ePE) for different insertion lengths (30/54

bp) on all of the endogenous loci that the Reviewer suggested (DNMT1, RNF2, VEGFA, HEK3, RUNX1) in HEK293T and HeLa cells.

We have further targeted 9 additional endogenous loci (ENO1-C, ENO1-N, AHS1, LDHB, PARP1, PKM, VIM, MIF, and GAPDH) in an application for fluorescent tagging of target proteins. We found that the Exo-PE strategy based on ePE shows an overall better editing efficacy without a loss in editing precision than 'PE2' (ANOVA analyses in the main text and new **Figures 3, Supplementary Figure 7, and Supplementary Figure 8**).

2. Although the authors mention many disadvantages of PE3, such as increased indels, and limitations of PE3 in non-mammalian systems, the improved strategy is still mostly based on PE3. In Fig. 2d-e, ePE increases editing efficiency based on PE2, while significantly increase in combination with PE3. Thus, the significantly improvement is mainly attributed to PE3.

3. In Fig.3, although addition of Cas9 mutation K918N or fusion of histone-like domains C-Sso7d E53L or K918N improve PE-efficacy, along with detrimental effects on editing precision (PE/mutEJ), mutEJ-mediated indels events are also increased. And these improvements are also based on PE3. Thus, the authors need to clarify whether ePE are a supplement to PE3 or an alternative strategy.

R2-2 and R2-3

We apologize if our terminology was not clear.

Somewhat confusingly, the original manuscript refers to PE2 as both the protein complex and the editing strategy (using a single pegRNA) as opposed to 'PE3', which is an editing strategy based on the identical PE2 protein but with a second sgRNA.

We thus tried to consistently refer to the editing **strategy** with single quote marks, as in 'PE2', while we refer to the **prime editor** protein as PE2.

Consequently, ePE denotes our enhanced prime editor (harboring a superNLS as well as fusions of RT to the C- or N-terminus, ePE-N and ePE-C), which can be used in 'PE2', 'PE3', or 'Exo-PE'.

We showed improvements using ePE for both the 'PE2' strategy and the 'PE3' strategy in the original **Figure 5e** (now **Figure 2b**).

We have now furthermore expanded on the characterization of ePE and compared it with the old PE2 editor and the latest PEmax editor from the same lab, which also comprises an improved NLS and codons and mutations that supposedly increase Cas9 nicking efficacy (R221K+N394K) (**Figure 1b**). The Sso7d fusion and K918N mutation did yield consistent benefits for eTLR but not for other loci, such that we used ePE-N and ePE-C without those additional modifications for validation on endogenous targets.

For 'Exo-PE' we have re-confirmed that the improvements shown in eTLR system (reconfigured **Supplementary Figure 3**) carried over for both the C- and N-terminal fusion variants not only in HEK293T cells (new **Figure 3**, new **Supplementary Figures 5, 8**) but also in HeLa cells (new **Supplementary Figure 7**) on multiple previously published targets.

We consider it an advantage of Exo-PE that it works without a second nick but can still show superior editing efficiency to PE2, approaching that of PE3 without losing editing precision.

4. The authors describe an aptamer-tagged pegRNA to improve editing efficacy, which is introduction of PP7 aptamer into the tetraloop junction between the crRNA and the tracrRNA, by recruiting overexpressed PCP:superNLS:5'-Exo to alleviate the incorporation of the de novo synthesized flap. In Fig. 4b-c, PP7/Pv3-tagged pegRNA and aptamer-mediated recruitment indeed provide a 7-fold increase in efficacy and few indel compared to PE2, but still slightly lower than PE3 efficacy along with a higher editing precision. However, whether these improvements have a big impact on actual genome editing, authors need to test endogenous sites to clarify this.

R2-4

We have now re-confirmed that we used the optimal configuration of the ePE editor also for its application with the Exo-PE strategy, which we then validated on all of the endogenous loci that you suggested (see our response **R2-1**, new **Figure 3**) and 9 additional loci for fluorescent tagging of proteins (new **Supplementary Figure 8**).

5. In Fig.5e, combining with a 3'-stabilization motif, Exo-PE (ePE/K918N/7v3/Q1) significantly a higher editing precision compared to original PE3, in this reporter. I recommend that authors compare the Exo-PE with PE4 or twin-PE, which are current alternative strategies for PE3.

R2-5

In the original Fig. 5e (now Figure 2b, pasted below), 'Exo-PE' based on ePE shows an improved editing efficacy on eTLR for the 'PE2' strategy but not the 'PE3' strategy. ePE also showed increased editing efficacy over the editor used by Chen *et al.* called PEmax for the 'PE3' strategy.

We have now also compared the performance of ePE for different strategies that necessitate a second sgRNA or pgrRNA, including twin-PE, which showed equivalent editing efficacy to 'PE3'.

Figure R1 | Comparison of the 'PE3' vs. 'PPE' vs. 'twin-PE' strategies using ePE-N. Bars represent the mean \pm s.d. (n = 3, biological replicates).

However, our main objective was to generate an alternative strategy to PE2 with improved editing efficacy but no need for a second nick via an additional sgRNA (PE3) or pgrRNA (PPE). Therefore, we chose PE2 as the relevant benchmark for the Exo-PE strategy, which allowed us to use identical pgrRNAs taken from reference publications, thus avoiding any confounds from the effect of the second sgRNA or pgrRNA, whose optimization has a different set of free parameters.

With respect to MMR inhibition (PE4): whereas recruitment of MHL1dn can help protect short mismatch intermediates from MMR, it has already been shown by Chen et al. (doi:10.1016/j.cell.2021.09.018) that the introduction of additional (silent) mutations is as effective as MLH1dn in evading MMR. Those are present in our long edits (30 and 54 bp).

Minor points:

1. The eTLR system is indeed well-designed and simple reporter to screen and compare editing efficacy and indels of different PE versions. There should be a parallel comparison of all three HDR/PE, mutEJ, and the ratio between HDR/PE and mutEJ values. For example, in Fig. 1c, it's obviously not true that two samples with no HDR donor or Cas9/sgRNA has a high HDR/mutEJ, which is due to weak HDR or mutEJ value. To better use it, I recommend that the authors could set evaluation criteria or baseline.

R2-6

We thank reviewer R2 for this positive assessment of the eTLR system. The threshold in our FACS analysis was set in an unbiased fashion such that about 0.1% false positive green was the maximum in the non-targeting control (ntc). We have added a corresponding note in the respective figure legend (now **Supplementary Fig. 1d**) that the HDR/mutEJ ratios are not informative for the controls due to the naturally low number of FACS events.

2. In this study, based on original, the authors created Exo-PE or ePE, and further various version. The authors need to rename and make a more clarity naming.

R2-7

As mentioned in our Response R2-2 above, we apologize for any confusion thins (ePESso7d), and recruiting a 5'-DNA-exonuclease. They demonstrate that their optimization strategies could be combined with an engineered pegRNA to further improve PE editing efficiencies. However, the authors need to incorporate additional data that is critical for acceptance of this manuscript, especially the evaluation of their technology at endogenous genomic target sites using their optimized PE. I recommend accepting the manuscript only if the following concerns can be addressed in a revised manuscript.

3. In Fig. 4c, Pv3 should be described. What's different to PP7v3?

R2-8

We are very sorry if the nomenclature led to any confusion. We have adjusted the figure labels.

R3-0

We thank the reviewer for the careful enumeration of the set of improvements we have introduced to the PE editing architecture.

Major concerns:

1. The authors need to evaluate the editing efficiency of their newly designed tools, ePEK918N-Sso7d and Exo-PE, at endogenous targets. This is a critical data point for all newly developed genome editing tools as many times the improvements in editing when evaluated using a reporter system do not translate to improvements in editing at endogenous sites. I recommend the authors compare editing efficiencies between their optimized prime editors and previous generations of prime editors across at least 10 target sites in at least two different cell types.

R3-1

We have now validated ePE with both the PE2 and the new Exo-PE strategy on the endogenous loci DNMT1, RNF2, VEGFA, HEK3, RUNX1 (new **Figure 3** and new **Supplementary Figure 5**) as well as ENO1-N, ENO1-C, AHSA1, LDHB, PARP1, PKM, VIM, MIF, and GAPDH (new **Supplementary Figure 8**). We also validated the data from HEK cells in HeLa cells (new **Supplementary Figure 7**).

We find an overall better editing efficacy of the 'Exo-PE' strategy in direct comparison to the 'PE2' strategy without compromise on editing precision, using identical pegRNAs (ANOVA results in the main text, new **Figure 3, Supplementary Figure 7**). We also provide data that the fusion type (N- or C-terminal fusion of RT to nCas9 can have a relevant effect for a given locus of interest. We provide all statistical results in **Supplementary Table 2**).

2. Considering the improvements in on-target editing efficiency of the newly optimized prime editors described in this manuscript, the authors should also evaluate the off-target effects of these improvements. Endogenous off-target effects of the optimized prime editor should be assessed across at least 10 endogenous off-target targets with less than 3 mismatches between the on-target and off-target pegRNAs.

R3-2

We have carefully assessed the editing errors from ePE in the 'Exo-PE' and 'PE2' approaches (new **Supplementary Figure 6**).

We have also assessed the off-target effects for the improved ePE complex for both 'Exo-PE' and 'PE2' in both the N- and C-terminal fusion types (ePE-N, ePE-C) on the well-characterized off-targets (new **Supplementary Table 1**).

We included OT1 of HEK3, which provides a perfect match for the PBS of the reference pegRNA previously used in the literature.

We found that Exo-PE showed increased editing activity on that site compared to PE2. Although it should not be surprising that this excellent target gets edited, we hypothesize in the discussion

section that the competition between the flaps in PE2 may also contribute to an apparent specificity in a locus-dependent fashion, which is now partially removed by the activity of the exonuclease. As an obvious conclusion, pegRNA design should avoid matches with other sites to avoid “off”-target effects for the increasingly more potent prime editors.

2. Can the authors explain why co-expressing the pegRNA with the PCP:superNLS:5'-Exo is more efficient than PE2 in Fig. 4b? The authors should include a more detailed description in the revised manuscript.

R3-2b

Thank you for this detailed observation regarding the original Fig. 4b (now moved to new **Supplementary Figure 3a**). We hypothesize that the co-expression of the non-recruited PCP-exonuclease can also address the 5' overhang, albeit less effectively. This is not surprising since recent publications also showed that even the reverse transcriptase of the PE enzyme does not have to be fused nor actively recruited to the nCas9-pegRNA complex (doi:10.1038/s41587-022-01255-9, doi:10.1038/s41587-022-01473-1) for a functional PE machinery.

3. Can the authors describe why the CAG-driven pegRNAs show lower editing efficiencies compared to the U6-driven pegRNAs with a CG pair in the pegRNA scaffold in Extended Data Fig. 4?

R3-3

The plasmid size of pCAG-driven pegRNA plasmid (5.2 kbp) is bigger than the pU6-driven pegRNA plasmids (3.5 kbp). Since we used equal masses of plasmids during the transfection, a lower molarity of plasmids was transfected. Since this information does not hold any value for the revised manuscript, we removed the respective figure.

4. I suggest the authors introduce the function of and thoroughly describe Sso7d in the context of “we next hypothesized that PE efficacy might benefit from an increased local concentration at nucleosomes without increasing the general expression strength. Therefore, we fused the histone-like 7d protein from the archaea *Sulfolobus solfataricus* (Sso7d with E35L to inactivate cryptic ds/ssRNase activity)²⁸ to the C-terminus of ePE.”

R3-4

Sso7d, as a DNA-binding enhancer, is used by many polymerase vendors, e.g., the Q5 or Phusion polymerase from New England Biolabs. Sso7d was also shown in several publications to enhance general DNA-associated enzymatic activities, such as polymerases, ligases, and integrases, without having detrimental effects on off-target or precision (doi:10.1093/nar/gkh271, 10.1093/protein/gzt024, 10.1093/protein/gzt024). We introduced the

mutation E35L to deactivate the cryptic RNase activity of Sso7d, which is expected to be detrimental to the pegRNA.

Minor concerns: 1. Please clarify "Pv3" in figure 4 legends.

We are sorry about this inconsistency. Pv3 is the same as PP7v3 and will be corrected in the revision.

Once again, we would like to thank the Reviewer for the valuable feedback, which greatly helped us to characterize the performance of Exo-PE. We believe that Exo-PE is a valuable addition to the PE toolbox that should help foster further improvements in gene editing technology.

Second round of Reviewers' Comments:**General remarks to all Reviewers**

We thank all Reviewers again for their thoughtful comments and suggestions, which we could all address in another round of careful experiments and NGS analyses, that helped us tremendously to furthermore benchmark the 'Exo-PE' strategy against the standard techniques 'PE2,3,4,5' on >10 endogenous loci for different insertion lengths and shorter substitutions (please see point-by-point answers below).

The additional editing experiments replicated the improved editing efficacy of 'Exo-PE' for insertions without compromising the editing precision, thus demonstrating 'Exo-PE' as a potent PE technique that is complementary to existing editing techniques optimized for shorter substitutions.

To accommodate the considerable amount of additional data and analyses, we have restructured the main figures and relocated subfigures to the supplemental section (all edits were documented in a head-to-head comparison provided as .pdf).

We have also included complete sequences for all improved PE proteins and optimized pegRNAs for the recruitment of the 5'-3'-exonuclease, which we will provide as freely available resources, to benefit the broader scientific community interested in generating complex genetic edits on a routine basis.

We would like to express our appreciation to the Reviewers for their valuable input. We believe that their insightful comments have significantly contributed to the enhancement of the characterization and presentation of 'Exo-PE'.

Reviewer #1:

Remarks to the Author:

Prime editing (PE) is a powerful CRISPR-Cas9 based precise genome editing technique that has gone since its inception through a plethora of refinement due to low genome editing efficiency. In this resubmitted manuscript, Truong et al explored several approaches of improving prime editing efficiency including refining fusion strategies of different effectors to PE machinery and optimizing pegRNA stability. One major improvement over the first version of the manuscript is the assessment of prime editors also in two different mammalian cell lines (HeLa and HEK293T) at several endogenous target sites and validating results with NGS aside from solely testing optimized prime editors using authors' improved traffic light reporter. While this manuscript takes into consideration several previously published approaches of improving PE, it also contributes novelty to the field with the exonuclease usage.

While the authors tested Exo-PE at several endogenous target sites (with different RT fusion in regards to Cas9n), they compare it solely to PE2 (with different RT fusion in regards to Cas9n). In the manuscript, it is explained why the PE3 strategy was not tested, however, I believe it would be appropriate to also compare it to PE4 (PE2 + MLH1dn) strategy previously published (Chen et al, 2021). PE4 on average improves PE efficiency approximately 2-fold over PE2 (Chen et al, 2021) bringing increase in PE editing with Exo-PE on par with "predicted" PE4 activity at several endogenous target sites tested. Therefore, to improve Exo-PE editing

efficiency above previously published iterations of PE technique, it would be prudent to test Exo-PE4 (Exo-PE + MLH1dn) performance. Based on the current data, the Exo-PE system is mostly on par with PE3 for editing efficiency. Without comparison with the best PE systems that are beyond PE3, I am not sure this technology represents a breakthrough for PE technology.

The authors have expanded the methods section significantly; however, I cannot find any information about depositing newly developed PE systems (at least Exo-PE –N, –C) to depository. Furthermore, plasmid maps are still missing as well as list of vectors and primers used in this study.

I appreciate the authors conducted additional experiments on endogenous target sites at two different cell lines. Overall, this work addresses one aspect of prime editing by more efficiently removing the 5' flap using exonuclease, which is novel. Whether Exo-PE represents a better PE system that others will use? Well, I am not sure. I will leave it to the editor to decide whether such novelty merits publishing at Nature Methods.

R1-11

We apologize for the missing plasmid maps and sequence information in our previous submission. We are committed to making our work broadly accessible and will certainly deposit the full 'Exo-PE' system to Addgene including all optimized pegRNA/sgRNA/amino acid sequences (**Supplementary Table 3**).

We have now included comparisons to the 'PE4' strategy (PE2 + MLH1dn) in our revised manuscript, based on your suggestion.

As the Reviewer mentioned, 'PE4' has been reported to improve prime editing efficacy by approximately 2-fold over 'PE2' (Chen et al., 2021). In our study, we found that 'PE4' indeed improved editing efficacy for small edits in some cases, but it did not significantly improve the efficacy of insertions compared to 'PE2' (**Figure 2b,c**), which is the main application for which we developed 'Exo-PE'.

We re-confirmed that also in the additional loci that we compared against 'PE2/PE4' and 'PE3/PE5', 'Exo-PE' shows a consistently increased editing efficacy for ≥ 30 bp insertions compared against 'PE2' (**new Figure 2c**).

We also tested the 'Exo-PE' + MLH1dn and found that it did not provide any additional improvement beyond what we observed with 'Exo-PE' (**new Figure 2b**).

In contrast, we found a substantial increase in error rates for 'PE3' vs. 'PE2' for 30 bp insertions (**new Figure 2c**) as well as for substitutions (**Supplementary Figure 10**) in line with previous publications (Chen et al., 2021, Anzalone et al., 2022).

These additional data jointly demonstrate the value of 'Exo-PE' as a potent editing tool that is complementary to 'PE4', with 'Exo-PE' showing superior performance for larger edits and 'PE4' for smaller edits of 'PE2', while both methods maintain the editing precision of 'PE2'.

Reviewer #2:

Remarks to the Author:

Truong et al. present a revised manuscript that addresses many of the original comments, producing data to support characterization of the new prime editing strategy EXO-PE at some endogenous sites in different cell types. For testing application of this new strategy, they targeted several endogenous loci by inserting a short split fluorescence peptide to enable biomolecular fluorescence complementation. However, some concerns still remain.

Major concerns

1. As a newly developed tool, EXO-PE should be compared with some of the well-characterized prime editors, at least for the case of PE4/PE5 which is mentioned by different reviewers, in different endogenous loci (at least 10 loci) by targeted deep sequence but not in the report system. Though the two strategies are involved in different pathways, the comparative information will be useful for application of EXO-PE, and crucial to further improvements of PE system. In addition, it is interesting to know whether combining EXO-PE with PE4/PE5 could further improve the efficacy of PE system.

R2-9

In response to your request, we have expanded our NGS benchmarking to now also include comparisons to 'PE3', 'PE4', and 'PE5' at additional endogenous loci (totaling 11 loci tested for insertions and 8 for substitutions) in **new Figure 2b,c, new Supplementary Figures 11,12**.

Furthermore, we have also tested 'Exo-PE' in combination with MLH1dn ('Exo-PE4') but found that it did not lead to any improvement beyond 'Exo-PE alone' (**new Figure 2b**). We also had previously tried to combine 'Exo-PE' also PE3 in the traffic light reporter system, which showed no significant benefit for efficacy (**Supplementary Fig. 3b**), such that we decided that it was too unlikely that 'Exo-PE5' would be useful on other loci given the poor effect of 'Exo-PE4' over 'Exo-PE', which already preserved the editing precision from 'PE2'.

2. Better representation of the technology is needed. The authors stated that "v3 showed a trend towards improved efficacy while maintaining improved editing precision in 'Exo-PE' (Supplementary Fig. 3b)", which should be confirmed at more sites, this is vital for further improvement and utilization of EXO-PE. Crucial characterization of EXO-PE was too less shown in the main figures. We suggest to depict the architecture of ePE and the design of variants for the PP7-aptamer-tagged pegRNA in Fig. 2, it would be easier and faster to get the key info for the readers.

R2-10

We thank the Reviewer for suggesting a more concise representation of the series of modifications that we made to the different components of the PE complex and carefully re-evaluated at each step of the validation pipeline to ensure optimal performance for the 'Exo-PE' strategy.

Also complying with the limited spaces for main figures in the Brief Communication format, we now depict the PE system optimized for 'Exo-PE' in the **new Figure 2a**.

To further ensure a more concise summary of the optimizations of engineered pegRNA with a PP7 aptamer, we have re-combined the relevant graphs in the **new Supplementary Figure 3**, together with additional schematics of the optimization steps. As noted in the main text, the different PP7 versions did not show a significant difference, but a slight positive trend, such that we chose PP7v3 for subsequent experiments (data and schematics in **Supplementary Figure 3b**).

3. Key components (i.e., Sso7d) and key mutations (i.e., N863A, K918N) of EXO-PE were only introduced without explanations or proper references in the context. For example, why N863A is used in some constructs, but H840A in others. Sso7d and K918N should be clearly described and referenced in the main text when mentioned at the very first time. The mutations of MLV RT hexamutant (x6) in Fig. S2a need to be specified, please note that a pentamutant RT is used in PE2/PE3/PEmax (PMID: 31634902).

R2-11

We apologize for the missing reference for the K918N mutation which has been added to the current manuscript draft. K918N and also SSo7d initially showed benefit for eTLR editing and were therefore carried on for a limited set of experiments. Because this benefit was inconsistent during our careful re-evaluations on eTLR and when re-testing on other loci, we decided to omit those modifications for NGS benchmarking, to ensure that the 'Exo-PE' system only harbors the minimal set of effective improvements.

We used H840A and N863A due to the different origins of the original vectors encoding Cas9 RuvC nickase, but they are equivalent to converting Cas9 nuclease into a Cas9 RuvC nickase by HNH domain inactivation ([doi:10.1038/srep38198](https://doi.org/10.1038/srep38198)).

Penta- or hexamutant both describe the same amino acid sequence of the Moloney murine leukemia virus reverse transcriptase. The difference in terminology originated from the fact that the reference sequence Anzalone *et al.* used and described as WT already contained the not described H8Y mutation (<https://www.uniprot.org/uniprotkb/P03355>).

This mutation has also been described beforehand in various patents ([doi:10.1016/j.csbj.2021.11.030](https://doi.org/10.1016/j.csbj.2021.11.030)).

We now include **Supplementary Table 3** as .xls showing all AA, pegRNA, and sgRNA sequences optimized for the 'Exo-PE' strategy.

Minor concerns:

1. We would suggest making a more clarity naming. Some names in the text could be mistaken for published tools, such as PPE (plant prime editor, PMID: 33767395), ePE (enhanced prime editing system, PMID: 34103663).

R2-12

Thank you very much for this suggestion. To avoid confusion with the current literature's terminologies of various prime editing strategies and engineered variants, we changed 'ePE' to 'iPE' (i for improved). Instead of PPE for paired prime editing, we will use 'paired PE.'

2. Please clarify "Cryptic splice donor sites" in Fig. S2a.

R2-13

We apologize for the missing explanation for 'cryptic' splice sites. The original unoptimized prime editing enzyme contained various strong splice motifs, which may lead to aberrant unwanted splicing within the coding sequence or occupation by splice factors (SFs) and subsequent nuclear retention of unspliced but SFs-occupied mRNAs. According to NetGene2, whose predictions were found useful in other work ([doi:10.1038/s41556-022-00998-6](https://doi.org/10.1038/s41556-022-00998-6)), these sites in the original PE2 enzyme (<https://www.addgene.org/132775/>) have a high potential as cryptic splice donors: TGGGAG|GTGACT, AACAAAG|GTACTC, GACAGC|GTAAGG, ACCGAG|GTAATC, CAGAAG|GTAAGA.

We extended our explanation regarding these cryptic splice sites in the main text.

3. "HEK293 eTLR cells (ECACC: 12022001, Sigma-Aldrich)", make sure the corresponding catalog number is right?

R2-14

We apologize for the wrong catalog number. #12022001 refers to HEK293T cells. HEK293 eTLR cells are derived from HEK293 cells without the SV40 large T-antigen, which has catalog number #85120602. We have now corrected the corresponding paragraph.

Reviewer #3:

Remarks to the Author:

Truong et al. demonstrated that 'Exo-PE' showed overall better editing efficacy than the 'PE2' strategy for ≥ 30 bp insertions without compromising editing precision. Compared to their previously submitted manuscript, the authors performed additional comparisons between Exo-PE and PE2 to demonstrate the superior editing efficiency of Exo-PE when editing endogenous sites. While PE3 typically edits more efficiently than PE2 and PE3 tends to generate more indels as well; it is interesting that the authors found Exo-PE to show a better balance between editing efficiency and accuracy. I recommend the authors perform additional head-to-head comparisons between Exo-PE and PE3 at many endogenous sites and then explore calculating an efficiency/accuracy ratio. I believe that this data is essential to demonstrate a superior therapeutic potential for Exo-PE.

Moreover, previous reports have demonstrated that the repair mechanism between point mutations and indels may be different and that these differences may serve as insightful strategies to further improve the editing of prime editors (Chen et al. Cell. 2021). (For example, they found that "the MLH1dn enhancement of prime editing efficiency declined as the length of the indels increased, consistent with previous reports that MMR repairs IDLs up to 13 nt in length"). Thus, since the authors only tested two types of fragment insertions when comparing Exo-PE and PE2, the authors should perform additional comparisons for single base substitutions across a number of endogenous sites.

R3-5

Thank you for the additional comments and helpful suggestions.

As described in the manuscript, we had intentionally focused on a direct comparison of 'Exo-PE' to the single-pegRNA 'PE2' strategy, in order to avoid additional 'degrees of freedom' related to a secondary nicking site. Furthermore, we had already with the construction of the eTLR FACS reporter system focused on longer insertions because 'Exo-PE' was likely to show the strongest benefits for this challenging editing type, for which flap competition is the most severe.

In response to the Reviewer's request, we have now also performed additional comparisons on 4 additional loci between 'Exo-PE' and 'PE3' to evaluate insertions (**new Figure 2c**) as well as single-base substitutions (**new Supplementary Figure 11a**).

For insertions, we observed a substantial increase in the 'PE3' InDel rates for the 30 bp insertions, similar to what was described before (Anzalone et al. 2022, <https://www.nature.com/articles/s41587-021-01133-w/figures/2>) in comparison to the better-characterized error rates for 'PE3' on shorter substitutions (Anzalone et al. 2019, Chen et al. 2022)

In contrast, 'Exo-PE' again showed an increase in correct insertions over 'PE2' on the same loci, without losing the editing precision of 'PE2' (**New Figure 2b,c**).

Upon the Reviewer's request, we have also conducted an evaluation of 'Exo-PE' on single-base substitutions. As expected for the much less severe flap competition for this edit

type, 'Exo-PE' did not show a substantial benefit in editing efficacy or precision over 'PE2' for most (6/8) loci (**new Supplementary Figure 11**)

Furthermore, we also compared 'Exo-PE' to 'PE4' and 'PE5' (= 'PE2'/'PE3' + MLH1dn expression) for 30 bp insertions as well as single-base substitutions. In line with previous reports (Chen et al., 2021), we found that MMR inhibition via MLH1dn was sometimes beneficial for base substitutions but not for insertions (**new Figure 2b, new Supplementary Figure 1**). MMR inhibition had, however, no additional benefits when combined with 'Exo-PE' ('Exo-PE4').

We again thank the Reviewer for requesting us to run these additional experiments and analyses, as we think that these findings now jointly show that 'Exo-PE' is a PE strategy with complementary properties to 'PE2' and 'PE3', and will be most valuable for introducing mid-range insertions.

Minor comments:

1. In Sup Fig. 2e, it's interesting that the MarathonRT showed a high editing efficiency in their eTLR reporter system. I suggest the authors compare this RT to the Moloney Murine Leukemia Virus RT in both a reporter system and across many endogenous sites.

R3-6

The data shown in Supplementary Figure 2e came from initial tests with a plasmid-based reporter via co-transfection together with the components encoding the prime editing components. Thus, the efficacy was much higher compared to the integrated reporter system.

In the meantime, Grünewald et al. showed a thorough characterization of MarathonRT and its engineered mutants and found that even the best MarathonRT mutant underperformed when compared directly with an MLV RT-based prime editor (doi:10.1038/s41587-022-01473-1), supporting our decision to stick with MLV RT.

2. In Sup Fig. 3e, what does PP7v3/e mean?

R3-7

We apologize for any ambiguity in our labels. We have simplified the nomenclature for the modification with the variants of the PP7 loop and the evopreQ1 modifications and now refer to the modified pegRNA as PP7v3/Q1 and Q1.

3. In Sup Fig. 3e, the lines for P value calculation are misaligned. Please modify.

R3-8

Thank you for spotting this formatting error. We have now aligned the characters properly.

We would like to thank all reviewers again for their valuable comments and suggestions, which significantly improved the characterization and presentation of the 'Exo-PE' strategy.

Decision Letter, second revision:

Dear Gil,

Thank you for submitting your revised manuscript "Exonuclease-enhanced prime editors" (NMETH-BC47950D). It has now been seen by the original referees and their comments are below. The reviewers find that the paper has improved in revision, and therefore we'll be happy in principle to publish it in Nature Methods, pending minor revisions to satisfy the referees' final requests and to comply with our editorial and formatting guidelines.

TRANSPARENT PEER REVIEW

Nature Methods offers a transparent peer review option for new original research manuscripts submitted from 17th February 2021. We encourage increased transparency in peer review by publishing the reviewer comments, author rebuttal letters and editorial decision letters if the authors agree. Such peer review material is made available as a supplementary peer review file. Please state in the cover letter 'I wish to participate in transparent peer review' if you want to opt in, or 'I do not wish to participate in transparent peer review' if you don't. Failure to state your preference will result in delays in accepting your manuscript for publication.

ORCID

Sincerely,
Madhura

Madhura Mukhopadhyay, PhD
Senior Editor
Nature Methods

Reviewer #1 (Remarks to the Author):

The authors have addressed my concerns and I have no more comments.

Reviewer #2 (Remarks to the Author):

Overall, the authors did a nice job answering the queries, and the manuscript is much improved. With some minor modifications, this work would be acceptable for publication.

The strategy for codon optimization of RT (including the removal of the potential splice sites) should be included in the main text or Methods section.

A recent work (PMID: 36958601) reported the increased PE efficiency in plants using the same strategy by introduction of the T5 exonuclease. The authors should state this in the discussion section.

If multiple panels were contained in a figure (e.g., Supplementary Fig. 1), they must be numbered specifically and consecutively as they appear in the text.

Reviewer #3 (Remarks to the Author):

The authors have conducted additional experiments and made revisions to their manuscript, effectively addressing all the concerns raised. However, I observed that PE3 exhibits a significantly high frequency in generating indels, whereas precise prime editing at the endogenous sites is almost non-existent. This finding contradicts previous reports, such as the study referenced by the DOI: 10.1038/s41586-019-

1711-4. In my opinion, the authors should provide a more detailed explanation regarding the disparities in the results.

Final Decision Letter:

Dear Gil,

I am pleased to inform you that your Article, "Exonuclease-enhanced prime editors", has now been accepted for publication in Nature Methods. The received and accepted dates will be 13th January 2022 and 19th December 2023. This note is intended to let you know what to expect from us over the next month or so, and to let you know where to address any further questions.

Acceptance of your manuscript is conditional on all authors' agreement with our publication policies (see <https://www.nature.com/natsustain/info/gta>). In particular your manuscript must not be published elsewhere and there must be no announcement of the work to any media outlet until the publication date (the day on which it is uploaded onto our website).

Over the next few weeks, your paper will be copyedited to ensure that it conforms to Nature Methods style. Once your paper is typeset, you will receive an email with a link to choose the appropriate publishing options for your paper and our Author Services team will be in touch regarding any additional information that may be required.

You will receive a link to your electronic proof via email with a request to make any corrections within 48 hours. If, when you receive your proof, you cannot meet this deadline, please inform us at rjsproduction@springernature.com immediately.

Please note that *Nature Methods* is a Transformative Journal (TJ). Authors may publish their research with us through the traditional subscription access route or make their paper immediately open access through payment of an article-processing charge (APC). Authors will not be required to make a final decision about access to their article until it has been accepted. [Find out more about Transformative Journals](https://www.springernature.com/gp/open-research/transformative-journals)

Authors may need to take specific actions to achieve [compliance](https://www.springernature.com/gp/open-research/funding/policy-compliance-faqs) with funder and institutional open access mandates. If your research is supported by a funder that requires immediate open access (e.g. according to [Plan S principles](https://www.springernature.com/gp/open-research/plan-s-compliance)) then you should select the gold OA route, and we will direct you to the compliant route where possible.

For authors selecting the subscription publication route, the journal's standard licensing terms will need to be accepted, including [self-archiving policies](https://www.springernature.com/gp/open-research/policies/journal-policies). Those licensing terms will supersede any other terms that the author or any third party may assert apply to any version of the manuscript.

Your paper will now be copyedited to ensure that it conforms to Nature Methods style. Once proofs are generated, they will be sent to you electronically and you will be asked to send a corrected version within 24 hours. It is extremely important that you let us know now whether you will be difficult to contact over the next month. If this is the case, we ask that you send us the contact information (email, phone and fax) of someone who will be able to check the proofs and deal with any last-minute problems.

If, when you receive your proof, you cannot meet the deadline, please inform us at rjsproduction@springernature.com immediately.

Once your manuscript is typeset and you have completed the appropriate grant of rights, you will receive a link to your electronic proof via email with a request to make any corrections within 48 hours. If, when you receive your proof, you cannot meet this deadline, please inform us at rjsproduction@springernature.com immediately.

To assist our authors in disseminating their research to the broader community, our SharedIt initiative provides you with a unique shareable link that will allow anyone (with or without a subscription) to read

the published article. Recipients of the link with a subscription will also be able to download and print the PDF.

Nature Portfolio journals [encourage authors to share their step-by-step experimental protocols](https://www.nature.com/nature-research/editorial-policies/reporting-standards#protocols) on a protocol sharing platform of their choice. Nature Portfolio 's Protocol Exchange is a free-to-use and open resource for protocols; protocols deposited in Protocol Exchange are citable and can be linked from the published article. More details can found at www.nature.com/protocolexchange/about.

Best regards,
Madhura

Madhura Mukhopadhyay, PhD
Senior Editor
Nature Methods